# AXL receptor tyrosine kinase regulates Golgi organization and function via an adhesion-Arf1 signalling axis in breast and lung cancer cell lines

Prachi Joshi*,‡, Arnav Saha‡, Radhika Malaviya, Debiprasad Panda, Grishma Mehta, Manojeet Pattanayak, Vibha Singh and Nagaraj Balasubramanian§

## ABSTRACT

Cell-matrix adhesion regulates membrane trafficking, Golgi organization, and function. Altered Golgi organization in cancer cells may influence trafficking and cargo processing. A simple screen revealed distinct, adhesion-dependent differences in Golgi organization across breast (MDAMB231 versus MCF7) and lung (A549 versus CaLu1) cancer cell lines. To identify regulators driving these differences, we performed an *in silico* analysis of differentially expressed genes in the Cancer Cell Line Encyclopedia dataset, integrating Golgi-associated functions from interaction networks and literature. This analysis highlighted AXL as a putative Golgi regulator. AXL is prominently localized to the Golgi and is displaced upon inhibition with R428, which disrupts Golgi organization. AXL knockdown also does the same. AXL-mediated regulation of the Golgi is adhesion dependent. Mechanistically, AXL controls Arf1 activation through an AMPK-GBF1 pathway. Targeting of AMPK activation thus significantly reverses R428-mediated Golgi disorganization. Loss of adhesion promotes AMPK and reduces Arf1 activity, displacing AXL and Arf1 from the Golgi, driving its disorganization. This impacts Golgi-associated functions, tubulin acetylation in MDAMB231 cells, and cell-surface glycosylation in A549 cells. Together, our findings identify an adhesion-AXL-AMPK-GBF1-Arf1 pathway governing Golgi organization and function in cancer cells.

KEY WORDS: Adhesion, Golgi organization, AXL, Arf1, Golgi function, Cancer

## INTRODUCTION

Cell-matrix adhesion, essential for the survival, growth, and proliferation of all eukaryotic cells, is often deregulated in pathologies such as cancer (Berrier and Yamada, 2007; Reddig and Juliano, 2005). Cancer cells overcome the need for cell-matrix adhesion, promoting anchorage-independent growth and metastasis (Janiszewska et al., 2020; Reddig and Juliano, 2005). Among the changes that drive this include adhesion-independent growth

signalling (Pawar et al., 2016) and changes in protein glycosylation (Läubli and Borsig, 2019; Rambaruth and Dwek, 2011). Sustained activation or trafficking of growth receptors to the plasma membrane (PM) without adhesion also promotes anchorage independence (Pawar et al., 2016; Schwartz, 1997). Alterations in glycosylation, help overcome loss of adhesion-mediated cell death (anoikis) in cancers (Petrosyan et al., 2014; Piyush et al., 2017). Glycan alterations are driven by changes in expression or localization of the glycosylation machinery and Golgi organization, the hub for these reactions (Bhat et al., 2017; Petrosyan, 2015).

In mammalian cells, the Golgi has multiple cisternae, which stack up in a defined sequence (cis-, medial and trans-Golgi), laterally linked to form a Golgi ribbon (Marsh and Howell, 2002; Nakamura et al, 2012; Rambourg and Clermont, 1997). Together with the ER-Golgi intermediate compartment (ERGIC) and the trans-Golgi network (TGN), the Golgi functions as a dynamic site for post-translational modifications (PTMs) and trafficking events (Huang and Wang, 2017; Ward et al., 2001). These depend on several Golgi-associated regulatory pathways, including the organization of the Golgi ribbon (Petrosyan, 2019; Stanley, 2011; Zhang and Wang, 2015). Alterations in these pathways and Golgi organization could change trafficking and glycosylation, to facilitate cancer progression (Petrosyan, 2015). Golgi organization is often reported to be inherently disorganized in several cancers (Kellokumpu et al, 2002; Petrosyan et al., 2014; Rivinoja et al., 2006), though its role and regulation are not yet completely understood (Bajaj et al., 2022; Bhat et al., 2017; Bui et al., 2021; Howley and Howe, 2018).

While Golgi organization and adhesion-dependent signalling are altered in cancer cells, earlier studies from our laboratory have shown that adhesion can also regulate Golgi organization (Singh et al, 2018). In non-transformed cells, loss of adhesion-mediated Golgi disorganization is regulated by the small GTPase Arf1, affecting its recruitment of Dynein motor protein at Golgi membranes. This is accompanied by distinct changes in cell surface glycosylation in these non-adherent cells. This suggests a cell adhesion-Arf1-Dynein-Golgi organization-Golgi function pathway in normal cells (Singh et al., 2018). If and how this adhesion-mediated regulation of Golgi is perturbed in anchorage-independent cancers remains unknown. Golgi organization in cancers is variable, with some cell lines having a 'dispersed' or 'fragmented' Golgi and others having a 'normal' intact Golgi (Bhat et al., 2017; Petrosyan, 2015; Petrosyan et al., 2014). Amongst the known regulators of Golgi organization, which could be perturbed in cancers, are Golgi matrix proteins (Witkos and Lowe, 2015; Xiang and Wang, 2011; Zhang and Wang, 2015), Golgi-associated GTPases (Goud et al, 2018; Thomas and Fromme, 2020; Ward et al., 2001), cytoskeletal proteins (Egea et al., 2015; Kulkarni-Gosavi et al, 2019), and kinases (Chia et al., 2012; Kimura et al, 2018; Mao et al., 2013; Weller et al., 2010). Cancer cells

Department of Biology, Indian Institute of Science Education and Research Pune, Pune 411008, India.
*Present address: Yale University Cardiovascular Research Center, New Haven, CT, USA.
‡These authors contributed equally to this work

§Author for correspondence (nagaraj@iiserpune.ac.in)

P.J., 0000-0002-0503-2701; A.S., 0009-0008-7801-4839; R.M., 0009-0002-4525-1244; N.B., 0000-0002-8219-8844

of similar origin with different Golgi organizations could constitute a unique system to evaluate this regulation and possibly identify novel regulators that could mediate it. Our simple screen in breast (MDAMB231 and MCF7) and lung cancer (A549 and Calu1) cell lines with organized versus dispersed Golgi allows us to evaluate differentially expressed genes (DEGs) as possible candidate regulators of the Golgi.

This led us to AXL, a top candidate in both breast and lung cancer cell lines. AXL is a transmembrane receptor tyrosine kinase from the TAM family and plays a significant role in cell proliferation, survival, adhesion, and apoptosis (Auyez et al., 2021; Wium et al., 2021). It was first discovered as an oncogene, and its role in cancer progression has been extensively researched. AXL overexpression is seen in several human malignancies, including breast cancer (Holland et al., 2010; Zajac et al., 2020), acute myeloid leukaemia (Hong et al., 2008), non-small cell lung cancer (NSCLC) (Iida et al., 2017; Zhang et al., 2012), ovarian cancer (Kanlikilicer et al., 2017), glioblastoma (Onken et al., 2016), and neuroblastoma (Debruyne et al., 2016). Altered AXL expression or activation promotes invasiveness and metastasis, epithelial-to-mesenchymal transition, drug resistance, and anchorage-independent growth (Debruyne et al., 2016; Lay et al., 2007; Scaltriti et al, 2016; Taniguchi et al., 2019; Ye et al., 2010). AXL is activated by its ligand Gas6 at a one-to-one receptor-to-ligand ratio and then dimerizes to promote trans-autophosphorylation and initiate downstream signalling (Tanaka and Siemann, 2021). Amongst the six phosphorylation sites on AXL, the three C-terminal sites – Tyr698, Tyr702, and Tyr703 – are relatively conserved and vital for its function (Auyez et al., 2021; Zhu et al., 2019).

A recent study showed that inhibition of AXL affects its localization at the Golgi and impairs directed migration in breast cancer cells (Hs578t) (Zajac et al., 2020). A screen for regulators of the Golgi organization targeting the kinome and phosphotome in HeLa cells also identified AXL, the knockdown of which caused the Golgi to become more condensed (Chia et al., 2012). AXL has also been reported to associate with the Golgi-localized GTPase Arf1 in breast cancer cells (Haines et al., 2015).

Our findings position AXL as a vital regulator of Golgi organization and function in cancer cells. Its adhesion-dependent control of Golgi organization and function highlights a broader physiological role. Moreover, the coordinated recruitment of AXL and active Arf1 to the Golgi, together with their functional crosstalk, defines a critical mechanistic axis that governs Golgi dynamics.

## RESULTS

### Adhesion-independent regulation of Golgi organization in cancers

Adhesion-dependent signalling regulates Golgi organization in anchorage-dependent cells (Singh et al., 2018). Cancer cells, by acquiring anchorage independence, bypass adhesion-dependent signalling and evade regulatory control to sustain their growth and survival. This involves deregulation of adhesion-dependent pathways, raising the question of whether similar mechanisms also govern Golgi organization and function in cancer cells. A simple screen for Golgi organization across multiple cancer cell lines using a trans-Golgi marker (GalTase-RFP) showed much variation across cell lines. Breast cancer MDAMB231 and lung cancer A549 cells showed a predominantly organized Golgi phenotype, while in breast cancer MCF7 and lung cancer CaLu1 cells the Golgi is predominantly disorganized (Fig. 1A). On loss of adhesion MCF7 (Fig. 1C) and CaLu1 (Fig. 1E), cells retained their disorganized Golgi, detected using the cis-medial (MannosidaseII-GFP) and trans-Golgi marker (GalTase-RFP). In A549 cells, the Golgi stays intact on loss of

adhesion (Fig. 1D), unlike the non-transformed lung epithelial BEAS2B cells, in which it is dispersed (Fig. S1A). In non-adherent MDAMB231 cells, the Golgi is dispersed (Fig. 1B) as seen in the non-transformed breast epithelial MCF10A cells (Fig. S1B,C). Thus, adhesion-dependent regulation of Golgi is also variable in different cancer cell lines even from similar tissue origins. We hence asked how these differences in Golgi organization could be regulated and affect cancer cell function.

### DEGs in breast and lung cancer cells – potential regulators of Golgi organization?

Differences in Golgi organization seen in breast and lung cancer cells could be regulated by differences in gene expression. To evaluate this, we designed an in silico comparative screen of DEGs in MDAMB231 versus MCF7 (breast), and A549 versus CaLu1 (lung) (Fig. 2A). Using the NCBI Gene Ontology database and a collated list of known regulators of Golgi organization and/or function reported in literature, we arrived at a list of 390 possible genes that could affect Golgi organization. The Cancer Cell Line Encyclopedia (CCLE) database was used to compare the differential mRNA expression of these regulators between breast and lung cancer cells. This revealed 42 DEGs in breast cancer and 35 DEGs in lung cancer that show a >10- and >5-fold change in expression, respectively. The fold change cut-off applied was adjusted for breast and lung cancer, such that the genes to be evaluated were in their respective top 50. A scoring system was further applied to these shortlisted DEGs. The first score component is based on the effect knockdown of the gene has on Golgi organization in literature. The second score component is based on the number of primary interactors of the gene in the STRING database that are known to regulate Golgi organization and/or function. Although several Golgi-resident or Golgi-associated genes – GOLGA2, GORASP1, GORASP2, AURKA, etc. – appeared in our in silico analysis, their low combined scores precluded their consideration as potential candidates for further testing. Setting a cut-off of 3 or more for this combined score, we ranked 20 (breast cancer) and 15 (lung cancer) genes as candidate Golgi organization regulators (Fig. 2A). Eight of these genes were common between breast and lung cancer cells. Among the top three candidates in both cancers, AXL was chosen for this study (Fig. 2B).

In concurrence with the CCLE data, AXL expression (mRNA and protein) was higher in MDAMB231 cells than in MCF7 cells (Fig. 2C) and in CaLu1 cells than in A549 cells (Fig. 2D). MCF7 cells have almost negligible expression of AXL (Fig. 2C). Endogenous AXL localization in stable adherent MDAMB231 and A549 cells, in which the Golgi is intact, shows a strong overlap with the cis-Golgi marker (GM130), cis-medial Golgi marker (ManII-GFP) and trans-Golgi marker (GalTase-RFP) (Fig. 2E,F; Fig. S2A,B). In CaLu1 cells, the dispersed cis-Golgi, cis-medial and trans-Golgi all show partial overlap with AXL (Fig. 2G). This suggested us to test the role of AXL in Golgi organization.

### AXL mediated regulation of Golgi organization in breast cancer cells

MDAMB231 (high AXL) and MCF7 (almost no AXL) are ideal for comparing the role of AXL in Golgi organization. Inhibition of AXL using increasing concentrations of R428 (Bemcentinib), a selective ATP competitive inhibitor of AXL, causes both cis- and trans-Golgi to be distinctly disorganized in stable adherent MDAMB231 cells (Fig. 3A,B). AXL is known to regulate Akt activation (Holland et al., 2010), which is also seen to drop on R428 treatment (Fig. 3C). This is accompanied by a concentration-dependent increase in AXL phosphorylation (Y702) levels (Fig. 3D). While some reports show

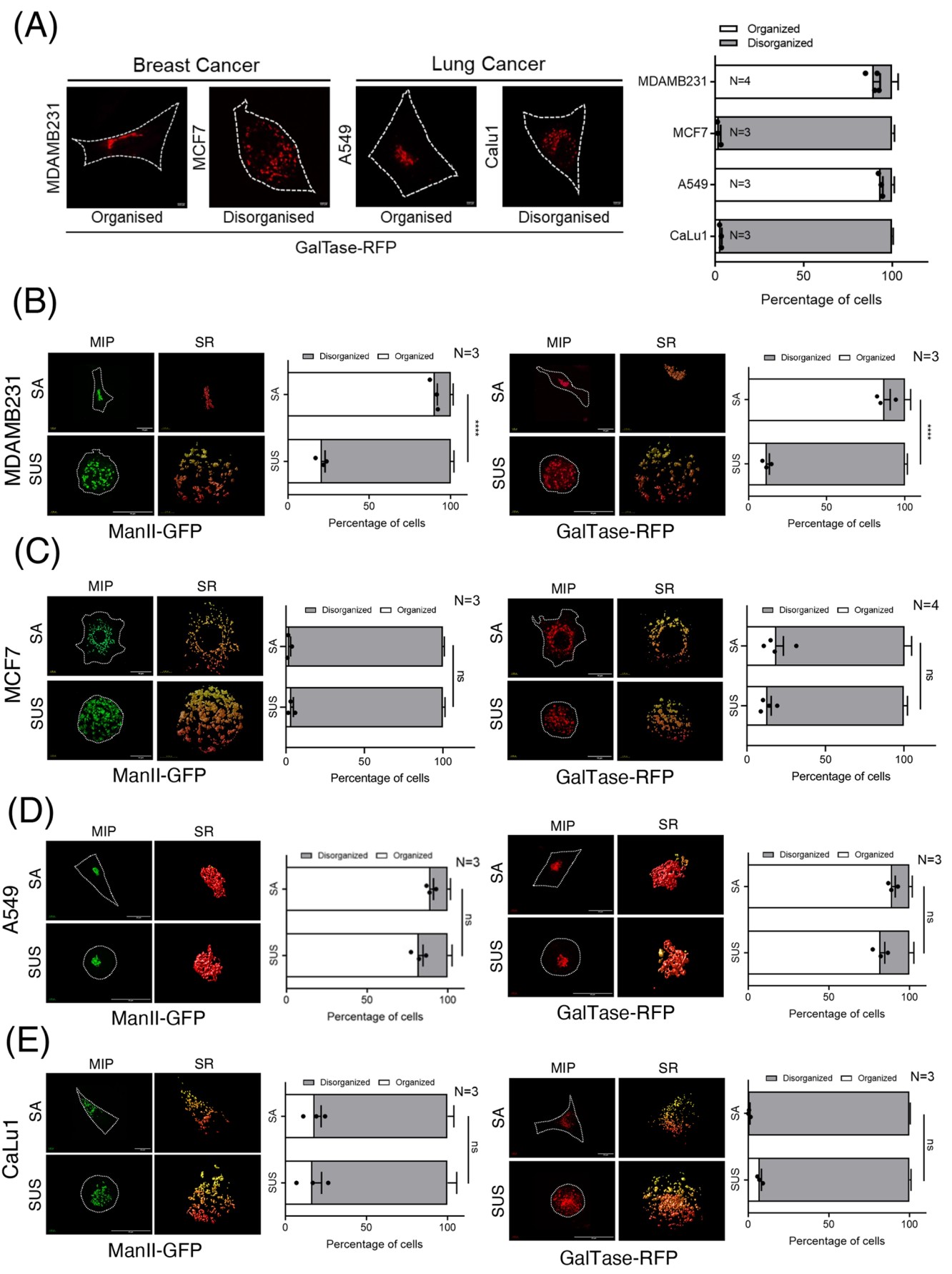

**Fig. 1.** See next page for legend.

**Fig. 1. Adhesion-independent regulation of Golgi organization in breast and lung cancer cells.** (A) Stable adherent MDAMB231, MCF7, A549 and CaLu1 cells transfected with GalTase-RFP. Representative cross-section images show the predominant Golgi organization phenotype. Percentage distribution profile of cells ($n \geq 200$) shows organized (white) and disorganized (grey) Golgi in cells. Graph represents mean±s.e.m. from three to four independent experiments. (B-E) Stable adherent (SA) and non-adherent (SUS) MDAMB231, MCF7, A549 and CaLu1 cells transfected with ManII-GFP or GalTase-RFP. Representative deconvoluted images show the predominant phenotype as maximum intensity projection (MIP) and a zoomed image of the Golgi with surface rendering (SR). Percentage distribution profiles of cells ($n \geq 200$) show organized (white) and disorganized (grey) Golgi in cells. Graphs represent mean±s.e.m. of percentage distribution from three to four independent experiments. Statistical analysis done using one-way ANOVA multiple comparisons test with Tukey's method for error correction. Scale bars: 4.22 µm (A) and 10 µm (B-E). (***$P \leq 0.001$; ns=not significant).

R428 to decrease AXL phosphorylation at Tyr702 (Iida et al., 2017), Tyr779 (Ghosh et al., 2011) and Tyr821 (Holland et al., 2010), another study shows a variable effect (drop and increase) on Tyr702 (Chen et al, 2018). R428 treatment time kinetics also shows a gradual increase in AXL phosphorylation (Y702) levels becoming significant over time (Fig. S3A). Such a R428 treatment time kinetics also shows a time-dependent increase in Golgi disorganization (Fig. S3C). pAkt levels drop and stay low across these treatment timepoints (Fig. S3B). The sustained drop in pAkt (Fig. S3B) on R428 treatment indicates early Akt inhibition, which, however, seems temporally disconnected from Golgi disorganization (Fig. S3C). This suggests that AXL possibly regulates the Golgi independently of Akt, which independent Akt inhibition studies have also confirmed (Saha et al, 2026). In AXL-lacking MCF7 cells, R428 treatment does not affect the Golgi (Fig. 3E) or pAkt (Fig. 3F), confirming a role for AXL. The phospho-AXL (Y702) detection in MDAMB231 cells, is absent in MCF7 cell lysates, confirming antibody specificity (Fig. S3D). We hence use loss of Akt activation and an increase in AXL(Y702) phosphorylation as readouts of R428 action, noting that phospho-AXL(Y702) changes are correlative at best with AXL inhibition in these studies.

siRNA-mediated knockdown of AXL, using two previously verified individual siRNA sequences (Holland et al., 2010), caused a significant and comparable reduction in AXL levels and disorganization of the Golgi in MDAMB231 cells (Fig. 3G). Loss of AXL is further seen to cause a loss in phospho-AXL (Y702) levels (Fig. S3D). AXL knockdown also expectedly caused a drop in Akt activation (Fig. S3E). These findings indicate that AXL levels and its activation state are critical for maintaining Golgi organization in these cells.

We further tested if R428 treatment and AXL knockdown-mediated disruption of Golgi organization can affect its basal functions. The Golgi is a major hub for nucleating non-centrosome microtubules (Chabin-Brion et al., 2001), the stabilization of which is regulated by distinct post-translational modifications such as acetylation (Eshun-Wilson et al., 2019). This could be affected by Golgi organization (Brodsky et al., 2022; Sanders and Kaverina, 2015). Acetylated tubulin levels showed a significant drop on R428 treatment (Fig. 3H) as well as AXL knockdown (Fig. 3I), in MDAMB231 cells (Fig. 3A,B,G). This suggests that AXL expression and activation could both contribute to regulating Golgi organization, and possibly function.

## AXL-Arf1 signalling axis regulates Golgi organization in adherent MDAMB231 cells

In MDAMB231, R428 treatment caused AXL to be displaced from the Golgi, and reflected in a significant reduction of its colocalization with cis-Golgi marker GM130 (Fig. 4A). GM130 levels were unaffected by R428 treatment (Fig. S4A). Earlier studies, including ours, have shown active Arf1 localization at the Golgi to be crucial for its organization and function (Rajeshwari et al., 2023; Ward et al., 2001). Arf1 levels and activation are also known to be deregulated in cancers (Casalou et al, 2016; Casalou et al, 2020; Xie et al., 2016). In MDAMB231 cells, Arf1 is also shown to bind AXL in immunoprecipitation studies (Haines et al., 2015).

On R428 treatment of MDAMB231 cells, while total Arf1 levels were unaffected (Fig. 4B), a significant decrease in Arf1 activation was observed (Fig. 4C). GGA3 pulldown controls also confirmed specificity, with Arf1-GTP being enriched in GST-GGA3 pulldowns more than GST-only beads (Fig. S4B). To further confirm the specificity of the GGA3 pull-down assay, siRNA-mediated Arf1 knockdown was done in MDA-MB-231 cells, revealing reduction in Arf1 levels in both whole-cell lysate and pulldown fractions (Fig. S4C).

To further test the relationship between the two proteins, AXL knockdown was performed, which led to a marked reduction in Arf1 activation (Fig. 4D). This also suggests that AXL levels and activation regulate Arf1 activation and, consequently, Golgi organization. R428 treatment, however, did not affect Arf GEF GBF1 levels (Fig. S4D), known to regulate Arf1 activation and localization at the Golgi (Rajeshwari et al., 2023). We further used ARHGAP10 Arf1 binding domain (ABD-GFP) to detect changes in active Arf1 localization in cells under different scenarios (Rajeshwari et al., 2023). Endogenous AXL colocalizes with ABD-GFP at the Golgi in adherent MDAMB231 cells, but this colocalization drops on R428 treatment (Fig. S4E). A corresponding loss of ABD-GFP from the Golgi (GM130) (Fig. 4E) likely reflects a drop in active Arf1 levels, consistent with our pulldown results (Fig. 4C), consequently causing Golgi disorganization. Together, they support the conclusion that Golgi-associated active Arf1 mediates AXL-dependent regulation of Golgi organization and function.

While AXL and Arf1 association could regulate this crosstalk, this remains to be fully characterized. Their functional association is evident, as expression of constitutively active Arf1 (Q71L) effectively rescues the R428-induced Golgi disorganization in these cells (Fig. 4F). Independently, inhibition of Arf1 activation using the GBF1 inhibitor Golgicide A (GCA) also disorganizes the Golgi in adherent MDAMB231 cells (Fig. 4G). Golgicide A (GCA)-mediated Arf1 inhibition in MDA-MB-231 cells did not affect total AXL levels (Fig. S4F), but caused an increase in AXL Y702 phosphorylation (Fig. S4G), accompanied by a loss of AXL localization from the Golgi (Fig. 4H). These suggests a possible reciprocal regulation between AXL and Arf1 that may contribute to the control of Golgi organization in these cells and warrants further investigation.

To understand how AXL could regulate Arf1 activation and localization to control Golgi organization, we examined the possible role of the Arf1 GEF, GBF1 in this pathway. GBF1 localization at the Golgi is known to regulate Arf1 activation (Kaczmarek et al., 2017; Walton et al., 2023). We observed R428 treatment to cause a concentration-dependent increase in AMPK activation (pThr172) in MDA-MB-231 cells (Fig. 4I). AMPK is reported to phosphorylate GBF1 and regulate its localization at the Golgi, which could in turn affect Golgi organization (Freemantle et al., 2024; Mao et al., 2013). We hence targeted AMPK activation using Compound C in R428-treated MDAMB231 cells and found it to significantly restore Golgi organization (Fig. 4J). This confirms that AMPK activation contributes to the R428-mediated Golgi disruption. As detecting GBF1 phosphorylation was limited by the availability of a reliable antibody, we instead examined GBF1 localization at the Golgi, which is regulated by its phosphorylation status (Freemantle et al.,

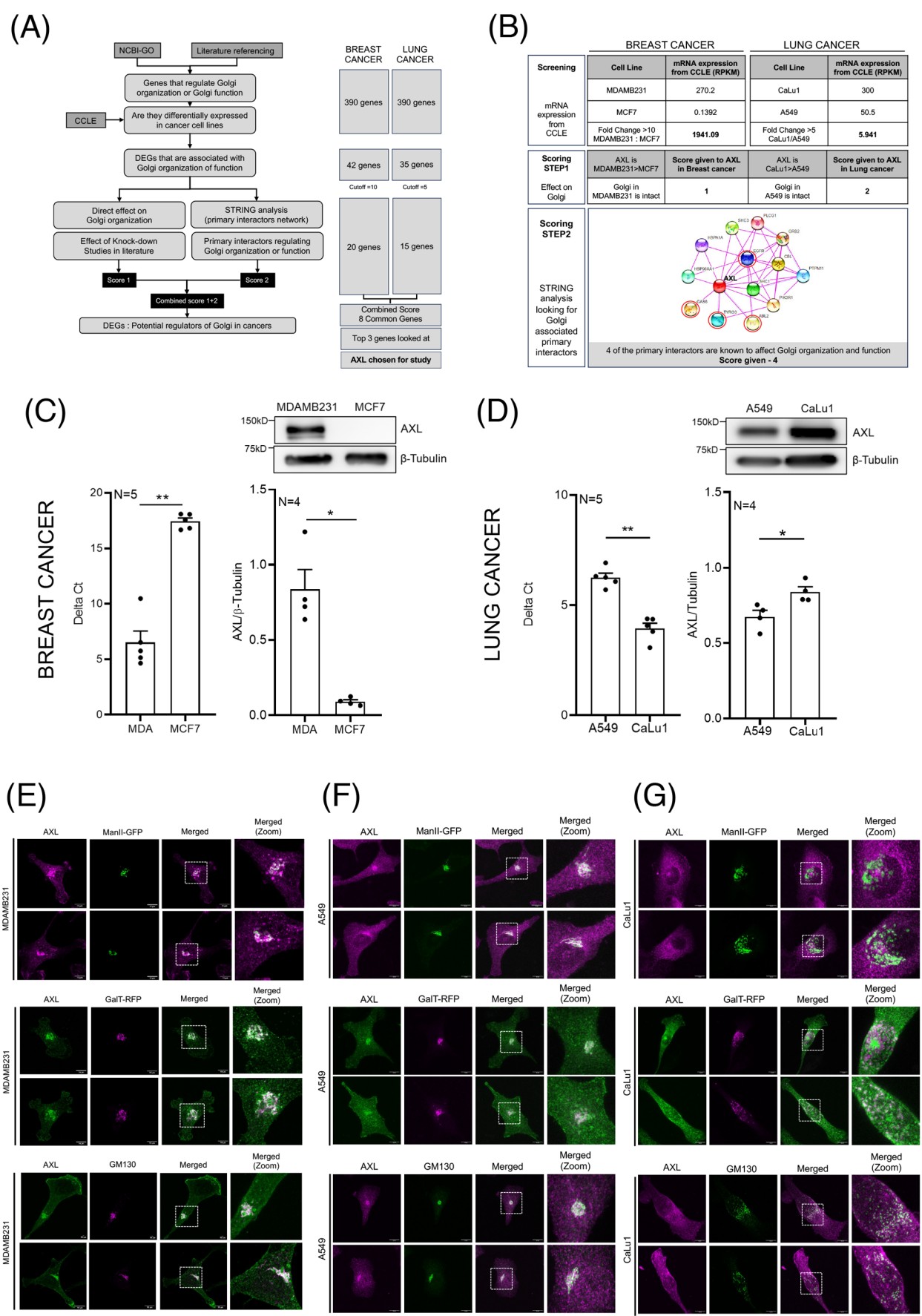

**Fig. 2.** See next page for legend.

**Fig. 2. Differentially expressed genes in breast and lung cancer cells.**
(A) Schematic representation shows the steps followed in the *in silico* analysis with the number of shortlisted genes at each stage. (B) Details of the *in silico* analysis for AXL in breast and lung cancer cell line pairs showing their screening based on mRNA expression in CCLE database (Screening), scores given based on their known effect on Golgi organization (Scoring STEP1) and based on number of primary interactors known to be associated with the Golgi using STRING analysis (Scoring STEP2). (C,D) Comparative mRNA and protein expression of AXL in (C) MDAMB231 and MCF7 and (D) A549 versus CaLu1 cells. Graph represents mean±s.e.m. of delta Ct values obtained from quantitative PCR analysis from five independent experiments (shown on left) and ratio of western blot densitometric band intensities for AXL and β-tubulin shown as mean±s.e.m. from four independent experiments. Representative western blots are shown above the graph. (E-G) Representative cross-section images and merged zoomed-insets for AXL localization with Golgi markers, GM130 (magenta in MDAMB231 and green in A549 and CaLu1 cells) or ManII-GFP (green) or GalTase-RFP (magenta) in (E) MDAMB231 and (F) A549 and (G) CaLu1 cells. The representative image for GM130 and AXL immunostaining in MDAMB231 (in E) is also used in Fig. 4A (bottom, SA-CNT) as these data came from the same experiment. Statistical analysis was done using Mann–Whitney U test. Scale bars: 10 μm. (*P≤0.05, **P≤0.01).

2024; Walton et al., 2023). Using GM130 as a Golgi marker, we observed that R428 treatment reduced the overlap between GBF1 and GM130, consistent with Golgi disorganization (Fig. 4J). Notably, this effect was significantly reversed upon Compound C treatment, concomitant with the restoration of Golgi organization (Fig. 4J). These changes were quantitatively reflected in the Pearson's coefficient analysis of colocalization between GBF1 and GM130 (Fig. 4J).

Together, these observations support the presence of an AXL-AMPK-GBF1-Arf1 signalling axis that could contribute to how AXL regulates Arf1 activation, localization at the Golgi, and thereby Golgi organization.

### AXL-Arf1 signalling axis on loss of adhesion mediated regulation of Golgi organization in MDAMB231 cells

Loss of adhesion also causes the Golgi to disorganize in MDAMB231 as is reported for 'normal' mouse fibroblasts and seen to be dependent on Arf1 activation (Singh et al., 2018). This leads us to ask if AXL-Arf1 signalling axis could also be regulated by adhesion to affect Golgi organization, and the role AMPK-GBF1 could have in mediating the same. Loss of adhesion caused a significant increase in AXL phosphorylation at Y702 (Fig. 5A), a response that mirrors the effect R428 treatment has in adherent MDA-MB-231 cells. This increase in phospho-AXL (Y702) was evident at both early (10 min) and later (120 min) time points of suspension (Fig. S5A). This is accompanied by a significant reduction in Arf1 activation (Fig. 5B) and pAkt levels (Fig. 5C; Fig. S5B). R428 treatment reduces Akt activation by ~50% (Fig. 3C), which decreases further upon loss of adhesion (~70%) (Fig. 5C), suggesting that Akt regulation in these cells may come from both AXL-dependent and adhesion-dependent inputs. Moreover, the increase in AXL phosphorylation (Y702) in suspended cells indicates that adhesion could influence AXL activity, which may in turn contribute to the regulation of Akt and Arf1 signalling.

In addition, like the increase in AMPK activation observed upon R428 treatment, suspended MDA-MB-231 cells show a significant increase in AMPK activation compared to stable adherent cells (Fig. 5D). This further supports a possible role for the AXL-AMPK-(GBF1)-Arf1 axis in loss of adhesion mediated Golgi disorganization in MDA-MB-231 cells. Recent studies from the laboratory show that pharmacological inhibition of Akt does not affect Golgi organization (Saha et al, 2026), suggesting that the AXL-mediated regulation of AMPK could be independent of Akt.

Furthermore, closely mirroring the effects observed upon R428 treatment, loss of adhesion led to the displacement of both AXL (Fig. 5E) and active Arf1 (ABD-GFP) (Fig. 5F) from the Golgi in MDA-MB-231 cells as the Golgi became disorganized. Consistent with this, the relative spatial overlap between AXL and active Arf1 was also lost upon loss of adhesion (Fig. S5C). Total GBF1 levels, however, remained unchanged upon loss of adhesion (Fig. S5D). Consistent with the functional relationship between AXL and Arf1, expression of constitutively active Arf1 (Q71 L) rescued the loss-of-adhesion-mediated Golgi disorganization in MDA-MB-231 cells (Fig. 5G). Additionally, loss of adhesion-induced Golgi disorganization was accompanied by a reduction in tubulin acetylation (Fig. 5H), like the effects observed upon R428 treatment and AXL knockdown (Fig. 3H,I). Together, these findings indicate that loss of adhesion phenocopies the effects of R428 treatment or AXL depletion, underscoring the AXL-Arf1 signalling axis as a key physiological regulator of Golgi organization.

### AXL regulates adhesion-dependent Golgi organization in lung cancer cells

Lung cancer A549 and CaLu1 cells have a Golgi phenotype that differs from non-transformed lung epithelial BEAS2B cells (Fig. S1A). Adherent CaLu1 cells have a disorganized Golgi (Fig. 1A) and A549 cells an organized Golgi, retained on loss of adhesion (Fig. 1D). R428 treatment causes a concentration-dependent drop in phospho-Akt levels (Fig. 6C,D) and an increase in phospho-AXL (Y702) levels (Fig. 6E,F) in both cell lines. In A549 cells, R428 treatment also causes the Golgi to disorganize (Fig. 6A) although does not affect the disorganized Golgi in CaLu1 cells (Fig. 6B). Interestingly, in suspended A549 cells lower R428 concentrations (1 μM) also caused significant disorganization of cis- (GM130), cis-medial (ManII-GFP), and trans-Golgi (GalTase-RFP) (Fig. 6G). siRNA-mediated knockdown of AXL in A549 cells also caused the Golgi to be prominently disorganized in stable adherent and suspended A549 cells (Fig. 6H). This suggests that AXL controls Golgi organization in A549 cells but its regulation by adhesion is different (Fig. 6G). Both R428 treatment and AXL knockdown affect Akt activation in A549 cells. (Fig. 6I). We, hence, tested AXL localization at the Golgi (GM130) in non-adherent A549 cells and found it to be reduced significantly on R428-mediated Golgi disorganization (Fig. 6J). This suggests that changes in AXL activation, localization and, hence, their crosstalk with Golgi regulators could be differentially affected on loss of adhesion.

### AXL-Arf1 axis regulates Golgi organization and function in non-adherent A549 cells

In A549 cells on loss of adhesion, active Arf1 levels (GGA3 pulldown) drop significantly (Fig. 7A), though this does not support Golgi disorganization. On R428 treatment of these cells, no effect on Arf1 activation in pulldowns was seen (Fig. 7B). Loss of adhesion did not affect AXL phosphorylation (Y702) (Fig. 7C), but on treatment with R428, it expectedly increases significantly (Fig. 7D), accompanied by a distinct drop in Akt activation (Fig. 7E). Despite a drop in active Arf11 levels on loss of adhesion, residual active Arf1 is seen localized at the Golgi (Fig. 7F) and could support its organization in non-adherent A549 cells. R428 treatment causes active Arf1, detected using ABD-RFP, to be displaced from the Golgi driving its disorganization in non-adherent A549 cells (Fig. 7G; Fig. S6A).

Expression of constitutively active Arf1 (Q71L), localized to the Golgi, prevents R428 mediated Golgi dispersal on loss of adhesion in A549 cells (Fig. 7G), as seen in MDAMB231 cells (Fig. 5G). This suggests a loss of AXL accompanied by a significant loss of

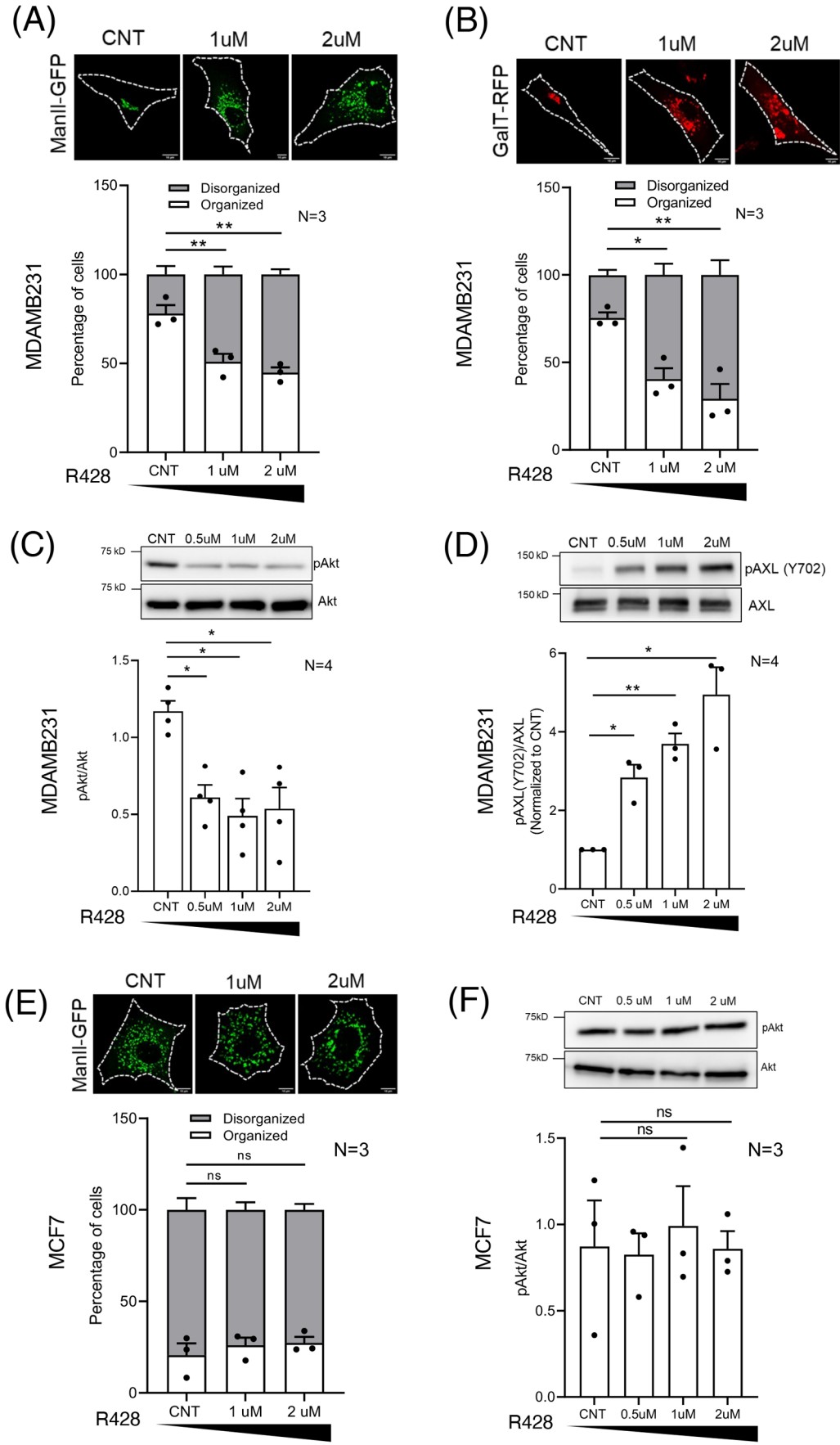

**Fig. 3.** See next page for legend.

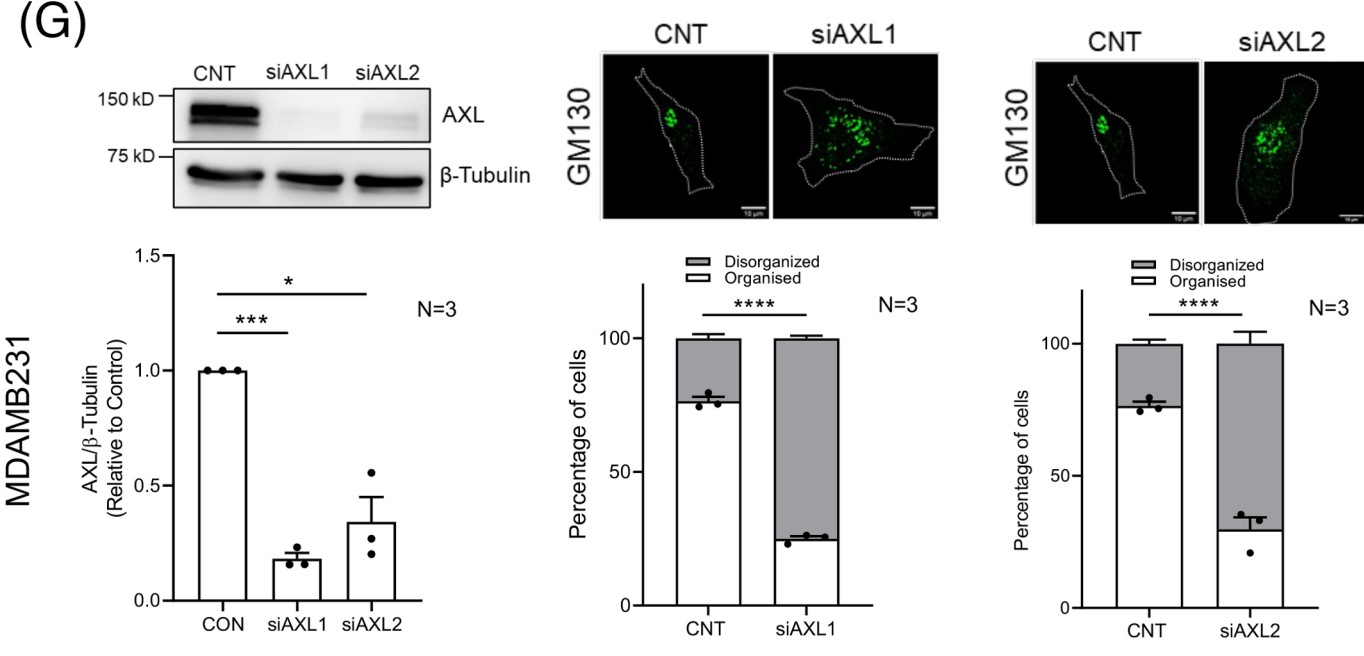

Fig. 3. See next page for legend.

**Fig. 3. AXL-mediated regulation of Golgi organization in breast cancer cells.** (A,B) Representative cross-section images of the predominant Golgi phenotype in MDAMB231 cells expressing (A) ManII-GFP (green) or (B) GalTase-RFP (red), treated with DMSO (CNT) or R428 (1 µM or 2 µM). Percentage distribution profile for cells ($n{\geq}150$) shows organized (white) and disorganized (grey) Golgi in adherent cells across treatments. The graphs represent mean±s.e.m. of percentage distribution from three independent experiments. (C,D) Representative western blots for (C) phosphorylated Akt (pAkt) and total Akt (Akt), and (D) Y702 phosphorylated AXL (pAXL) and total AXL (AXL), in cell lysates from DMSO-treated (CNT), and 0.5 µM, 1 µM and 2 µM R428-treated MDAMB231 cells. Graph represents ratio of densitometric band intensities (normalized to CNT for D) as mean±s.e.m. from four independent experiments. (E) Percentage distribution profile for MCF7 cells ($n{\geq}150$) expressing ManII-GFP, with organized (white) and disorganized (grey) Golgi, in DMSO-treated (CNT), and 1 µM and 2 µM R428-treated cells. Representative cross-sectional confocal images of the predominant Golgi phenotype shown. The graph represents mean±s.e.m. of percentage distribution from three independent experiments. (F) Representative western blots for phosphorylated Akt (pAkt) and total Akt (Akt) in cell lysates from DMSO-treated (CNT), and 0.5 µM, 1 µM and 2 µM R428-treated MCF7 cells. Graph represents ratio of densitometric band intensities as mean±s.e.m. from three independent experiments. (G) Representative western blots for AXL in cell lysates from control (CNT), siAXL1- and siAXL2-treated MDAMB231 cells. Graph represents ratio of densitometric band intensities (normalized to CNT) as mean±s.e.m. from three independent experiments. Representative cross-sectional images of the predominant Golgi phenotype in GM130 (green) immunostained in CNT, siAXL1- and siAXL2-treated MDAMB231 cells. The GM130 CNT images are the same, as they are from a common control for both siRNA treatments. Percentage distribution profile of cells ($n{\geq}150$) showing organized (white) and disorganized (grey) Golgi in all treatments shown. The graphs represent mean±s.e.m. of percentage distribution from three independent experiments. (H,I) Representative western blots for acetylated tubulin and β-tubulin in cell lysates from (H) DMSO-treated (CNT), and 0.5 µM, 1 µM R428-treated and (I) CNT, siAXL1- and siAXL2-treated MDAMB231 cells. Graphs represent ratio of densitometric band intensities (normalized to CNT) as mean±s.e.m. from four and three independent experiments. The black bar below graphs represents the gradient of increasing concentration of R428 treatment. Statistical analysis was done using one-way ANOVA multiple comparisons test with Tukey's method for error correction, for the distribution profiles, Mann–Whitney $U$ test for non-normalized and single sample $t$-test for normalized western blotting results. Scale bars: 10 µm. (*$P{\leq}0.05$, **$P{\leq}0.01$, ***$P{\leq}0.001$, ****$P{\leq}0.0001$; ns=not significant).

Arf1 from the Golgi is needed for its disorganization. Interestingly, treatment of suspended A549 with GBF1 inhibitor Golgicide-A (GCA) causes the Golgi to disperse with loss of AXL localization (Fig. 7H). This is likely mediated by a loss of active Arf1 from the Golgi on GCA treatment.

We see a drop in tubulin acetylation on loss of adhesion in A549 cells (Fig. 7I), which is accompanied by a drop in Arf1 activation (Fig. 7A). This is further enhanced on Golgi disorganization by R428 treatment (Fig. 7I). This suggests that, in non-adherent A549 cells, both Arf1 activation (Zhang et al., 2023) and Golgi organization contribute to regulate tubulin acetylation.

Cell surface glycosylation changes, detected by lectin labelling, are a sensitive measure of changes in Golgi function (Bekier et al., 2017; Zhang and Wang, 2016). Quantitative flow cytometric measurements of lectin binding (ConA, WGA, PNA) in non-adherent A549 cells show no significant change (Fig. 7J). R428 treatment of these cells causes a modest but significant change in cell surface bound ConA (binding mannose) and WGA [binding sialic acid and N-acetylglucosamine (N-GlcNAc) glycans] lectin levels. No change in PNA (binding N-GalNAc) lectin levels were seen (Fig. 7J). This suggests that AXL inhibition-mediated Golgi disorganization affects Golgi function in A549 cells. As reported earlier (Sun et al., 2022), R428 treatment of A549 cells significantly reduces their

anchorage-independent growth (Fig. 7K), which may be associated with accompanying changes in Golgi organization and dependent cell-surface glycosylation.

Together, our studies in breast and lung cancer cells (MDAMB231 and A549) identify the AXL-Arf1 signalling axis as a critical determinant of Golgi organization and function. This is mediated through AXL-dependent regulation of Arf1, which could be mediated by an AMPK-GBF1 pathway that controls its recruitment and activation of Arf1 at the Golgi. Importantly, the adhesion-dependent regulation of both AXL and Arf1 underscores their coordinated role in maintaining Golgi integrity and function.

## DISCUSSION

AXL is frequently overexpressed in multiple malignancies, including breast cancer, NSCLC, and acute myeloid leukaemia, and its elevated levels often correlated with poor prognosis and reduced survival rates (Auyez et al., 2021; Zhu et al., 2019). Changes in the organization and function of the Golgi, including its regulation of cell-surface glycosylation, can enhance cancer cell metastasis (Rambaruth and Dwek, 2011), immune evasion (Demetriou et al., 2001), survival (Petrosyan et al., 2014; Piyush et al., 2017) and drug resistance (Britain et al., 2018; Lopez Sambrooks et al., 2018; Ohashi et al., 2018), making the Golgi an attractive target in cancer therapy. (Bhat et al., 2017; Ohashi et al., 2018; Petrosyan et al., 2014). While Golgi fragmentation has been speculated to support cancer phenotypes (Makhoul et al, 2019; Petrosyan, 2015), it remains unclear whether differential Golgi organization contributes to cancer progression or arises because of it. The differential regulation of Golgi organization in breast (MDA-MB-231 versus MCF7) and lung (A549 versus Calu1) cancer cell lines, together with the identification of DEGs that may govern this process, provides important insight into Golgi regulation. Notably, the identification of AXL as a DEG of interest highlights its potential relevance in modulating Golgi organization and function in these cancers.

Seen to prominently localize at the Golgi, targeting of AXL protein using siRNA knockdown and R428 treatment are seen to regulate Golgi organization in breast cancer and lung cancer cells. Both regulate Akt activation downstream of AXL, missing in AXL-lacking MCF7 cells, confirming R428 specificity. While AXL was suggested to localize at the Golgi (Zajac et al., 2020), our study confirms its distinct overlap with cis-Golgi, cis-medial-Golgi, and trans-Golgi compartments.

This Golgi localization of AXL appears important for maintaining Golgi organization through its regulation of Arf1 activation and localization. Previous studies have shown receptor tyrosine kinases to influence Arf1 activity, Gefitinib enhances Arf1 recruitment to EGFR, increasing Arf1 activity at the PM (Haines et al., 2015). Our findings suggest that AXL regulates Arf1 in the context of the Golgi. Targeting AXL using R428 or AXL knockdown reduces Arf1 activation and disrupts Golgi organization. Importantly, this phenotype is rescued by expression of constitutively active Arf1 (Q71L) and inhibition of AMPK activation, supporting a functional relationship between AXL, Arf1 and AMPK in maintaining Golgi structure. This could be mediated through an AXL-AMPK-GBF1 axis that influences Arf1 activation and localization at the Golgi. Consistent with this, inhibiting GBF1 using Golgicide A, which is known to affect Arf1 activation (Sáenz et al., 2009), also alters AXL phosphorylation (Y702) and its localization, indicating that the crosstalk between these proteins may be reciprocal, although this will require further investigation. Together, these observations support a model in which AXL affects Arf1 localization and activity at the Golgi, with AMPK-GBF1 mediated regulation of Arf1 activation providing a possible mechanistic link.

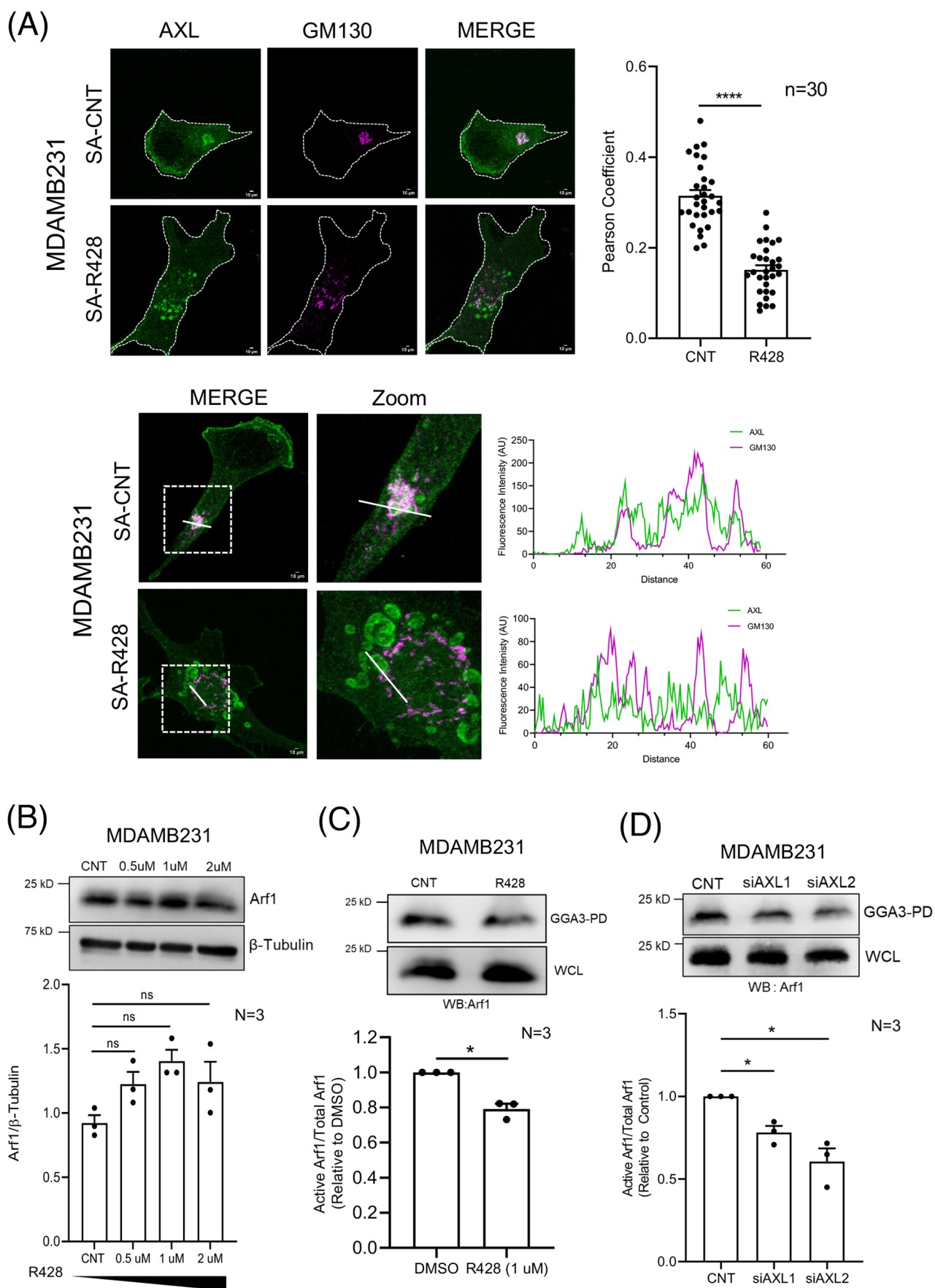

**Fig. 4.** See next page for legend.

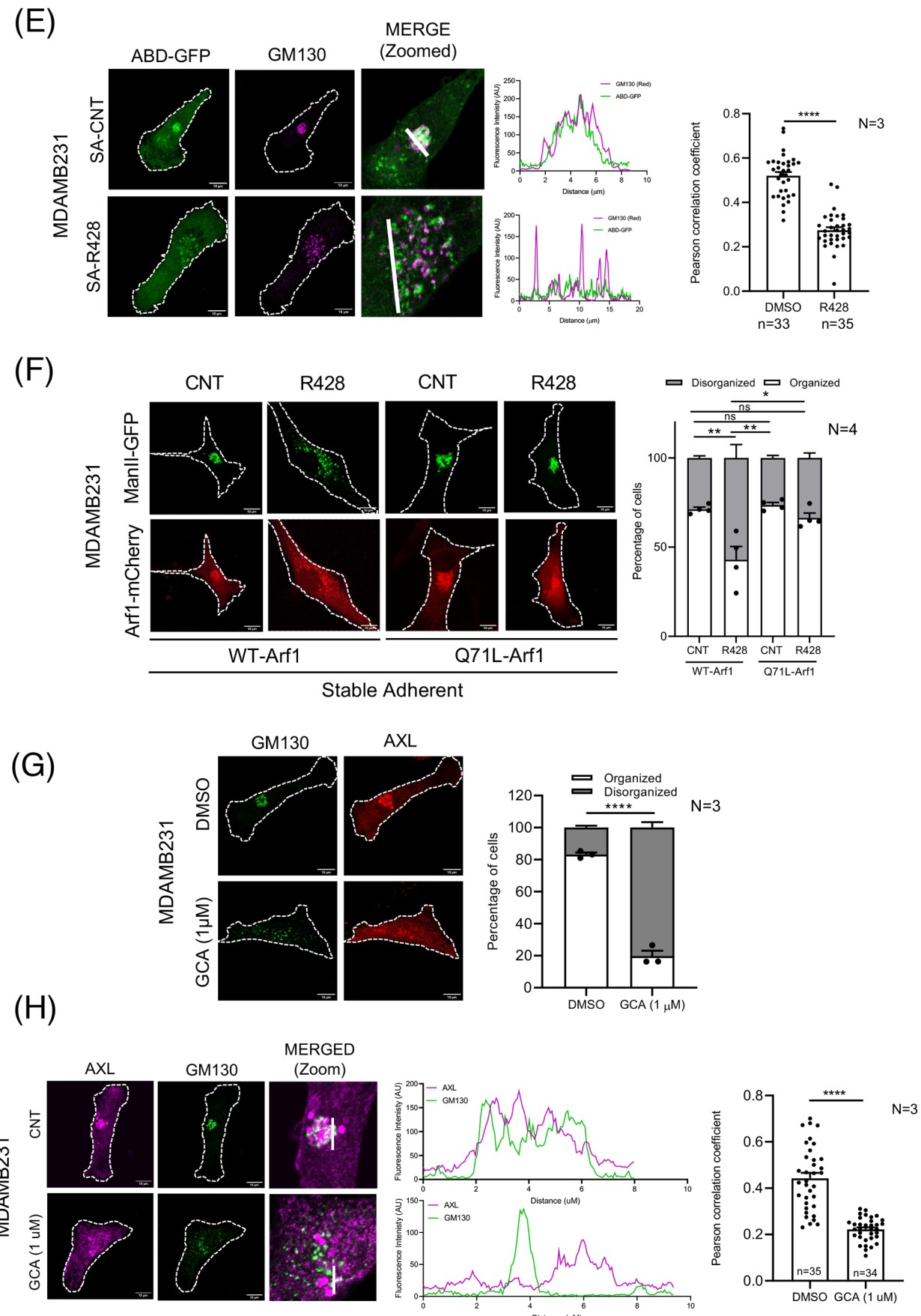

**Fig. 4.** See next page for legend.

(I)

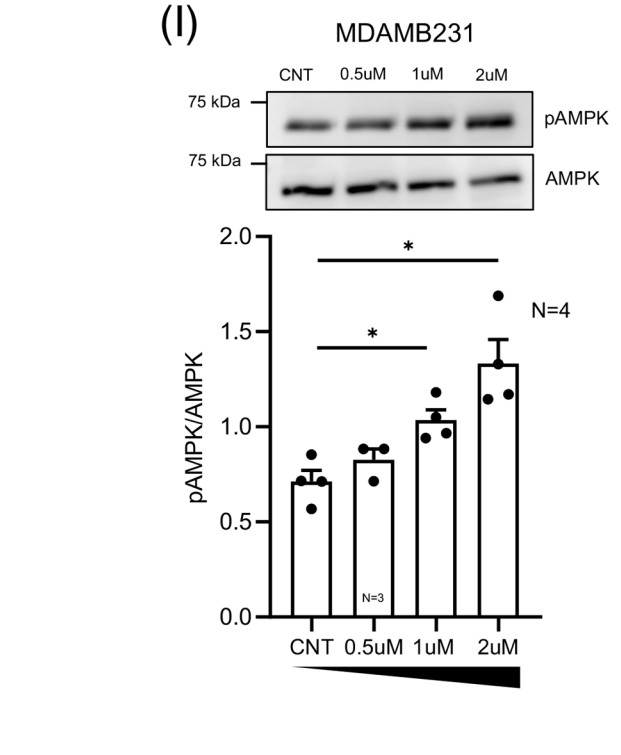

(J)

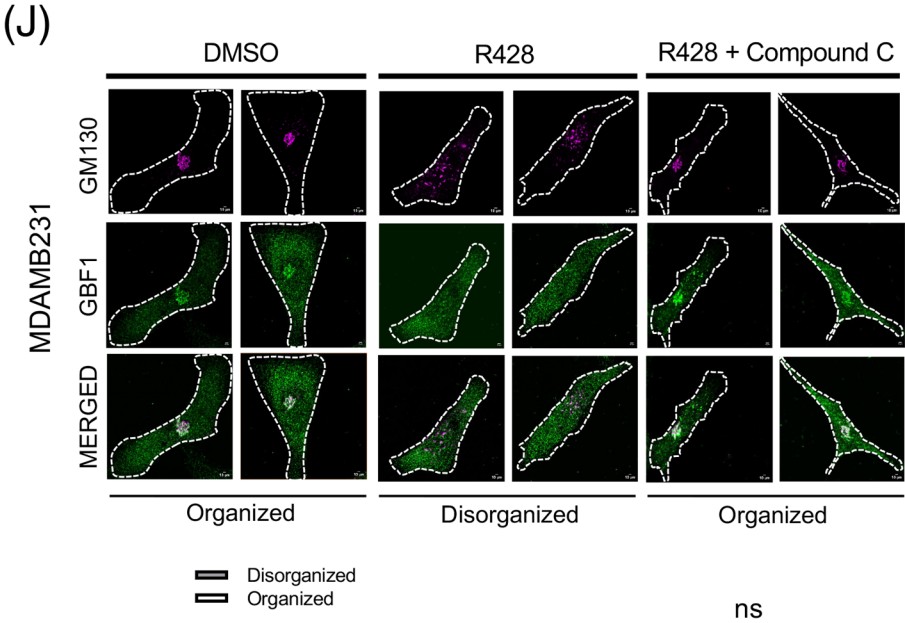

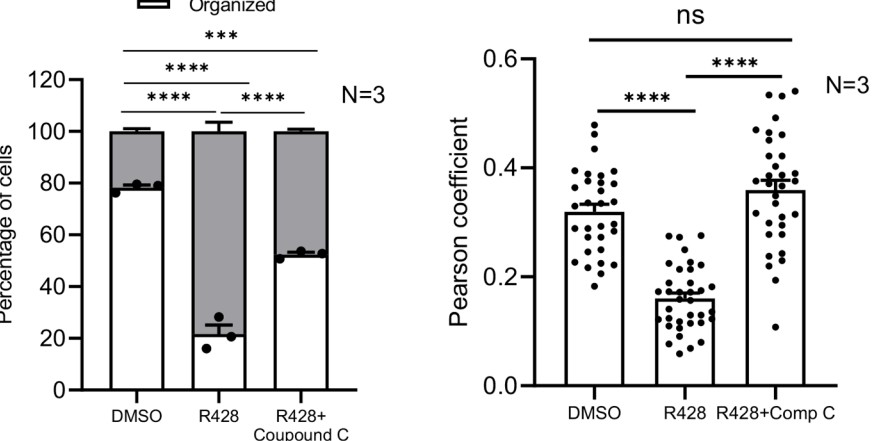

**Fig. 4.** See next page for legend.

**Fig. 4. AXL-Arf1 crosstalk regulates Golgi organization in adherent MDAMB231 cells.** (A) Top panel: representative deconvoluted Z-stack maximum intensity projection (MIP) images for AXL (green) and GM130 (magenta) immunostained DMSO-treated (SA-CNT) or R428-treated (SA-R428) MDAMB231 cells. The graph shows the mean±s.e.m. of Pearson's coefficient for colocalization of AXL and GM130 ($n$=30 cells) from three independent experiments. Lower panel: representative cross-section and corresponding zoomed-in image (marked box). Line plot analysis shows intensity profile for AXL (green) and GM130 (magenta) along a solid line in merged image treated with DMSO (SA-CNT) and R428 (SA-R428). The representative image for GM130 and AXL immunostaining in MDAMB231 is the same as in Fig. 2E, as these data are from the same experiment. AU, arbitrary units. (B) Representative western blots for Arf1 and β-tubulin in cell lysates from DMSO-treated (CNT), and 0.5 μM, 1 μM and 2 μM R428-treated MDAMB231 cells. Graph represents ratio of densitometric band intensities as mean±s.e.m. from three independent experiments. The black bar below the graph represents the gradient of increasing concentration of R428 treatment. (C) Representative western blots for Arf1 (WB:Arf1) in pulldown using GST-GGA3 and in whole-cell lysate (WCL) of DMSO-treated (CNT) and R428 (1 μM)-treated MDAMB231 cells. Graph represents ratio of densitometric band intensities (normalized to CNT) in GGA3-PD to WCL as mean±s.e.m. from three independent experiments. (D) Representative western blots for Arf1 (WB: Arf1) in pull down using GST-GGA3 (active Arf1) and in whole-cell lysate (WCL) of control (CNT) and AXL knockdown (using siAXL1 and siAXL2) MDAMB231 cells. Graph represents ratio of densitometric band intensities (normalized to CNT) in GGA3-PD to WCL as mean±s.e.m. from three independent experiments. (E) Representative cross-section images of cells expressing ABD-GFP (green) and the Golgi immunostained with GM130 (magenta) in adherent MDAMB231 cells, treated with DMSO (SA-CNT) or R428 (SA-R428). Line plots for ABD-GFP and GM130 shown next to their respective images. Graph represents the Pearson's correlation coefficients for ABD-GFP (green) and GM130 (magenta) colocalization plotted as mean±s.e.m. for $n{\geq}30$ cells from three experiments. (F) Representative cross-section images of stable adherent MDAMB231 cells expressing Q71L-Arf1-mCherry (red) or WT-Arf1-mCherry (red) and ManII-GFP (green) in DMSO-treated (CNT) and R428-treated cells. Graph shows percentage distribution profile for control and R428-treated WT-Arf1- and Q71L-Arf1-expressing cells ($n{\geq}100$) with organized (white) or disorganized (grey) Golgi. The graph represents mean±s.e.m. of percentage distribution from four independent experiments. (G) Percentage distribution profile of cells ($n{\geq}200$ cells) of adherent MDAMB231 cells treated with DMSO (CNT) or GCA (1 μM) showing organized (white) and disorganized (grey). Representative cross-section confocal images of cells immunostained with GM130 (green) and AXL (magenta) showing predominant Golgi organization phenotype for each condition. Graphs represent mean±s.e.m. from three independent experiments. (H) Representative cross-sectional images with merged zoom insets and line plots for colocalization of AXL (magenta) and GM130 (green) immunostained adherent MDAMB231 cells, treated with DMSO (CNT) and 1 μM GCA for 30 mins. Graph represents the Pearson's coefficients for AXL (red) and GM130 (green) colocalization plotted as mean±s.e.m. for $n{\geq}30$ cells for three experiments. (I) Representative western blots for phosphorylated AMPK (pAMPK) and total AMPK (AMPK) in cell lysates from DMSO-treated (CNT), and 0.5 μM, 1 μM, and 2 μM R428-treated MDA-MB-231 cells. Graph represents the ratio of densitometric band intensities as mean±s.e.m. from three or four independent experiments. (J) Representative cross-section images of the predominant Golgi phenotype in MDA-MB-231 cells immunostained for GBF1 (green) and GM130 (magenta) treated with DMSO (CNT), R428 (1 μM), and R428+Compound C (5 μM). Percentage distribution profile of cells ($n{\geq}200$) showing organized (white) and disorganized (grey) Golgi in adherent MDA-MB-231 cells under these conditions. Bar graph (with data points) represents Pearson's coefficients of colocalization for GBF1 (green) and GM130 (magenta), plotted as mean±s.e.m. for $n{\geq}30$ cells from three independent experiments. Statistical analysis was done using one-way ANOVA for Pearson's colocalization analysis and distribution profile. Mann–Whitney $U$ test was used for non-normalized and single sample $t$-test for normalized western blotting results. Scale bars: 10 μm. (*$P{\leq}0.05$, **$P{\leq}0.01$, ***$P{\leq}0.001$, ****$P{\leq}0.0001$; ns=not significant).

In MDAMB231 on loss of adhesion, a drop in Arf1 activation accompanied by an increase in AXL phosphorylation and their displacement from the Golgi further suggests this crosstalk to have a

role in physiological circumstances. We hence report, for the first time, cell-matrix adhesion to regulate AXL activation and localization at the Golgi. AXL knockdown, R428 treatment, and loss of adhesion all lead to Golgi disorganization, which extends to Golgi-associated functions such as reduced tubulin acetylation, thereby impacting microtubule stability (Eshun-Wilson et al., 2019). Microtubule acetylation is affected by Arf1 activation (Zhang et al., 2023) and Golgi organization (Brodsky et al., 2022; Sanders and Kaverina, 2015), making it a vital functional outcome of AXL-dependent regulation of the Golgi in cancers. Golgi-associated MTs have also been implicated as 'fast tracks' for anterograde trafficking (Hao et al., 2020) and regulators of directed cell migration (Wu et al., 2016), both having implications for cancers.

Knowing that AXL-lacking MCF7 cells have a dispersed Golgi, elucidating the effect restoring AXL expression has on the Golgi in these cells could establish the role its differential expression could have in cancers. Stable AXL expression in MCF7 restores Golgi organization and elevates Arf1 activity (in comparison to control MCF7 cells), supporting its role in regulating Arf1 to drive Golgi organization (Saha et al., 2026).

In MDAMB231 and A549 cells, AXL localizes at the Golgi, and its targeting (using R428 and knockdown) is seen to disrupt the Golgi in both. This is accompanied by AXL being lost from the Golgi, with a drop in Arf1 activation. Golgi organization is restored by Q71L Arf1 in both. AXL inhibitor (R428) treatment also affects Golgi function (MDAMD231-tubulin acetylation/A549-tubulin acetylation and glycosylation) in both cell types, confirming the presence of an AXL-Arf1 functional crosstalk that regulates Golgi organization and function.

Differences in the AXL-mediated regulation of the Golgi in MDAMB231 and A549 cells are also insightful. Organized Golgi in adherent MDAMB231 cells is dispersed on loss of adhesion but stays intact in A549 cells. On R428 of A549 cells, pronounced disorganization of the Golgi was seen only on loss of adhesion, and not in stable adherence. This could be mediated by the fact that R428 treatment inhibits Arf1 activation only in stable adherent MDAMB231 cells, not A549 cells. Loss of adhesion-mediated regulation of AXL [loss of localization at the Golgi and an increase pAXL (Y702) levels] is also seen in MDAMB231 cells, but not in A549 cells. In suspended A549 cells, both AXL and active Arf1 are retained at the Golgi. This could be the reason for the Golgi staying organized in A549 cells on loss of adhesion.

The absence of detectable displacement of active Arf1 (ABD-RFP) from the Golgi in suspended A549 cells may reflect either (1) partial retention of Golgi-localized active Arf1 despite an overall drop in Arf1 activity, or (2) limited sensitivity of the ABD overexpression system, which may only reveal large changes at the Golgi. Notably, when ABD-GFP/RFP-labelled active Arf1 remains at the Golgi, Golgi organization is preserved in both MDAMB231 and A549 cells. In contrast, R428 treatment causes loss of AXL from the Golgi (with increased pAXL Y702) and loss of active Arf1, leading to Golgi disorganization. Together, these findings suggest that the adhesion-AXL-Golgi axis operates canonically in MDAMB231 cells but may be regulated differently in A549 cells.

The possible role R428-mediated inhibition of Akt activation could have in the regulation of Golgi organization is challenged by the fact that direct Akt inhibition does not affect Golgi organization and function (Saha et al., 2026).

Changes in cell surface glycosylation levels, tubulin acetylation, and anchorage-independent growth all suggest that AXL targeting and resulting disruption of the Golgi could have function

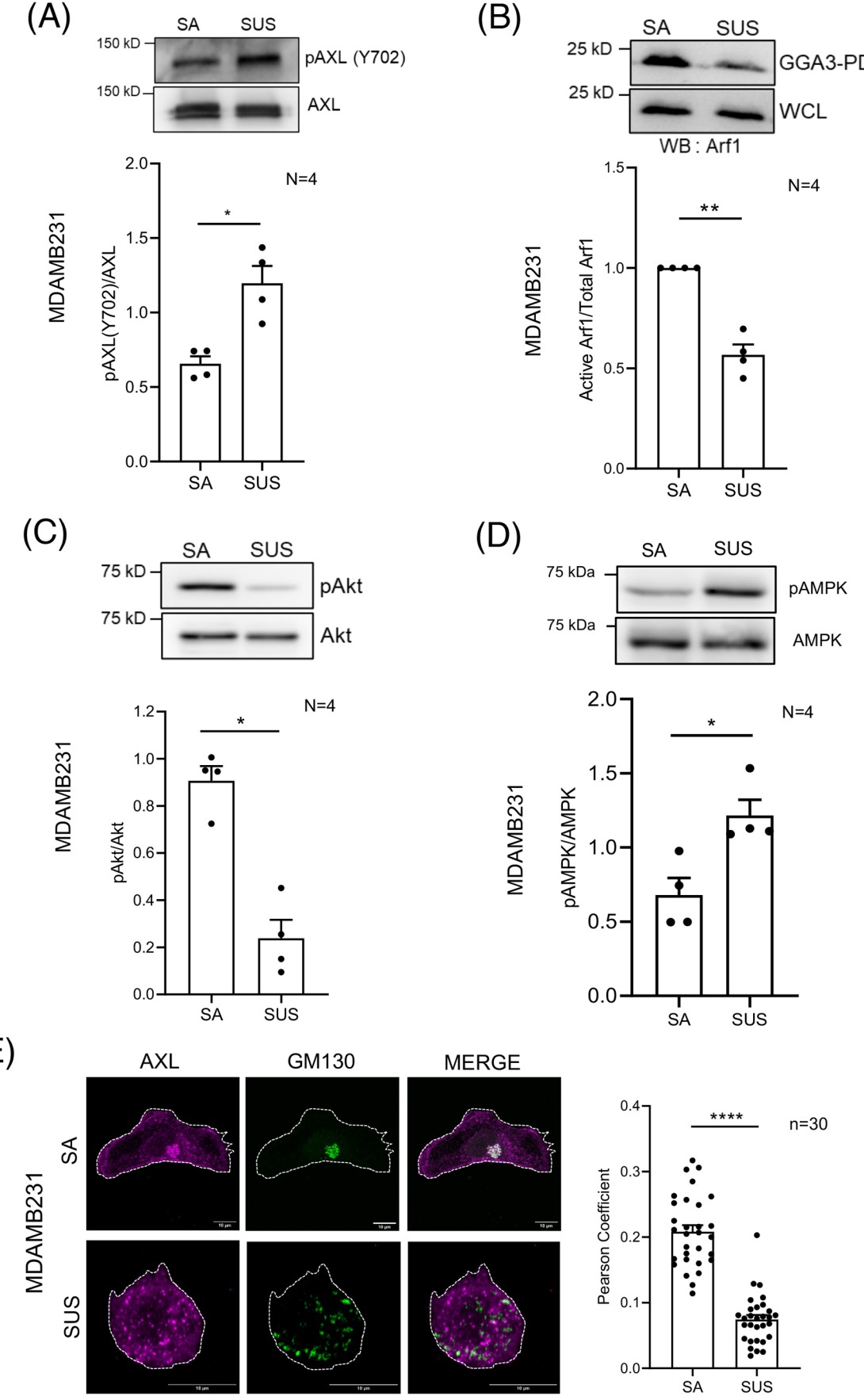

**Fig. 5.** See next page for legend.

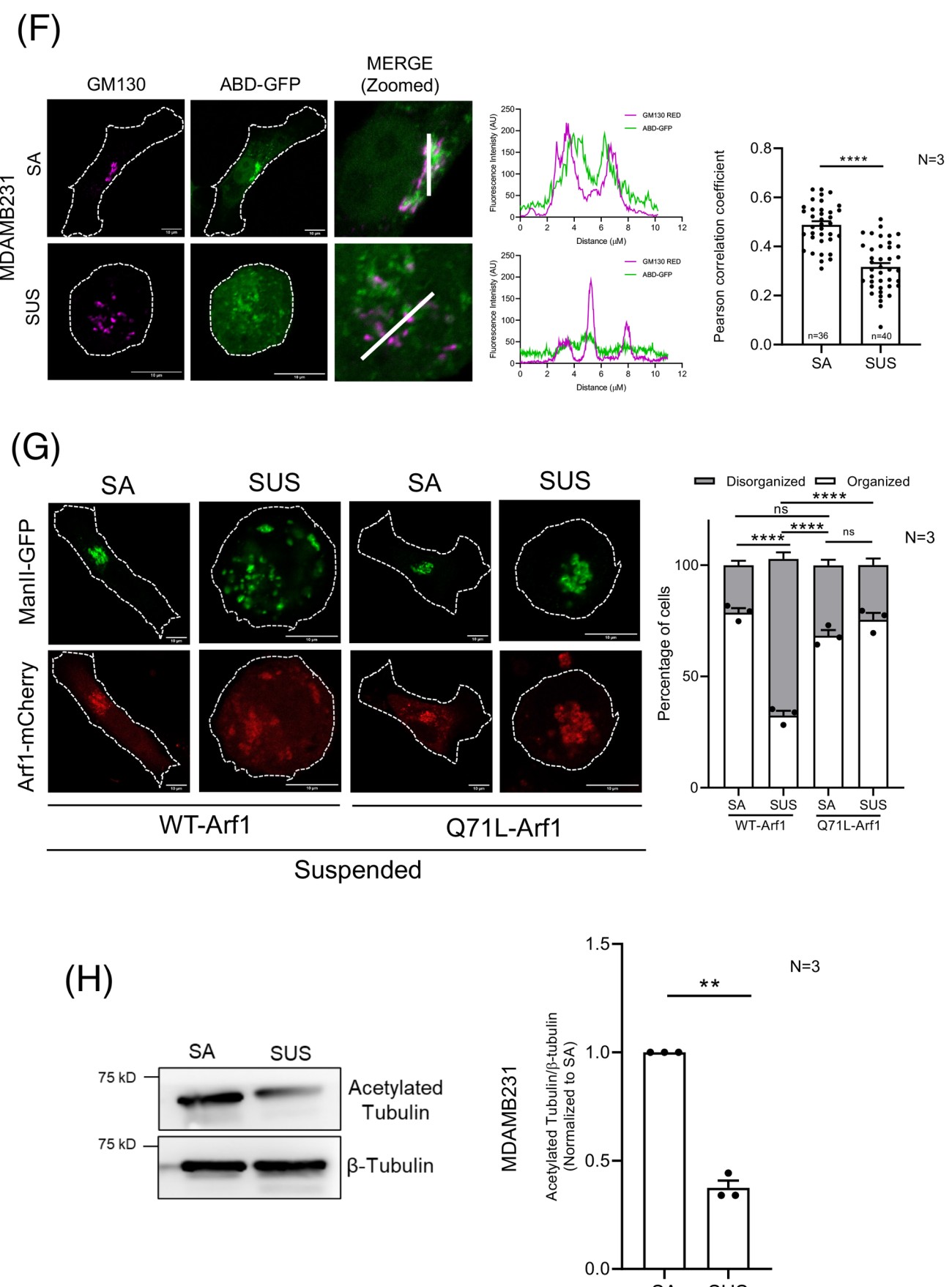

**Fig. 5.** See next page for legend.

(I)

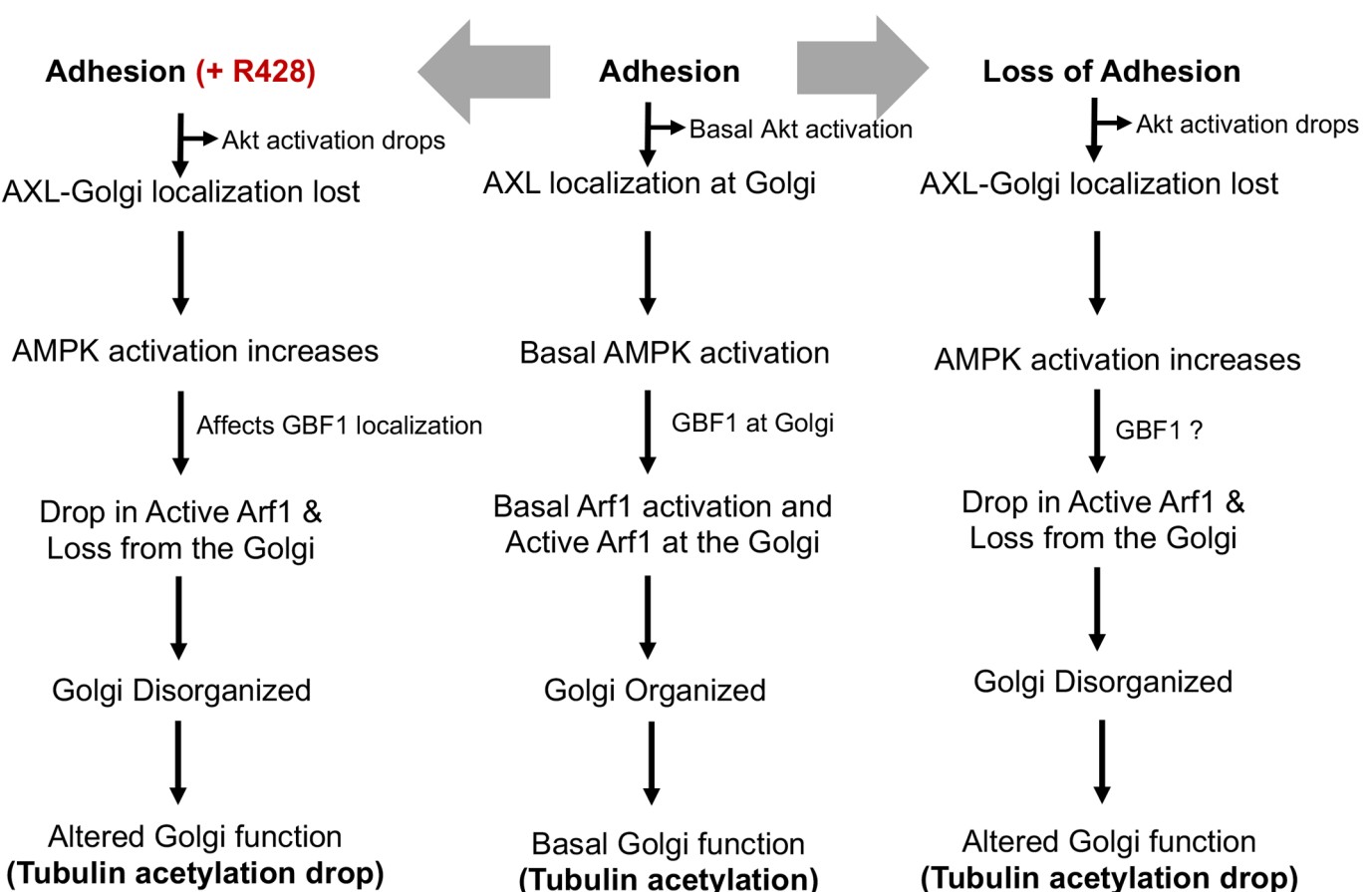

Fig. 5. Role of AXL and Arf1 in loss of adhesion-mediated Golgi disorganization in MDAMB231 cells. (A) Representative western blots for Y702 phosphorylated AXL (pAXL) and AXL in cell lysates of stable adherent (SA) and non-adherent (SUS) MDAMB231 cells. Graph represents ratio of densitometric band intensities as mean±s.e.m. from four independent experiments. (B) Representative western blots for Arf1 (WB:Arf1) in pull down using GST-GGA3 (GGA3-PD) and whole-cell lysate (WCL) of stable adherent (SA) and non-adherent (SUS) MDAMB231 cells. Graph represents ratio of densitometric band intensities (normalized to SA) in GGA3-PD to WCL as mean±s.e.m. from four independent experiments. (C,D) Representative western blots for (C) phosphorylated Akt (pAkt) and Akt, and (D) Phospho-AMPKα and AMPKα, in cell lysates of stable adherent (SA) and non-adherent (SUS) MDAMB231 cells. Graph represents ratio of densitometric band intensities as mean±s.e.m. from four independent experiments. (E) Representative deconvoluted images show immunostained AXL (magenta) and GM130 (green), in stable adherent (SA) and non-adherent (SUS) MDAMB231 cells. The graph shows the mean±s.e.m. of Pearson's coefficient for colocalization of AXL and GM130 (n=30 cells) from three independent experiments. (F) Representative cross-section images with merged zoom insets of MDAMB231 cells expressing ABD-GFP (green) and immunostained with GM130 (magenta) in stable adherent (SA) and non-adherent (SUS) conditions. Line plots to measure intensity for ABD-GFP and GM130 are shown next to their respective images. Graph represents the Pearson's coefficients for ABD-GFP (green) and GM130 (magenta) for colocalization plotted as mean±s.e.m. for n≥35 cells from three experiments. (G) Representative images for stable adherent (SA) and non-adherent (SUS) MDAMB231 cells expressing Q71L-Arf1-mCherry (red) or WT-Arf1-mCherry (red) and ManII-GFP (green). Graph represents the percentage distribution profile of WT-Arf1- and Q71L-Arf1-expressing cells (n≥150 cells) with organized (white) or disorganized (grey) Golgi. The graph represents mean±s.e.m. from three independent experiments. (H) Representative western blots for acetylated tubulin and total-tubulin in stable adherent (SA) and non-adherent (SUS) MDAMB231 cell lysates. Graph represents ratio of densitometric band intensities (normalized to SA) as mean±s.e.m. from three independent experiments. (I) Schematic illustrating the adhesion-AXL-AMPK-GBF1-Arf1 axis in regulating Golgi organization and function in MDA-MB-231 cells, highlighting the comparable effects of adhesion loss and R428 treatment on this pathway. Statistical analysis was done using one-way ANOVA multiple comparisons test with Tukey's method for error correction, for Pearson's colocalization analysis and distribution profile. Mann–Whitney U test was used for non-normalized and single sample t-test for normalized western blotting results. Scale bars: 10 µm. (*P≤0.05, **P≤0.01, ****P≤0.0001; ns=not significant).

implications for cancer progression. This could be evaluated by specifically targeting AXL, localized at the Golgi. Changes in the A549 glycocalyx could regulate viral infectivity in lung epithelial cells (Lai et al., 2025), which adds to the role AXL-Arf1-Golgi pathway could have in disease. AXL (Auyez et al., 2021; Debruyne et al., 2016; Hong et al., 2008; Okura et al., 2020; Scaltriti et al., 2016; Zhang et al., 2012), Arf1 (Haines et al., 2015; Luchsinger et al., 2018) and Golgi (Ohashi et al., 2018) are all independently

shown to promote drug resistance in cancers, which could be a concerted role. Similarly, AXL (Bi et al., 2017; Onken et al., 2016; Zajac et al., 2020), Arf1 (Casalou et al., 2016; Lewis-Saravalli et al, 2013), and Golgi organization (Isaji et al., 2014; Millarte and Farhan, 2012; Zajac et al., 2020) are independently reported to enhance cancer cell migration. The AXL-Arf1-Golgi organization axis, could have a role in both drug resistance and cancer cell migration.

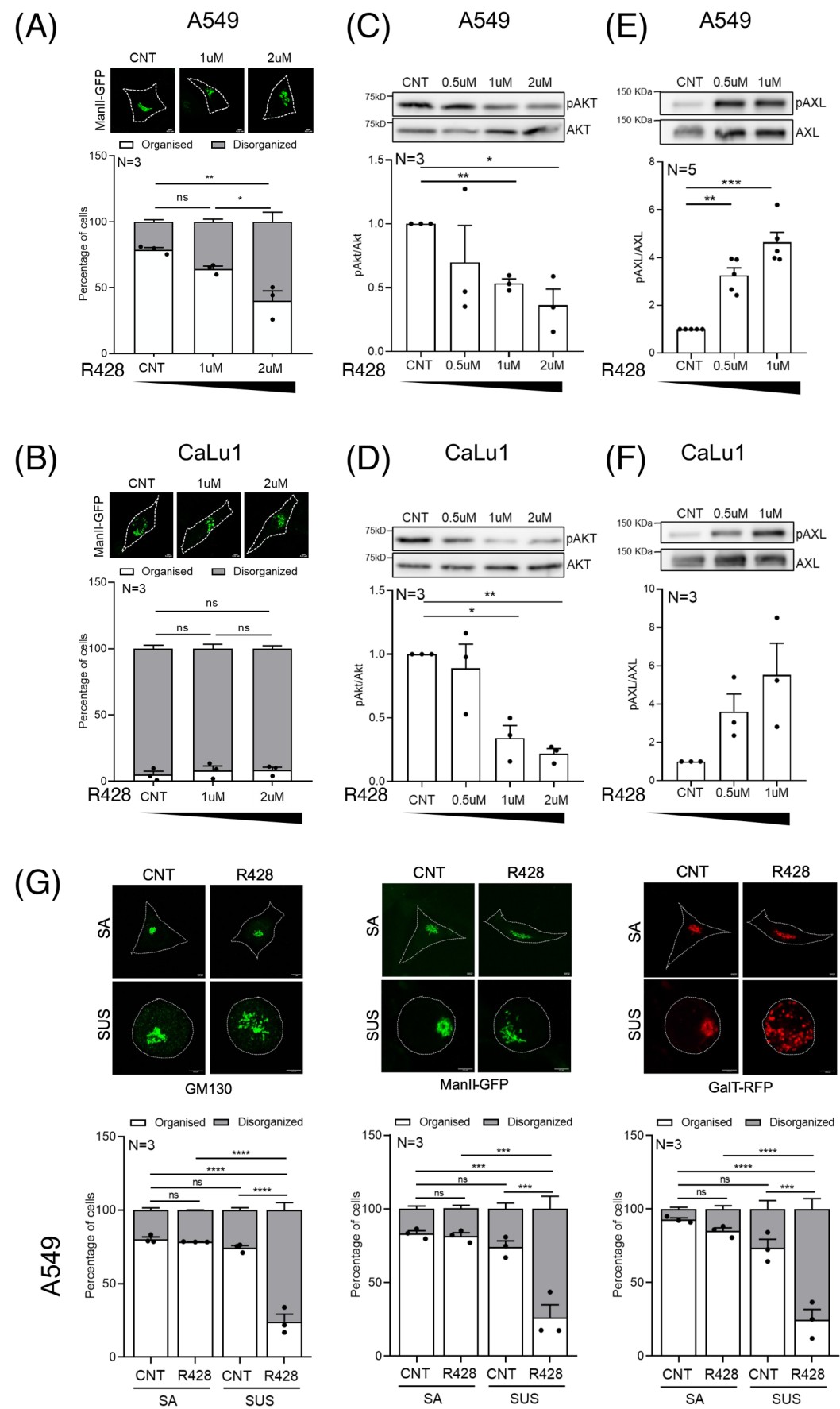

**Fig. 6.** See next page for legend.

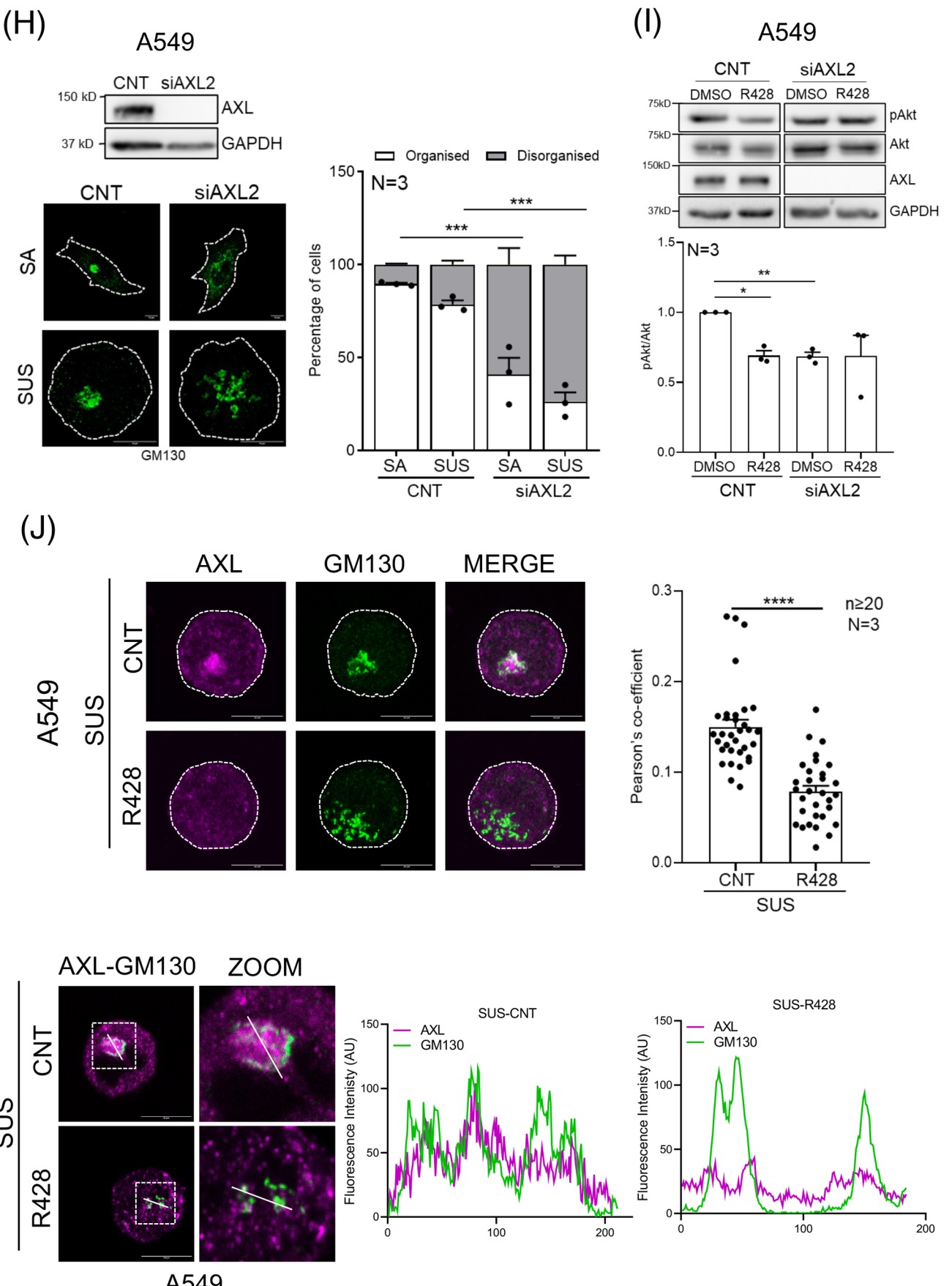

**Fig. 6.** See next page for legend.

**Fig. 6. AXL regulates adhesion-independent Golgi organization in lung cancer cells.** Representative cross-section images of the predominant Golgi phenotype in (A) A549 and (B) CaLu1 cells expressing ManII-GFP (green) treated with DMSO (CNT) or R428 (1 µM or 2 µM). Percentage distribution profile for cells ($n \geq 100$) shows organized (white) and disorganized (grey) Golgi in adherent cells across treatments. The graphs represent mean±s.e.m. from three independent experiments. (C,D) Representative western blots for phosphorylated Akt (pAkt) and total Akt (Akt) in cell lysates from DMSO-treated (CNT), and 0.5 µM, 1 µM and 2 µM R428-treated (C) A549 and (D) CaLu1 cells. (E,F) Representative western blots for Y702 phosphorylated AXL (pAXL) and total AXL (AXL) in cell lysates from DMSO-treated (CNT), and 0.5 µM and 1 µM R428-treated A549 (E) and CaLu1 (F) cells. Graphs represent ratio of densitometric band intensities (normalized to CNT) as mean±s.e.m. from three or five independent experiments. (G) Percentage distribution profile for A549 cells ($n \geq 150$) immunostained for GM130 (green), expressing ManII-GFP (green) or expressing GalTase-RFP (red) with organized (white) and disorganized (grey) Golgi, in DMSO-treated (CNT) or R428-treated stable adherent (SA) or non-adherent (SUS) cells. Representative deconvoluted z-stack maximum intensity projection (MIP) images of the predominant Golgi phenotype are shown. The graph represents mean±s.e.m. of percentage distribution from three independent experiments. (H) Representative deconvoluted MIP images show the predominant Golgi organization phenotype, immunostained for GM130 (green) in control (CNT) and AXL knockdown (siAXL2-treated) stable adherent (SA) versus non-adherent (SUS). Representative western blots for AXL and GAPDH to validate AXL knockdown in cell lysates from CNT versus siAXL2-treated A549 cells. Graph represents mean±s.e.m. of percentage distribution profile for cells ($n \geq 200$ cells) with organized (white) and disorganized (grey) Golgi from three independent experiments. (I) Representative western blot shows the detection of phosphorylated Akt (pAkt) and total Akt (Akt) and total AXL (AXL) and GAPDH in cell lysates from DMSO (CNT) and R428 treated control (CNT) and AXL knockdown (siAXL2) A549 cells. The graphs represent ratio of densitometric band intensities (normalized to CNT) as mean±s.e.m. from three independent experiments. (J) Representative images shown are deconvoluted z-stack MIP, of A549 cells immunostained for AXL (magenta) and GM130 (green), in non-adherent (SUS) cells treated with DMSO (CNT) or R428. The SUS-CNT images in panels G and J show the same representative GM130 immunostaining image, as analysis of both Golgi organization and AXL colocalization was performed in the same experiment. The graph shows the mean±s.e.m. of Pearson's coefficients for colocalization of AXL and GM130 ($n \geq 20$ cells) from three independent experiments. Below are cross-section images and corresponding zoomed images of marked region. Line plot shows intensity profile along the line for AXL (magenta) and GM130 (green) in DMSO (CNT) or R428. The black bar below graphs represents the gradient of increasing concentration of R428 treatment. Statistical analysis was done using one-way ANOVA multiple comparisons test with Tukey's method for error correction, for the distribution profiles, single sample $t$-test for normalized western blotting results and Mann–Whitney $U$ test for Pearson's colocalization analysis. Scale bars: 10 µm. (*$P \leq 0.05$, **$P \leq 0.01$, ***$P \leq 0.001$, ****$P \leq 0.0001$; ns=not significant).

Finally, understanding the regulatory and functional implication of AXL being localized at the Golgi and PM remains of much interest. Arf1 is well established to have roles at both sites (Boulay et al., 2008; Donaldson et al, 2005; Lewis-Saravalli et al., 2013) This could have implications for the AXL-Arf1 crosstalk at both locations. Does AXL regulate Arf1 at the PM, like we show at the Golgi? When displaced from the Golgi on R428 treatment AXL re-distribution in cells is variable, localized in distinct intracellular structures in MDAMB231 and being more dispersed in A549 cells. Re-localization of AXL and Arf1 from the Golgi to the PM could have implications for their role at both sites.

The regulation and significance of AXL phosphorylation at Y702 in this pathway remain of interest. We observe a consistent, time-dependent increase in pAXL (Y702) upon R428 treatment (12 h; MDAMB231 and A549), raising the possibility that elevated pAXL may correlate with AXL inhibition in these cells. Similarly, Arf1

inhibition (GCA) and loss of adhesion both increase pAXL (Y702) levels and disrupt Golgi organization, further supporting this correlation. How these phosphorylation changes reflect AXL inhibition and contribute to downstream functional outcomes requires careful evaluation. Adhesion-dependent regulation of pAXL may offer a useful framework to dissect this relationship.

While AXL as a transmembrane receptor tyrosine kinase has its ligand binding domain localized outside the cell at the PM (Zhu et al., 2019), access to the same could be limited when localized at the Golgi. Understanding how AXL ligand, Gas6, impacts AXL-Arf1-Golgi organization pathway remains to be tested. An additional role for this pathway could also be in cellular mechanosensing. Shown now to be regulated by adhesion, AXL and ROR2 are reported to alter rigidity sensing by regulating local mechanosensory contractions (Yang et al., 2016). Differential regulation of the Golgi in response to mechanical cues, while speculated (Romani et al., 2019), remains to be established. Our independent studies test whether the adhesion-AXL-Arf1-Golgi crosstalk constitutes a mechanosensory pathway that influences cell function (Saha et al., 2026). The identification of an AXL-Arf1-Golgi pathway not just validates the simple screen we began with, but also shows how it could reveal new Golgi regulatory paradigms with implications for normal cell function and disease.

## MATERIALS AND METHODS
### Reagents
Accutase (Cat. No. #A6964), Fibronectin (Cat. No. #F2006) and Triton-X 100 (Cat. No. #T8787) were purchased from Sigma. Lipofectamine 2000 was purchased from Invitrogen (Cat. No. #11668019). Fluorophore-conjugated lectin probes were purchased from Invitrogen Molecular Probes – ConA-Alexa488 (Cat. No. #C11252), WGA-Alexa488 (Cat. No. #W11261), PNA-Alexa488 (Cat. No. #L21409). DMSO was purchased from Sigma (Cat. No. #D2438). Bemcentinib (R428) was purchased from Cayman USA (Cat. No. #21523), and Golgicide A (Cat. No. #G0923) was purchased from Sigma. Compound C inhibitor (Cat. No. #171260) was purchased from Sigma. Immobilon Western Chemiluminescence substrate was purchased from Millipore (Cat. No. #WBKLS0500). Trizol was purchased from Ambion (Cat. No. #15596018). Fluoramount-G was purchased from Southern Biotech (Cat. No. #0100-01). cDNA synthesis kit (Takara Cat. No. #6210A) was obtained from Dr Nishad Matange, IISER Pune. SYBR green mix was purchased from Takara (Cat. No. #RR820A). BCA protein estimation kit (Cat. No. #23225) was purchased from Pierce. DAPI was purchased from Merck (Cat. No. #5.08741.0001).

### Antibodies
pAkt S473 (1:1000) (Cat. No. #4060S), Akt (1:2000) (Cat. No. #9272S). AXL Clone C89E7 (1:2000 for western blotting and 1:400 for immunofluorescence assay) (Cat. No. #8661S) pAXL D12B2 (Y702) (1:500) (Cat. No. #5724S), Phospho-AMPKα (Thr172) (Cat No. #2531), AMPKα (Cat. No. #2532) (1:1000 dilution for western blotting) were purchased from Cell Signaling. GAPDH (1:5000) (Cat. No. #9545) was purchased from Sigma. GM130 Clone 35 (1:250 for western blotting and 1:100 for immunofluorescence assay) (Cat. No. #612008) was purchased from BD Transduction. Arf1 Clone 1D9 (1:500) (Cat. No. # MA3-060) was purchased from Invitrogen. β-tubulin Clone E7 (1:2000) (Cat. No. #AB_2315513) and Alpha Tubulin Clone 12G10 (1:200) (Cat no. #AB_1157911) were purchased from Developmental Studies Hybridoma Bank (DSHB). Anti-Acetylated Tubulin (1:2000) (Cat. No. T7451) was purchased from Sigma. Anti-GBF1 antibody (1:1000 for western blotting and 1:200 for immunofluorescence assay) (Cat. No. ab86071) was purchased from Abcam. Anti-ARF1 antibody (1:3000) (Cat. No. #ab183576) was purchased from Abcam. Secondary fluorescent-conjugated antibodies (Alexa488 and Alexa568) were purchased from Invitrogen Molecular Probes and used at a dilution of 1:1000. HRP-conjugated secondary antibodies were purchased from Jackson Immunoresearch and used at a dilution of 1:5000.

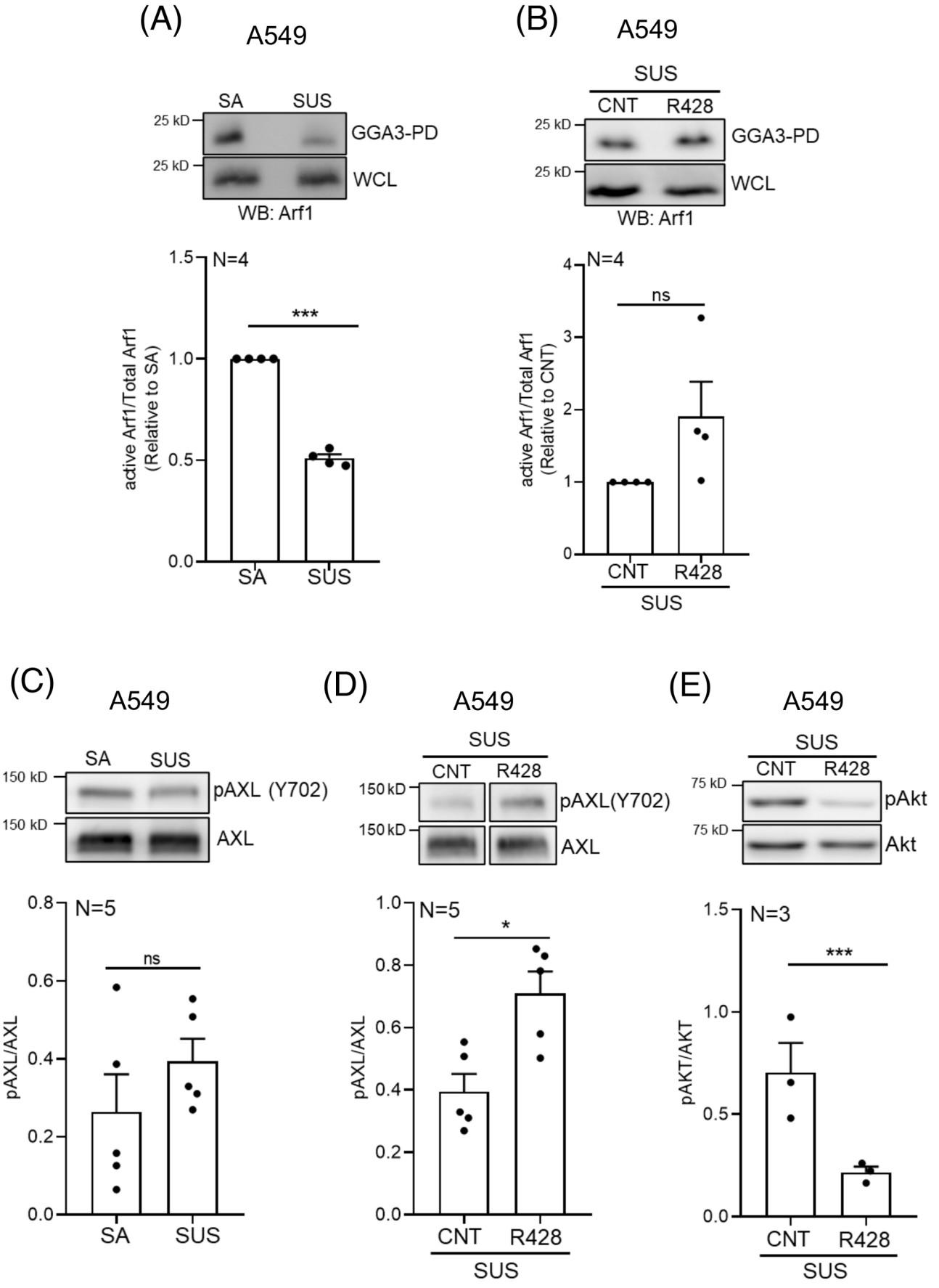

**Fig. 7.** See next page for legend.

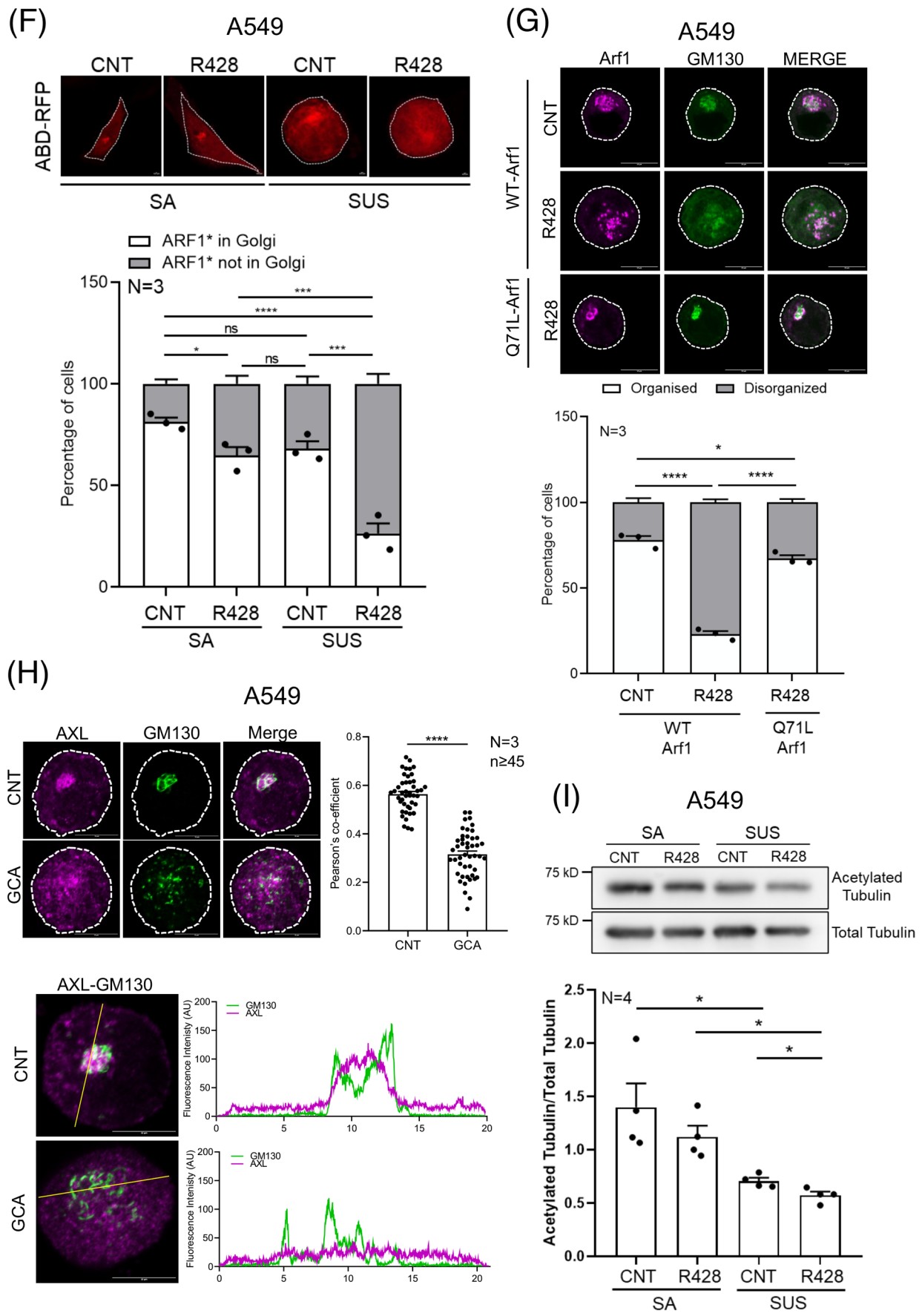

**Fig. 7.** See next page for legend.

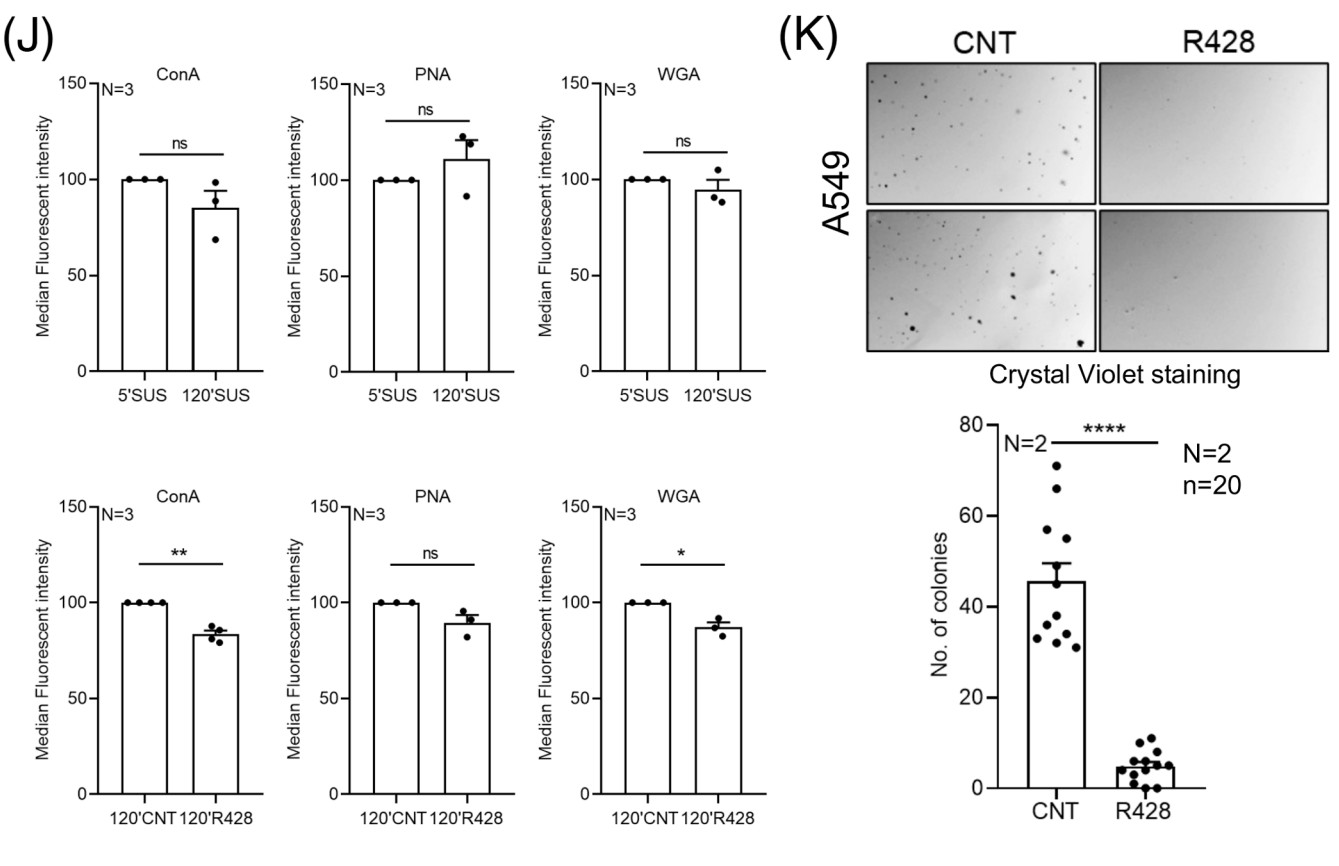

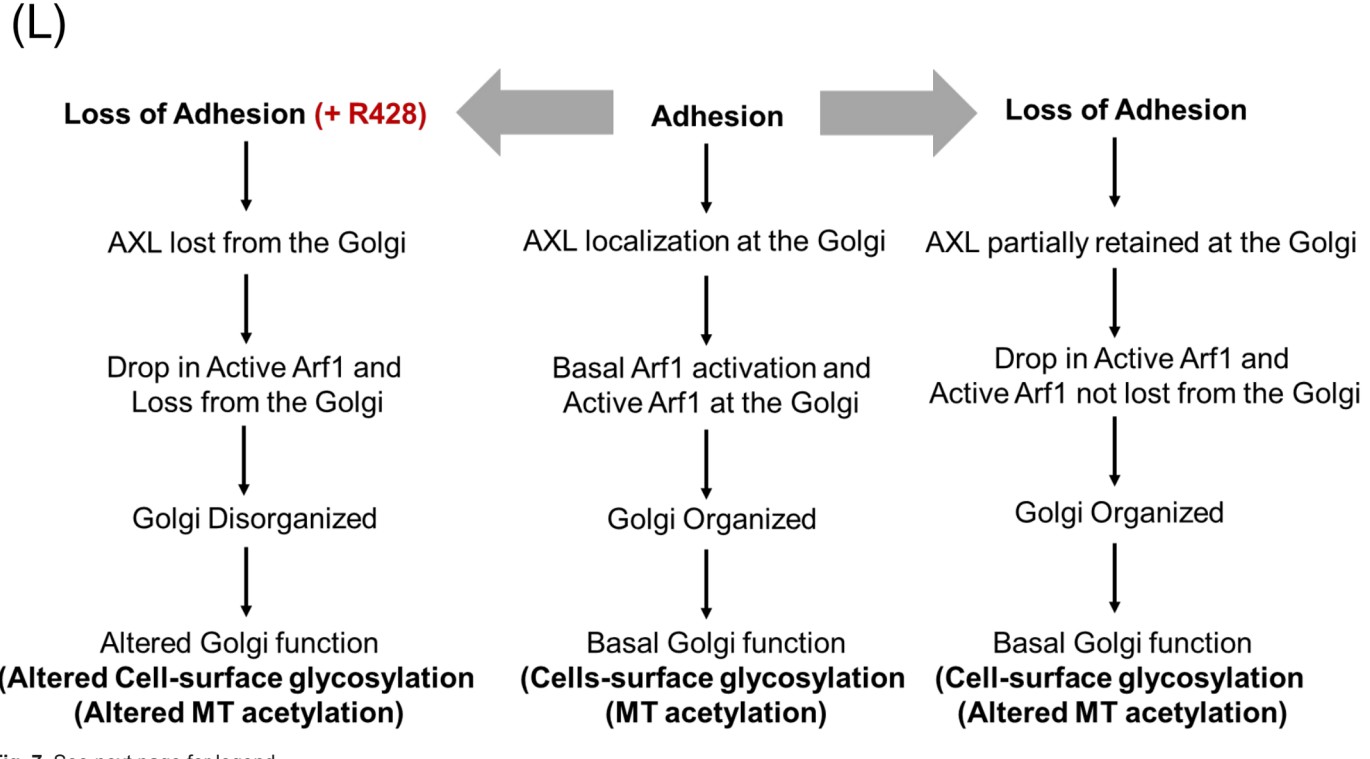

**Fig. 7.** See next page for legend.

**Fig. 7. AXL-Arf1 crosstalk regulates Golgi organization in non-adherent A549 cells.** (A,B) Representative western blots for (A) Arf1 (WB: Arf1) in pulldown using GST-GGA3 (GGA3-PD) and whole-cell lysate (WCL) of (A) stable adherent (SA) and non-adherent (SUS) and (B) non-adherent (SUS) DMSO-treated (CNT) and R428-treated A549 cells. Graph represents ratio of densitometric band intensities (normalized to SA and SUS-CNT) in GGA3-PD to WCL as mean±s.e.m. from four independent experiments. (C-E) Representative western blots for (C,D) Y702 phosphorylated AXL (pAXL), AXL and (E) phosphorylated Akt (pAkt) and Akt in cell lysates of (C) stable adherent (SA) versus non-adherent (SUS) and (D,E) DMSO-treated (CNT) versus R428-treated non-adherent A549 cells. Graph represents ratio of densitometric band intensities as mean±s.e.m. from five or three independent experiments as mentioned in the graphs. (F) Percentage distribution profile for cells ($n \geq 200$ cells) expressing ABD-RFP (red) to detect active Arf1 enriched at an intracellular location, confirmed previously to overlap (white) or not overlap with the Golgi marker (grey) in adherent (SA) and non-adherent (SUS) A549 cells treated with DMSO (CNT) or R428. Representative cross-section confocal images show the predominant Golgi phenotype. The graph represents mean±s.e.m. from three independent experiments. (G) Representative cross-section confocal images show the predominant Golgi phenotype in A549 cells expressing Q71L-Arf1-mCherry (magenta) or WT-Arf1-mCherry (magenta), and immunostained for GM130 (green) in DMSO (CNT) and R428 treated non-adherent A549 cells. Graph shows percentage distribution profile for non-adherent control and R428 treated WT-Arf1 and Q71L-Arf1 expressing cells ($n \geq 100$) with organized (white) or disorganized (grey) Golgi. The graph represents mean±s.e.m. from three independent experiments. (H) Representative cross-sectional confocal images with line plots for colocalization of immunostained AXL (magenta) and GM130 (green) in non-adherent A549 cells treated with DMSO (CNT) and 1 μM GCA. Graph represents Pearson's coefficients for AXL (magenta) and GM130 (green) colocalization plotted as mean±s.e.m. for $n \geq 45$ cells from three experiments (I) Representative western blot for acetylated tubulin and total tubulin in cell lysates from DMSO-treated (CNT) and R428-treated stable adherent (SA) and non-adherent (SUS) A549 cells. The graph represents ratio of densitometric band intensities as mean±s.e.m. from four independent experiments. (J) Graphs represent mean±s.e.m. of median fluorescent intensities of cell surface-bound ConA, PNA and WGA lectins, detected by flow cytometry at early (5 min) or late (120 min) suspension timepoints (normalized to 5 min) and in DMSO-treated (CNT) and R428-treated (normalized to CNT) A549 cells from three independent experiments. (K) Representative images of Crystal Violet-stained colonies in soft agar of DMSO-treated (CNT) and R428-treated A549 cells. The graph represents mean±s.e.m. of number of colonies counted from 20 images obtained from two independent experiments. (L) Schematic describes the AXL-Arf1 crosstalk and its regulation of Golgi organization and function downstream of adhesion, loss of adhesion and loss of adhesion on R428 treatment in A549 cells. Statistical analysis was done using one-way ANOVA multiple comparisons test using Tukey's method for error correction for the distribution profiles, Mann–Whitney $U$ test for non-normalized western blotting results and AIG assay, and single sample $t$-test for normalized flow cytometry results. Scale bars: 10 μm. (*$P \leq 0.05$, **$P \leq 0.01$, ***$P \leq 0.001$, ****$P \leq 0.0001$; ns=not significant).

## Plasmids and oligonucleotides

GalTase-RFP and Mannosidase-GFP constructs were obtained from Dr Jennifer Lippincott Schwartz (Howard Hughes Medical Institute). ABD-GFP and ABD-RFP constructs were obtained from Dr Satyajit Mayor, National Centre for Biological Sciences, Bangalore (NCBS), India. mCherry-tagged Arf1-WT and Arf1-Q71L constructs were made by releasing the Arf1 gene from GFP constructs and cloning the same into an empty mCherry-N1 vector. Primers used for RT-PCR experiments were designed using a Primer design tool from Integrated DNA Technologies and were ordered from Eurofins. Primer sequences for AXL were as follows: Forward – GTCT-AGCTGACCGTGTCTAC; Reverse – CCTGGCGCAGATAGTCATAA.

AXL siRNA sequences were confirmed in earlier studies (Holland et al., 2010). AXL and Arf1 siRNAs were purchased from Merck (Sigma) and were as follows. siAXL#1: Forward – GAAAGAAGGAGACCCGGTTA; Reverse – TAACGGGTCTCCTTCTTTC. siAXL#2: Forward – CCAA-GAAGATCTACAATGG; Reverse – CCATTGTAGATCTTCTTGG.

siArf1: Forward – ACAGCAAUGACAGAGAGCGUGUGAA; Reverse – TTCCCGCTCTCTGTCTTGCTGT. All sequences were checked and confirmed for target specificity using the NCBI nucleotide BLAST tool.

## Cell culture and transfections

MDAMB231 and CaLu1 cell lines were obtained from European Collection of Authenticated Cell Cultures. A549 cell line was obtained from American Type Culture Collection. The MCF7 cell line was obtained from Professor Sanjay Gupta at ACTREC, Navi Mumbai, India. All the above cell lines were cultured using Gibco DMEM from Thermo Fisher Scientific, with the addition of 10% Penstrep and 5% FBS. 0.05% Trypsin was used to detach cells, and an excess culturing medium was used to neutralize the action of trypsin. BEAS2B cell line was obtained from Professor Shantanu Chaudhary's laboratory at Institute of Genomics and Integrative Biology, New Delhi, and the cells were cultured using LHC9 media. 0.05% trypsin was used to detach cells, and trypsin inhibitor purchased from Roche (Cat. No. #10109886001) was used to neutralize trypsin. MCF10A cell line was obtained from Dr Madhura Kulkarni's laboratory at Centre for Translational Cancer Research, IISER Pune and Prashanti Cancer Care Mission (PCCM), and the cells were cultured using Gibco DMEM/F12 media (Cat. No. #12500062) with the addition of 5% horse serum (Invitrogen, Cat No. #16050-122), 10% Penstrep, 20 ng/ml EGF, 0.5 mg/ml hydrocortisone (Sigma, Cat. No. #H0888-5G), 100 ng/ml cholera toxin and 10 μg/ml (Sigma, Cat. No. C8052-1MG) and Insulin (Sigma, Cat. No. #I1882-100MG). For transfection studies, cells were seeded in 6 cm dishes to attain a confluency of 60% and allowed to attach and spread for 5 h. Using Gibco OptiMEM medium and transfection agent Lipofectamine 2000 from Thermo Fisher Scientific, the transfection mix was prepared and kept at room temperature for 30 min before adding to the cells seeded in 6 cm dishes. Medium in transfected dishes was changed 12 h post-transfection, and cells were used for experiments 36 h post-transfection.

## Suspension assay experiments

Cell lines maintained in their respective media were grown up to 75% confluency in 10 cm or 6 cm dishes, as required. 0.05% trypsin or Accutase (for lectin labelling experiments) was washed out using excess media to detach cells. One aliquot was kept aside from collected cells and processed for the 5-min (just detached) timepoint if required. For the suspension assay, cells were gently mixed with a culture medium containing 1% methylcellulose and incubated at 37°C with 5% $CO_2$ for a specified duration. Post-suspension, methyl cellulose was washed out using culture medium, and 4°C conditions were maintained throughout to harvest cells from suspension. Collected cells were distributed to suspension and re-adhesion time points as required. For re-adhesion time points, cells post-suspension was seeded on 22×22 mm coverslips or 6 cm dishes pre-coated with 2 mg/ml or 10 mg/ml Fibronectin, respectively. For inhibition experiments, respective inhibitors were added to the suspension mix, media washes and re-adhesion timepoints. Each timepoint was then processed for lectin labelling, fixed with PFA for immunofluorescence assay (IFA)/imaging, or kept at −80°C for preparing cell lysates and for active Arf1 pulldown experiments. Cell numbers used for suspension assays were variable depending on the experiment.

## Lectin labelling and flow cytometry analysis

Cells harvested from suspension were given a PBS wash to remove residual media, then added to a lectin labelling mix containing fluorophore-conjugated lectin diluted in 1× PBS at optimized lectin concentration. The labelling reaction was kept in ice under dark conditions for 15 min, followed by two washes with cold 1× PBS. PFA (3.5%) was used to fix the lectin-labelled cell samples, which were then resuspended in cold 1× PBS (350 μl) for flow cytometry analysis. Samples were run on BD Celesta Flow Cytometer, and data analysis was done using BD FACS Diva software. A morphologically uniform cell population presented by the forward and side scatter (FSC and SSC) plot was selected using polygon gating for data acquisition. A maximum of 15,000 events were recorded in the gated area for every sample, and the median fluorescent intensity obtained for the selected population was used.

## IFA

For IFA, cells fixed with PFA were incubated in a permeabilization buffer for 15 min at room temperature. A permeabilization buffer was made with Triton X-100 (0.05%) diluted in 5% BSA and made in 1× PBS. Post permeabilization, two washes were given with 1× PBS followed by blocking with 5% BSA at room temperature for 30 min to 1 h. Two washes were given after blocking. Samples were incubated with primary antibody overnight at 4°C. Three washes were given post-incubation with primary antibody followed by a 1-h incubation with secondary fluorescent antibody (1:1000 dilution in 5% BSA) at room temperature. Samples were given three washes and then mounted on slides using Fluoramount-G. Slides were maintained at room temperature under dark conditions to dry appropriately, then moved to 4°C until confocal imaging.

## Inhibitor studies

Cells grown to 75% confluency were treated with increasing concentrations of AXL inhibitor R428 (0.5 μM, 1 μM, 2 μM) for 12 h before using cells for preparing lysates or preparing slides for confocal imaging. Optimal 1 μM R428 concentration was chosen similarly used to treat cells for 12 h before being held in suspension with the inhibitor and further processed for lysis or confocal imaging. Cells were incubated with 1 μM R428 for increasing timepoints (10 min, 20 min, 30 min, 40 min, 60 min and 12 h) to evaluate its effect in time kinetic studies. For experiments inhibiting Arf GEFs, cells were suspended for 90 min and then treated with 0.5 or 1 μM GCA for 30 min in suspension. Adherent cells were treated with GCA (0.5 μM/1 μM) for 30 min and lysed or fixed as required. In experiments using R428 or GCA, DMSO was used as a control. During these experiments, respective inhibitors were included in all incubation steps of the protocol. In suspension assays, inhibitors were also included in the washes between incubation and lysis or fixing.

## siRNA-mediated knockdowns

For siRNA-mediated knockdowns, cells were seeded in 3.5 cm dishes to attain 50% confluency and allowed to attach and spread for 4 h. Transfection mix was prepared using Gibco OptiMEM, transfection reagent RNAi MAX and 50 pmol siAXL and allowed to incubate at room temperature for 30 min before adding to the cells seeded on 3.5 cm dishes ($5 \times 10^5$ cells for MDAMB231 and $4 \times 10^5$ cells for A549). A second shot of siRNA was given to cells after 24 h. Cells were used for experiments after 48 h post-second shot. Control cells were treated with only RNAi MAX.

## Arf1 activity assay

Arf1 activation was assessed using a GGA3 pulldown assay, in which GST-GGA3 fusion protein bound to glutathione-sepharose beads selectively captures GTP-bound Arf1. GGA3 (Golgi-localized Gamma-adaptin ear homology domain) ARF-binding protein functions as an effector of Arf1-GTP, and bait to pulldown active Arf1 from cell lysates. Pulldown samples and whole-cell lysates were immunoblotted for Arf1, and the ratio of pulldown to total Arf1 was used as a measure of Arf1 activation. For each experiment, pulldown assays were performed simultaneously for all time points/samples to ensure equal bead usage. Samples were either processed immediately at the respective time points or snap-frozen at −80°C until use. Pulldowns were carried out using glutathione-sepharose A beads pre-bound to 60 μg GST-GGA3. Samples were lysed with 500 μl activity assay buffer, of which 400 μl was incubated with beads at 4°C for 35 min at 9 rpm. The remaining 100 μl of lysate was mixed with 5× Laemmli buffer and used as the whole cell lysate fraction. Post incubation with beads, samples were washed, and GST-GGA3-bound Arf1 was eluted with 1× Laemmli buffer. Samples were boiled at 95°C and given a short spin before performing western blotting. Blots were incubated with the required antibodies developed using Immobilon reagents and analysed using ImageJ software (described in the 'SDS-PAGE and western blotting' section).

To determine the percentage activity of Arf1, the following calculation was used as described in earlier studies:

$$\text{Percentage activity} = \frac{\text{pulldown band intensity} \times 100}{\text{corresponding WCL band intensity} \times \text{dilution factor}}.$$

## Protein estimation with BCA kit

Samples were lysed in RIPA buffer with freshly added protease inhibitors and kept in ice for 30 min of lysis. Lysates were then spun down at 14,000 rpm, 4°C for 15 min. Supernatant was collected in a fresh tube as the lysate. 10 μl of samples was set aside for BCA, and the rest was lysed in Laemmli buffer and boiled at 95°C, followed by a short spin. Lysates were stored at −20°C until gel run and western blotting. Protein estimation was done as follows: in a 96-well plate, unknown samples diluted to 1:5 times with RIPA buffer were loaded in triplicates along with freshly prepared standards for BSA diluted in RIPA buffer (range – 0 mg/ml to 2 mg/ml). Working reagent from the Pierce BCA kit was prepared by mixing Reagent A and Reagent B in a ratio of 50:1. 200 μl of working reagent was added to each well containing 10 μl of sample or BSA standard. The plate was incubated at 37°C for 30 min and then scanned at 562 nm to determine absorbance using a plate reader (PerkinElmer, Ensight). Absorbance values for BSA concentrations were plotted to obtain a standard curve, which determined the concentration for the unknown sample.

## SDS-PAGE and western blotting

Required amount of protein (20 μg, 30 μg) or cell equivalent amount of lysate was loaded in gels for SDS-PAGE. Protein samples run on gels were transferred to the PVDF membrane using a methanol-containing buffer or sodium bicarbonate buffer. Post transfer, the PVDF membrane was blocked in 5% skimmed milk (made in 1× TBS-Tween 20 (0.1%)) at room temperature for 1 h at 10 rpm. The membrane was washed with 1× TBST; then, individual blots were incubated in respective primary antibody dilutions in 5% BSA in 1× TBST overnight at 4°C. Blots were washed three times with 1× TBST, followed by incubation at room temperature with respective HRP-conjugated secondary antibody solutions made in 2.5% skimmed milk in 1× TBST. After this incubation, blots were washed thrice with 1× TBST and developed with direct or diluted (with 1× TBST) Immobilon substrate. Blot images were captured using LAS4000 Chemiluminescent Imager. Densitometric analysis was done using ImageJ FIJI software. Blot transparency data are shown in Fig. S7.

## Anchorage-independent growth (AIG) assay

Cells grown to 75% confluency were treated with DMSO or R428 (1 μM) for 12 h before performing anchorage-independent growth assay. Detached cells were counted, and 5000 cells from control and R428 treated cells each were mixed with 0.3% agar containing DMEM and layered on top of 0.5% agar base per well in six-well plates. Each of the control and inhibitor-treated cells were plated in duplicates. The agar was allowed to solidify, and 1.5 ml of culture medium with DMSO or R428 was added. These dishes were maintained in the incubator for 15 days, and DMSO or R428 was added freshly when culture medium was changed every 3 days. The colonies formed in agar at the end of the 15-day incubation were stained with 0.05% Crystal Violet dissolved in 20% ethanol for 1 h at room temperature and de-stained by repeated washing with distilled water until stained colonies were visible. The colonies were then imaged on an Olympus MVXC10 microscope at 1× zoom with 1× objective in the HDR mode and counted using the particle analysis tool of ImageJ software.

## Confocal imaging

Slides were imaged using the ZEISS confocal (LSM710) using a 63× oil immersion objective. For profiling Golgi organization, cells observed at confocal were categorized manually as being with organized Golgi or disorganized Golgi. For ABD-RFP profiling studies, cells showing localization of ABD-RFP at the Golgi versus cells without were identified and counted. A minimum of 100 and a maximum of 200 cells were counted for each experimental treatment and categorized according to their phenotype to arrive at a percent population of cells in either category. Images were captured with an average of 4, zoomed at 5×/6× for suspended cells and 2× for stable adherent cells. A scan speed of 5 for cross-sectional images and 7 for obtaining z-stacks was used. Leica DM6 was used for evaluating active Arf1 localization using an ABD-RFP marker expression in cells.

## Image analysis and quantitation

ImageJ FIJI was used for all imaging experiments to process images with scale bars until otherwise specified. Overlap of markers in images when

looked at using a line plot was done using the ImageJ FIJI software. Huygens Professional software (SVI) was used to deconvolution LSM files. Deconvoluted images were used to generate maximum intensity projection (MIP) images and 3D surface-rendered (SR) images of the Golgi using the Visualization tool in Huygens. Colocalization analysis for data showing AXL, ABD-GFP and GM130 markers were performed using the Huygens software 'Colocalization Analyzer Advanced' tool. Costes' method was used to set a threshold for each image, and the GM130 or ABD-GFP channel was manually thresholded and analysed by the software to obtain Pearson's coefficient value.

### Gene expression analysis for regulators of Golgi organization (*in silico* study)

A comprehensive listing of genes associated with the Golgi apparatus was done using the NCBI Gene Ontology tool. A search using the terms 'Golgi organization human' yielded 131 human genes associated with the term 'Golgi organization'. Gene expression data (RPKM) were obtained from the CCLE database using the UCSC Xena browser for MDAMB231, MCF7, A549 and CaLu1 cell lines. Fold change in gene expression was calculated between the pair of breast cancer (BC) cell lines and the pair of lung cancer (LC) cell lines independently, and a difference of $\geq 10$ (BC) $\geq 5$ (LC) was considered significant to obtain the DEGs. This cut-off was set to ensure that the number of genes picked is in the top 50. STRING-DB version 11.0 created a primary protein interaction network for each DEG. No restriction was applied over the number of interactors displayed in the network to cover all possible primary interactions. Interactions based on experimental evidence with a confidence score of $\geq 0.4$ were considered in the study. Each of the primary interactors obtained was evaluated for any known role in the regulation of Golgi organization or Golgi function.

### Scoring of genes

Knowing the Golgi organization phenotype in the selected cell lines, the primary list of selected genes was scored based on available data in the literature for the effect of gene knockdown on Golgi organization. Accordingly, genes for which knockdown affected the Golgi phenotype comparable to a drop in levels observed in cancer cell lines were scored as 2. Those that are involved with the Golgi in literature studies but for which expression did not agree with their knockdown Golgi phenotype from literature were scored as 1. Genes having no available data to suggest their knockdown affects Golgi organization were scored as 0. A second score was given based on the number of primary interactors obtained from STRING analysis that have a direct role in Golgi's organization or function. The two scores were added for each gene, and the DEGs with a combined score of 3 or higher were selected for further evaluation. This gave us a shortlist of 20 DEGs for breast cancer and 14 DEGs for lung cancer. Amongst them, eight genes were common between both breast and lung cancer. Having looked at genes with the top three scores in this list of eight, we chose AXL to evaluate its role in regulating Golgi organization and function in these cancers.

### RNA isolation and cDNA synthesis

For preparing RNA samples, $8 \times 10^5$ cells were seeded for 14 h were lysed in Trizol (Ambion), followed by RNA isolation using the PCI method (phenol, chloroform, and isopropyl alcohol). After ethanol washes, the RNA was allowed to dry for 15 min at 37°C and resuspended in 50 µl nuclease-free water. RNA concentration and purity were checked using NanoDrop 2000. 1 µg of RNA was mixed with gel loading buffer and run on 10% agarose gels to check the quality of RNA samples. These RNA sample stocks were stored at −80°C. Using the PrimeScript cDNA synthesis kit (Takara), cDNA samples were prepared from at least five sets of RNA (1 µg) for each MDAMB231, MCF7, A549 and CaLu1 cell line. Details of the PCR cycle run were as follows: 25°C for 2 min, 25°C for 15 s, 37°C for 25 min, 85°C for 1 min 30 s. cDNA samples were stored at −20°C for real-time PCR studies.

### Real-time PCR

Primers were tested for efficiency by running RT-PCR for a range of cDNA dilutions. Primer efficiency was determined by three criteria – single peak for melt curve, Ct value for the highest concentration of cDNA, and slope

value for a range of cDNA concentrations. Post RT-PCR, the product was run on agarose gels to confirm if the product ran at the expected band size. All RT-PCR experiments for comparative gene expression analysis were performed using 1:1 dilution of cDNA. SYBR Green Mix (Takara) was used to detect and quantify gene expression. Actin was used as the standard control. 5 µl reaction mix was loaded in triplicates for each sample. The plate was given a quick spin before running the RT-PCR cycle, the protocol for which was as follows: 95°C for 3 min (once per sample). Plate reading was recorded after the two steps of 95°C, 10 s − 60°C; 25 s; this cycle was repeated 40 times per sample. Recorded data was analysed for delta Ct and fold change. The relative gene expression between MDAMB231 and MCF7 or A549 and CaLu1 cell lines was compared to confirm the differential expression of AXL.

### Statistics

All the statistical analysis was done using GraphPad Prism analysis software. Statistical analysis for western blotting using absolute data (not normalized to a condition or control) was done using the two-tailed unpaired Mann–Whitney $U$ test. For western blot comparisons, data normalized to control or a certain experimental condition, the two-tailed single-sample $t$-test was used. Statistical analysis for Golgi profiling data, comparing the percentage of cells with a specific Golgi phenotype at different experimental or treatment conditions, was done using one-way ANOVA, a multiple comparison test, with Tukey's method for error correction. The Pearson's coefficient analysis data for localization of AXL at the Golgi was tested for statistical significance using one-way ANOVA, a multiple comparison test with Tukey's method for error correction.

### Acknowledgements

We acknowledge the extensive support provided by the IISER Pune Microscopy Facility for imaging and the IISER Pune FACS Facility.

### Competing interests

The authors declare no competing or financial interests.

### Author contributions

Conceptualization: P.J., A.S., R.M., N.B.; Data curation: P.J., A.S., R.M., D.P., G.M., V.S., N.B.; Formal analysis: P.J., A.S., R.M., D.P., M.P., G.M., V.S.; Funding acquisition: N.B.; Investigation: P.J., A.S., R.M., D.P., M.P., G.M., V.S., N.B.; Methodology: P.J., A.S., R.M., N.B.; Project administration: N.B.; Resources: N.B.; Software: N.B.; Supervision: N.B.; Validation: P.J., A.S., R.M., D.P.; Visualization: P.J., A.S., R.M., N.B.; Writing – original draft: P.J., A.S., R.M., N.B.; Writing – review & editing: P.J., A.S., R.M., N.B.

### Funding

This work is supported by an Anusandhan National Research Foundation, formerly Science Engineering and Research Board, CRG grant (CRG/2022/001813) to N.B. P.J. was supported by a fellowship from Indian Institute of Science Education and Research Pune (IISER Pune). A.S. was supported by Prime Minister's Research Fellowship and a fellowship from IISER Pune. R.M. is supported by a fellowship from Council of Scientific and Industrial Research, India. Open Access funding provided by IISER Pune. Deposited in PMC for immediate release.

### Data and resource availability

All relevant data and details of resources can be found within the article and its supplementary information.

### First Person

This article has an associated First Person interview with the joint first authors of the paper.

### Peer review history

The peer review history is available online at https://journals.biologists.com/bio/lookup/doi/10.1242/bio.062581.reviewer-comments.pdf

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
