## [Peer Review File · Biology Open]

AXL receptor tyrosine kinase regulates Golgi organization and function via an adhesion-Arf1 signalling axis in breast and lung cancer cell lines

Prachi Joshi, Arnav Saha, Radhika Malaviya, Debiprasad Panda, Manojeeet Pattanayak, Grishma Mehta, Vibha Singh and Nagaraj Balasubramanian
10.1242/bio.062581

Editor: Alissa Armstrong

Review timeline

Submission to sister journal:	28 February 2025
Editorial decision at sister journal:	30 May 2025
First revision received:	22 August 2025
Editorial decision at sister journal:	29 September 2025
Second revision received:	21 November 2025
Editorial decision:	21 January 2026
Transfer to Biology Open:	21 March 2026
Accepted:	31 March 2026

Original submission to sister journal

First decision letter

MS Title: AXL receptor tyrosine kinase regulates Golgi organization and function via an adhesion-Arf1 signalling axis in breast and lung cancer cell lines

Authors: Nagaraj Balasubramanian, Prachi Joshi, Arnav Saha, Radhika Malaviya, Debiprasad Panda, Manojeeet Pattanayak, Grishma Mehta and Vibha Singh

We have now reached a decision on the above manuscript.

To see the reviewers' reports and a copy of this decision letter, please go to:

As you will see from their reports, the reviewers raise a number of substantial criticisms that prevent me from accepting your paper for publication.

I am very sorry to give you such disappointing news, but we are currently under great pressure for space and it takes a very enthusiastic recommendation by the referees for a manuscript to be accepted.

I do hope you find the comments of the reviewers helpful in allowing you to revise the manuscript for submission elsewhere, and many thanks for sending your work to us.

Reviewer 1

SUMMARY OF THE ADVANCE MADE IN THIS PAPER AND ITS POTENTIAL SIGNIFICANCE TO THE FIELD
This manuscript presents evidence of a role for the tyrosine kinase Axl in the organization of the Golgi apparatus and the response of this organization to adhesion-mediated signaling. The authors frame this study in the context of cancer, as Axl is often dysregulated in transformed cells. Their data suggest that Axl function is coordinated with that of Arf1, a small GTPase that controls many aspects of Golgi function. This is an interesting and poorly understood connection with potentially important implications for cancer cell biology.

SUGGESTIONS TO AUTHORS

Although the authors are commended for testing this Axl/Arf1/Golgi axis in several different cancer cell lines, the results are confusing and sometimes contradictory. The clearest results are observed via comparison of two breast cancer cell lines, MCF7 (Axl-low) and MDA-MB-231(Axl-high) where the presence of Axl appears to be important for Golgi organization. Things get more confusing when compared to lung cancer lines (A549 and CaLu-1) which exhibit very different responses to Axl inhibition or knockdown, and to loss of adhesion. The authors might be better served to focus solely on the breast cancer lines, in which there is a clearer case to be made for Axl function.

While the data as presented are intriguing, there are many problems with the manuscript in its current form. These include:

1. Almost of all of the images need higher magnification insets or separate panels to support the conclusions drawn by the authors. Specifically, it is not clear whether Axl localizes to the entire Golgi stack or to individual cisternae. It is impossible to tell at the resolution of the images provided.
2. A related point - line scans are a poor substitute for unbiased colocalization analysis (Pearson's and/or Manders' coefficients). While Pearson's coefficients are shown for some sets of images, they should be used throughout.
3. A higher N is needed for all of the imaging. 4 cells is simply not enough to draw statistically significant conclusions.
4. Presumably most of the Axl staining is of endogenous protein? This is never clearly stated.
5. How does inhibition of Axl activity lead to its increased phosphorylation (Fig. 3D)?
6. The authors note that Akt is downstream of Axl. What happens to Golgi organization if Akt is inhibited?
7. The relationship between Axl and Arf1 is confusing. A previous study reported that inhibition of Axl enhances its association with Arf1 and increases Arf1 activity. Here using the same cell line (MDA-MB-231), Axl inhibition apparently reduces Arf1 activation (though not by much) while retaining the ability of the two proteins to co-precipitate. What is the evidence that Arf1 interaction with Axl is GTP-dependent?
8. Although the Axl inhibitor R428 appears to cause its dissociation from the Golgi (Fig. 4A), it is still punctate. Where is it? Because it is a transmembrane protein it must be associated with other intracellular membranes, but what are they?
9. The authors introduce the GGA pulldown to measure Arf activation but don't describe it. This would be useful for investigators unfamiliar with Arf biology.
10. The authors claim that R428 reduces Axl colocalization with Arf1 in adherent MB-231 cells (Fig. 4F) but the images shown are not convincing. And Pearson's would be required to quantify any loss of colocalization.
11. The authors suggest that phosphorylation of Axl at Y702 is inhibitory. What is the evidence for this?
12. Fig. 5A shows that Arf1 activity is reduced in suspended MB-231 cells, yet the amount of Axl that coprecipitates with it is unchanged (Fig. 5B). How do the authors explain this?
13. There is also a drop in Akt activity upon suspension of MB-231 cells (Fig. 5D) but this could be due to a loss of more proximal adhesion-dependent signaling (e.g. FAK).
14. The authors claim that loss of adhesion does not affect Axl phosphorylation (Y702) in A549 cells (Fig. 7D), but the blot shown certainly suggests that it does, and to a similar extent as Axl inhibition with R428.
15. In their discussion the authors state that Axl inhibition leads to loss of both Axl and Arf1 from the Golgi in both MB-231 and A549 cells, yet Arf1 activation is only affected in MB-231. This doesn't seem possible, as Arf1 activity requires its association with membranes. How do the authors explain this?

Reviewer 2

SUMMARY OF THE ADVANCE MADE IN THIS PAPER AND ITS POTENTIAL SIGNIFICANCE TO THE FIELD

In this manuscript, Prachi Joshi, Arnav Saha and colleagues set out to characterize proteins that regulate Golgi morphology of cancer cells, in a manner dependent on their adhesion status. After carrying out a literature analysis, they chose AXL, a plasma membrane (PM) receptor tyrosine kinase, for their study. They show that AXL localizes to the Golgi, and claim that AXL interacts at the Golgi with Arf1-GTP (but see point 4 below). The authors show that there is a decrease in phosphorylation of Akt and an increase in phosphorylation of AXL upon treatment of MDAMB231 cancer cells with the AXL inhibitor R428. They show that the Golgi is disorganized upon treatment of these cells with R428, but that MCF7 cells, which do not express AXL, fail to undergo a change in Golgi organization upon R428 treatment. These and other experiments lead the authors to conclude

that upon loss of adhesion, AXL is regulating Golgi morphology through interaction with Golgi-localized Arf1-GTP. However, the authors have only shown correlations between perturbations of AXL localization to the Golgi and Golgi morphology changes. The experiments to address mechanisms, by assaying Arf1 activation status upon various conditions, do not demonstrate that it is the Golgi pool of Arf1 that is involved in the interaction. It has already been demonstrated that Arf1 is recruited to the PM upon EGFR signalling and interacts with AXL (Haines et al. 2015 *Cancer Biol Ther.* 16(10):1535-1547 Boulay et al. 2008 *J Biol Chem.* 283(52):36425-34). Indeed, previous work has shown that a number of PM signalling pathways affect Golgi structure and function (Chia et al. 2012. *Mol Syst Biol.* 8:629). Here, insufficient data is presented to determine whether the effects seen on the Golgi in this manuscript are indirect effects of previously demonstrated mechanisms at the PM. In addition, there are some critical control experiments that are missing. Because of these limitations, the data presented remain descriptive and do not clearly show a new mechanism by which AXL regulates Golgi structure and function.

SUGGESTIONS TO AUTHORS

1. In Figure 1, four different cancer cell lines, transfected with a cis-medial Golgi marker (ManII-GFP) and a trans-Golgi marker (GalTase-RFP), were assayed for Golgi structure. Two of these cell lines showed compact Golgi morphology (called "organized"), and two showed more dispersed Golgi morphology (called "disorganized"). Adherent or non-adherent cells for each cancer cell line were then examined for Golgi organization. For three of the cell lines, both adherent and non-adherent cells had similar levels of Golgi organization. In the fourth cell line, MDAMB231 breast cancer cells, the authors conclude that nonadherent cells had disorganized Golgi and adherent cells had compact Golgi, stating « In non-adherent MDAMB231 cells the Golgi is dispersed (Fig 1B) as seen in the non-transformed breast epithelial MCF10A cells. ». This result is clear according to the quantifications shown, but the adjacent images show the opposite result (Fig. 1B). Whether this discrepancy is due to mislabelling of "SA" and "SUS" images or not must be addressed. In Fig. 1C as well, SA and SUS labels are reversed compared to the adjacent quantifications.

2. In the "in silico" analysis shown in Fig. 2, were any proteins whose major function is at the Golgi identified in the pipeline shown in Fig. 2A?

3. It is necessary for the authors to carry out a rescue experiment in MCF7 cells in order to determine whether the phenotypes reported are due to lack of expression of AXL or some other difference between these cells and those expressing AXL, such as MDAMB231 cells.

4. In Fig. 4C, the effect of R428 on the level of Arf1-GTP in MDAMB231 cells, as monitored by a GGA3 pull-down assay, is very small. The variability in the control levels of Arf1-GTP is masked by normalizing the levels in the R428-treated samples to the control condition. Why are Arf1-GTP levels normalized in this way when such normalization is not carried out in other cases, such as in Fig. 4D? What concentration of R428 was used in this experiment?

5. Page 16, lines 29-31. The conclusion that "GGA3-GST pulldown of active Arf1 also brings down AXL with it; this association unaffected by R428 treatment (Fig 4D)" requires further controls, such as experiments to determine whether AXL binds to beads alone or to GGA3 without Arf1.

6. In the Abstract, Page 2 lines 16-19, the sentence "AXL prominently localized at the Golgi, undergoes displacement from the Golgi when inhibited by R428 and knocked down using siRNA, causing the Golgi to disorganize" is hard to understand. The authors are stating that AXL undergoes displacement from the Golgi when knocked down by AXL siRNA treatment, which is of course the case because AXL levels are drastically reduced. What exactly is "causing the Golgi to disorganize" in the statement above is not clear.

First revision

Author response to reviewers' comments

Dear Reviewers,

We thank you for the constructive feedback. We have now carefully revised the manuscript in line with the suggestions provided. We have also addressed the major concerns raised and incorporated some new experiments were requested to further strengthen our conclusions. In addition, we have improved multiple aspects of the manuscript's presentation, figures, and analyses based on the reviewers' recommendations.

All revisions and clarifications are detailed point-by-point in our response document. We hope that these comprehensive revisions will be favourably considered and that the manuscript will be considered for publication.

This manuscript (**MS1**) is paired with an additional manuscript **MS2** that explores the role of the AXL-Arf1-Golgi axis in context of mechanoresponsive regulation of the Golgi in MDAMB231 cells.

MS1- AXL receptor tyrosine kinase regulates Golgi organization and function through an adhesion-Arf1 signalling axis in breast and lung cancers (*jcs.263955R1*)

MS2 - Differential AXL expression and regulation of Arf1 control stiffness-dependent Golgi organization in breast cancer cells (*jcs.263956R1*)

These two manuscripts are closely related in how they collectively contribute to a more comprehensive understanding of AXL-mediated regulation of the Golgi. Nevertheless, we have ensured that no data overlap exists, allowing each manuscript to stand independently and be reviewed on its own merit.

In a few instances, we refer to data from MS2 solely to support and contextualize the findings of MS1. We believe this also illustrates why publishing MS1 and MS2 together would be most impactful. Presenting them as back-to-back manuscripts in JCS would not only highlight the coherence of the overall study but also enhance the visibility and impact of the work. We sincerely hope that both papers will be considered favourably for publication in JCS.

Comments from the Reviewers:

Reviewer 1: SUMMARY OF THE ADVANCE MADE IN THIS PAPER AND ITS POTENTIAL SIGNIFICANCE TO THE FIELD

This manuscript presents evidence of a role for the tyrosine kinase Axl in the organization of the Golgi apparatus and the response of this organization to adhesion-mediated signaling. The authors frame this study in the context of cancer, as Axl is often dysregulated in transformed cells. Their data suggest that Axl function is coordinated with that of Arf1, a small GTPase that controls many aspects of Golgi function. This is an interesting and poorly understood connection with potentially important implications for cancer cell biology.

SUGGESTIONS TO AUTHORS Although the authors are commended for testing this Axl/Arf1/Golgi axis in several different cancer cell lines, the results are confusing and sometimes contradictory. The clearest results are observed via comparison of two breast cancer cell lines, MCF7 (Axl-low) and MDA-MB-231 (Axl-high) where the presence of Axl appears to be important for Golgi organization. Things get more confusing when compared to lung cancer lines (A549 and CaLu-1) which exhibit very different responses to Axl inhibition or knockdown, and to loss of adhesion. The authors might be better served to focus solely on the breast cancer lines, in which there is a clearer case to be made for Axl function.

Thank you for your thoughtful review of our manuscript. We agree that the data from MDAMB231 cells are more straightforward to interpret compared to those from A549 cells. Our aim in this study, however, was to address this complexity by demonstrating the adhesion-dependent regulation of AXL and its role in Golgi organization in both MDAMB231 and A549 cells, thereby providing a more comprehensive understanding of this mechanism. We had also hoped that having the A549 cells data will not take away from the clarity that MDAMB231 data has, but only reveal how this regulation could vary between cell types. Our data suggests that in MDAMB231 cells the Adhesion-AXL-Golgi axis could operate in a more canonical adhesion-dependent pathway which can be deregulated in certain cancers such as A549. To address the reviewer's concern regarding confusing outcomes, we outline below the **key findings** of our study which are conserved in

MDAMB231 and A549 and differences that exist.

Similar in both MDAMB231 and A549 cells.

- AXL is differentially expressed in these cells relative to MCF7 and CaLu1 respectively.
- AXL localises to the Golgi in both cell types.
- AXL inhibition / knockdown can disrupt the Golgi in both.
- AXL inhibition mediated Golgi disorganisation needs a drop in Arf1 activation in both.
- AXL inhibition mediated Golgi disorganisation is restored by Q71L -Arf1 in both.
- AXL association with the Golgi lost on its inhibition in both.
- AXL inhibition promotes pY702 AXL phosphorylation in both.
- AXL inhibition affects Golgi function (MDAMB231-MT acetylation/ A549-Acetylation and Glycosylation) in both.

This data in both cell lines supports new information about AXL and its role in regulation of the Golgi. It confirms the Golgi localization of AXL, and how its targeting can affect Golgi organisation. This we find is mediated by an AXL-Arf1 functional crosstalk, which could be supported by their association with each other at the Golgi. This could further affect Golgi-associated functions.

Differences in MDAMB231 and A549 cells

- Adhesion-dependent Golgi organisation is different in both: Organised Golgi in adherent cells is dispersed on loss of adhesion in MDAMB231 but stays intact in A549.
- In adherent cells R428 treatment can inhibit Arf1 activation in MDAMB231 cells but not A549 cells.
- Loss of adhesion mediated regulation of AXL (loss of localization from the Golgi and an increase pY702 AXL) is seen in MDAMB231 cells, but not in A549 cells. This could be the reason for the Golgi to stay organised in A549 cells on loss of adhesion. Therefore, in A549 cells R428 treatment (loss of AXL localization at the Golgi and an increase p702 AXL levels) can cause Golgi disorganisation.

The comparison between **MDAMB231 vs. MCF7** is easier as AXL is almost absent in MCF7. Reconstitution of MCF7 with AXL also becomes a useful way to assess this (**MS2**). In **A549 vs CaLu1** cells the difference in AXL levels is much less making their comparative evaluation harder. The schematics in the revised manuscript is meant to capture these similarities and differences in this pathway between MDAMB231 and A549 cells. The revised discussion now aims to make the overall understanding of the key findings of the paper more clear now.

MDAMB231 cells**A549 cells**
While the data as presented are intriguing, there are many problems with the manuscript in its current form.

These include:

1. Almost of all of the images need higher magnification insets or separate panels to support the conclusions drawn by the authors. Specifically, it is not clear whether Axl localizes to the entire Golgi stack or to individual cisternae. It is impossible to tell at the resolution of the images provided.

We do understand this concern and have now revised our current representation with magnified insets for better visualization of colocalization of AXL and the Golgi markers.

Fig 2E REVISED: Representative zoomed in merged cross-section images to show AXL overlap with GM130 (Cis Golgi), ManII-GFP (Cis/medial) and GalT-RFP (Trans) Golgi markers.

In our original submission, we examined AXL colocalization with markers for different Golgi compartments - GM130 (cis-Golgi), Mannosidase II (medial-Golgi), and GalTase (trans-Golgi). Our confocal imaging data suggests that AXL localizes to each of these three Golgi compartments. To address the concern about resolution, we have now also included higher-magnification insets, representative line plots and Pearson analysis for a subset of images (for MDAMB231) and organized them into individual panels for each cell line (shown above is revised Figure 2E and Fig S2A, B). The revised representation better illustrates the spatial distribution of AXL in the different Golgi compartments suggesting that AXL localizes throughout the Golgi stacks, and is not restricted to any specific cisterna. We have now also revised all the other figures and manuscript

with zoomed magnification insets and colocalization data including Pearson's correlation analysis in Fig 4F, 4J, S4F, 5F, S5C, and 7I.

2. A related point - line scans are a poor substitute for unbiased colocalization analysis (Pearson's and/or Manders' coefficients). While Pearson's coefficients are shown for some sets of images, they should be used throughout.

We appreciate the reviewer's suggestion. In our original submission, we used Pearson's correlation coefficients to quantify the colocalization of AXL with cis-Golgi marker GM130 in the presence and absence of R428 and to compare adherent vs non-adherent cells. This specific analysis was included because AXL localization at the Golgi is being quantitatively reported ifor the first time. Analysis for all subsequent data for AXL localization changes were done using line intensity plots rather than Pearson's coefficient of correlation. We agree with the reviewer that adding Pearson's correlation analysis for all colocalization data will maintain uniformity across datasets and would strengthen our conclusions.

As mentioned in the earlier response, we have now revised all the figures in the manuscript with colocalization data with Pearson's correlation analysis in Fig 4F, 4J, S4F, 5F, S5C, and 7I. An example of changes made is shown in the new version of Fig S2 and Fig 4F Revised (shown below).

Fig REVISSED (Fig S2): Representative zoomed in merged images and line plots to show AXL overlap with ManII-GFP (Cis/medial) and GalT-RFP (Trans) Golgi markers. Pearson correlation analysis for 10 (n) cells for AXL and ManII- GFP and GalT-RFP markers respectively.

Fig 4F REVISSED: Representative images with line plots and Pearson analysis plot to show AXL-Active Arf1 (ABD- GFP) colocalization between DMSO and R428 treated MDAMB231 cells.

3. A higher N is needed for all of the imaging. 4 cells is simply not enough to draw statistically significant conclusions.

We thank the reviewer's feedback regarding our confocal imaging data. We have now updated some of the figures with further biological replicates. Additionally, for figures where only line plots were shown previously, we have now also included Pearson's correlation coefficients based on a greater number of cells, as mentioned in our earlier response. ABD-GFP experiments in supplementary figures (**Fig. S4F, S5C**) were the experiments, we had trouble adding more replicates when studying their colocalization with an additional marker as the transfection and getting optimal expression of this construct was a challenge in our cell lines of interest. We have now successfully added more experiments along with quantitative analysis to support our observations and are included in the revised manuscript.

4. Presumably most of the Axl staining is of endogenous protein? This is never clearly stated.

Yes, it is rightly pointed out that all data pertaining to AXL levels and localization in the manuscript is collected from cells with endogenous AXL expression. This clarification is now made in the revised version of our manuscript (**Page No. 6 and 9**).

5. How does inhibition of Axl activity lead to its increased phosphorylation (Fig. 3D)?

We have grappled with the question ourselves and hence have added multiple experiments that represent the effect AXL inhibitor R428, has on its phosphorylation. The data in published literature had shown mixed outcomes for the effect R428 mediated AXL inhibition has on AXL phosphorylation (pY702 AXL). Chen et al., *Am J Cancer Res.*, 2018 reported an initial decrease in pAXL (Y702) upon R428 treatment, followed by a time- dependent **increase** in phosphorylation over 24 hours. Other studies, have shown AXL phosphorylation to decrease on R428 treatment (Holland et al. 2010; Woo et al. 2019).

In this study, we consistently observe an increase in AXL phosphorylation upon R428 treatment for 12 hours (**Fig 3D, 6E and 6F**). This is also reflected in a gradual time dependent increase in pAXL levels in MDAMB231 cells on R428 treatment further supporting the change in phosphorylation observed (**Fig S3A**). Loss in the detection of pAXL (Y702) in AXL knockdown MDAMB231 cells and MCF7 cells (lacking endogenous AXL expression) further confirms the specificity of pAXL (Y702) antibody detection.

Along with the effects on pAXL seen following R428 treatment, we see inhibition of Arf1 (by GCA) to increase pAXL (Y702) levels (**Fig 4I**). Both are accompanied by Golgi disorganisation. Similarly, loss of adhesion mediated disorganisation of the Golgi in MDAMB231 cells is accompanied by a drop in Arf1 activation (**Fig 5C**), and an increase in pAXL (Y702) levels (possibly reflecting AXL inhibition). This increase in AXL phosphorylation if reflecting in its inhibition, could in principle agree with our hypothesis that inhibition of both AXL and Arf1 drive Golgi disorganization.

Despite these observations, we recognize that the functional consequence of a change in AXL phosphorylation remains unclear. We have now revised the manuscript carefully to describe this observation as a treatment-dependent change in phosphorylation, which could possibly correlate with changes in AXL activation status, and cite studies (e.g., Chen et al., *Am J Cancer Res.*, 2018) where similar phosphorylation changes have been observed.

6. The authors note that Akt is downstream of Axl. What happens to Golgi organization if Akt is inhibited?

AXL is a known upstream regulator of the PI3K/Akt pathway, with activation of Akt typically detected via phosphorylation at the Ser473 residue (Holland et al., *Cancer Biol Ther.*, 2010). Consistent with this, we observe that treatment with R428 results in a decrease in pAkt levels (**Fig 3C**). However, our data reveal a critical distinction between AXL's regulation of Akt and its role in Golgi organization. Specifically, qualitative time- course analysis of R428 (1 μ M) treated MDAMB231 cells shows Golgi disorganization to change steadily over time (**Fig. S3C**).

Fig S3B and C: Time course of R428 treatment in MDAMB231 and its effect on Golgi organization and Akt activation.

In contrast, pAkt levels drop rapidly and remain low across all these time points (**Fig. S3B**), indicating that Akt activity is suppressed early and persistently. This temporal disconnect between Akt inhibition and the onset of Golgi disorganization suggests that AXL's effect on Golgi organization is not mediated through the Akt pathway.

To experimentally address if Akt is directly involved in regulating AXL mediated Golgi organisation, we have now performed additional experiments using 1 μ M of Akt Inhibitor VIII in MDAMB231 cells. This caused a robust reduction in pAkt (Ser473) levels, comparable to that observed with R428, confirming effective inhibition of Akt activation. Despite this, no significant change in Golgi organization was observed in MDAMB231 cells. This suggests that changes in Akt activity do not regulate Golgi organisation in MDAMB231 cells and hence does not contribute to AXL-mediated regulation of the Golgi (**Shown below in Fig**).

As we mentioned at the start, this study and manuscript is accompanied by a manuscript in revision in JCS- MS2 - **Differential AXL expression and regulation of Arf1 controls matrix stiffness-dependent Golgi organization in breast cancer cells**. This Akt inhibition experiments were done for the revision of MS2. It is also one of the reasons, we would like MS1 and MS2 to both be considered together for JCS. The impact of having these two studies presented with each other we feel will help make a comprehensive case for both.

Fig (MS2): Effect of Akt VIII (1 μM) mediated inhibition of Akt on Golgi organization profile of MDAMB231 cells.

7. The relationship between Axl and Arf1 is confusing. A previous study reported that inhibition of Axl enhances its association with Arf1 and increases Arf1 activity. Here using the same cell line (MDA-MB-231), Axl inhibition apparently reduces Arf1 activation (though not by much) while retaining the ability of the two proteins to co-precipitate. What is the evidence that Arf1 interaction with Axl is GTP-dependent?

The one previously published work looking at this crosstalk has indeed suggested that AXL targeting using R428 promotes Arf1 activation (Haines et al., *Cancer Biol Ther.*, 2015). This was one of the motivations for us to examine the AXL-Arf1 crosstalk in greater detail in this study and check its downstream implications on Golgi organisation.

In our experiments, R428 treatment of MDAMB231 cells consistently resulted in a modest but significant reduction in Arf1 activity (Fig 4C). This finding was independently validated using siRNA-mediated AXL knockdowns, which similarly led to a ~30-40% decrease in Arf1 activation (Fig 4E). The drop in Arf1 activation was consistent and is supported by the regulation of its spatial localization at the Golgi (Fig 4F & S4F). While this change was contrary to what was reported earlier, we now have multiple evidences to support our claim.

One of the prominent pieces of evidence is the fact that expression of constitutively active Arf1 in R428 treated cells restored Golgi organisation in MDAMB231 (Fig 4G) and A549 (Fig 7H). Without an AXL targeting mediated drop in Arf1 activation, the expression of a constitutively active Arf1 will not impact Golgi organisation this way. We have now included a brief discussion of this outcome explaining why it supports our claim (Page 13).

Adding to the above claims is also the fact that on loss of adhesion MDAMB231 cells show a drop in Arf1 activation (Fig 5A) accompanied by the disorganisation of the Golgi (Fig 5E). This is accompanied by an increase on pAXL (Y702) (Fig 5C). We understand the increase in AXL phosphorylation on its inhibition, in MDAMB231 cells, remains corelative in this study. But if considered this also supports our data that increase in pAXL corelates with Arf1 inhibition.

Supplementing the above observations we also wanted to share data from **MS2 - Differential AXL expression and regulation of Arf1 controls matrix stiffness- dependent Golgi organization and function in breast cancer cells**. As part of this study, we have evaluated the basal activation of Arf1 in MCF7 (lacking AXL) and AXL- MCF7 cells (stably reconstituted with AXL). This piece of data (shown below) reveal expression of AXL in MCF7 cells enhances Arf1 activation (at 23 kPa and Glass). They suggest AXL expression regulates Arf1 activation in these cells (**MS2**).

Fig (MS2): Basal Arf1 activation comparison of WT-MCF7 and AXL-MCF7 cells plated on 23 kPa PA gel and Glass (Both coated with 25 ug/ml collagen). AXL detected in GGA3 pull down fractions of AXL-MCF7 cells and not in WT- MCF7 cells (with no endogenous AXL).

QUERY: What is the evidence that Arf1 interaction with Axl is GTP-dependent?

Across all GST-GGA3 pulldown experiments- where only active Arf1 is selectively captured- we consistently detect AXL in the pulldown fractions (**Fig. 4D, 5B, 7C**). While this strongly suggests that AXL associates with active Arf1 (directly or indirectly), we agree that this does not by itself establish GTP-dependence of this interaction. To test whether AXL binding is specific to Arf1, we first performed siRNA-mediated Arf1 knockdown followed by GST-GGA3 pulldown. Arf1 depletion markedly reduced the amount of total Arf1 levels in cell lysates without affecting AXL levels. This reflects in a loss of active Arf1 in GST-GGA3 pulldowns for siRNA knockdown lysates. Importantly, AXL levels in the pulldown fractions dropped on Arf1 depletion, indicating that the presence of Arf1 is essential for this interaction observed in GST-GGA3 pull down fractions. A small residual AXL signal in Arf1 knockdown pulldowns likely reflects low- level nonspecific binding to GST-GGA3 beads. Together, these results support an Arf1- dependent interaction of AXL in the pulldown fractions, consistent with their interaction being enriched in the active Arf1 pool.

FigS4C NEW: Representative immunoblots for Arf1 and AXL detection in GST-GGA3 pull down and whole cell lysate fractions of Control and Arf1 knockdown MDAMB231 cells.

To assess if this association with Arf1 is indeed regulated by its activation, MDAMB231 lysates were treated with 400 μ M GTP γ S to lock Arf1 in its active conformation. Despite a robust increase in Arf1 activation as seen in the GST-GGA3 pulldown, AXL levels in pulldown fractions remained comparable to control (Fig. S4D).

This could suggest that the association between AXL and endogenous active Arf1 is likely influenced by their spatial localizations within cells. Although GTP γ S treatment efficiently locks Arf1 in its active (GTP-loaded state), this activation occurs in lysates and may not preserve the subcellular context required for AXL-Arf1 association. It is also possible that constitutively locking Arf1 in the GTP γ S-bound conformation disrupts its GDP-GTP cycling, which may be required for AXL engagement. Additionally, AXL's own activation status (detected by its Y702 phosphorylation) may influence its ability to bind active Arf1. If limiting in cells, this could affect AXL interaction with the GTP γ S-loaded Arf1 pool.

Together, these considerations suggest that the AXL-Arf1 association detected in GST-GGA3 pulldowns reflects a spatially constrained and possibly a tightly regulated basal interaction that is hence not affected global GTP γ S activation of Arf1 in lysates.

Fig S4D NEW: Effect of GTP-gamma-S treatment in MDAMB231 lysates on AXL binding observed using GGA3 pull down assay.

The functional crosstalk between AXL and Arf1 is further underscored by the observation that constitutively active Arf1 (Q71L), but not wild-type Arf1, robustly rescues the Golgi disorganization induced by either R428 treatment or loss of adhesion in these cell lines (Fig. 4G; Fig. 5G, MDAMB231; Fig. 7H, A549). This ability of Arf1-Q71L to bypass AXL-dependent effects, highlights Arf1 as a key regulatory player in AXL-dependent -regulation of Golgi organization.

8. Although the Axl inhibitor R428 appears to cause its dissociation from the Golgi (Fig. 4A), it is still punctate. Where is it? Because it is a transmembrane protein it must be associated with other intracellular membranes, but what are they?

Figure provided in confidence to reviewers has been removed.

9. The authors introduce the GGA pulldown to measure Arf activation but do not describe it. This would be useful for investigators unfamiliar with Arf biology.

While the GGA3 pulldown assay has been described in detail in our previous publications (Singh et al., JCS, 2018; Rajeshwari et al., *Bio Open*, 2023), we recognize that a summary would be useful for readers unfamiliar with this method. We have now added a description of the same in the methods section (Page 20).

Briefly, Arf1 activation was measured using a GGA3 pulldown assay that selectively captures the GTP-bound (active) form of Arf1. This assay relies on the use of GST-GGA3 fusion protein bound to glutathione-conjugated Sepharose beads. GGA3 (Golgi-localized Gamma-adaptin ear homology domain) ARF-binding protein functions as an effector of Arf1-GTP, and bait to pulldown active Arf1 from cell lysates. When normalised to Arf1 levels in whole cell lysates this provides an effective

way of determining what fraction of Arf1 is active in cells.

10. The authors claim that R428 reduces Axl colocalization with Arf1 in adherent MB-231 cells (Fig. 4F) but the images shown are not convincing. And Pearson's would be required to quantify any loss of colocalization.

We agree that the inclusion of quantitative colocalization analysis would significantly strengthen our interpretation. In response, we have revised the relevant figure to include Pearson's correlation coefficients quantifying the degree of colocalization between endogenous AXL and active Arf1 (ABD-GFP). To ensure robustness, this analysis was performed using Huygens Professional software on approximately 30-35 deconvoluted z-stack images per condition, across three independent biological replicates of control and R428-treated MDAMB231 cells. The results demonstrate a statistically significant reduction in AXL-active Arf1 colocalization upon R428 treatment, supporting our claim that AXL activity is critical for maintaining Arf1 activation and localization to Golgi. We have updated the manuscript accordingly to include these quantitative findings and clearly include number of cells analysed (**Fig 4F legends - Page No. 28**).

Fig 4F REVISDED: Representative merged & zoomed in images and Pearson correlation coefficient analysis plot to show AXL-Active Arf1 (ABD-GFP) overlap between DMSO and R428 treated MDAMB231 cells.

11. The authors suggest that phosphorylation of Axl at Y702 is inhibitory. What is the evidence for this?

We agree that the role of AXL phosphorylation at Y702 site remains unclear, and current literature offers inconsistent interpretations, as discussed earlier in our **response to comment #5**. We have now clarified in the revised manuscript how the data presented here suggests increased phospho-AXL (Y702) to correlate with AXL inhibition (on R428 treatment or loss of adhesion), but the mechanistic understanding of how this happens remains unknown (**Page No. 7**).

In this study, we consistently observe an increase in AXL phosphorylation upon R428 treatment for 12 hours in MDAMB231 (**Fig 3D**) and A549 (**Fig 6E**). R428 treatment also causes a drop in pAKT levels (**Fig 3C**), known to be a direct readout of AXL inhibition (Holland et al., *Cancer Res.*, 2010). Similarly, R428 treatment of MDAMB231 cells over time, causes gradual time dependent increase in pAXL (Y702) levels supporting the change in phosphorylation to strongly correlated with AXL inhibition (**Fig S3A**). In MDAMB231 cells detection of pAXL (Y702) is lost in AXL knockdown cells and MCF7 cells (lacking AXL expression) confirming the specificity of the antibody (**Fig S3D**). We have now revised the manuscript carefully to describe these observations and the correlations they help us make.

12. Fig. 5A shows that Arf1 activity is reduced in suspended MB-231 cells, yet the amount of Axl that coprecipitates with it is unchanged (Fig. 5B). How do the authors explain this?

We thank the reviewer for this insightful comment. In MDAMB231 cells, loss of adhesion leads to a substantial decrease in active Arf1 levels (**Fig. 5A**). However, the amount of AXL detected in the GGA3-GST pulldown, reflecting the fraction of AXL associated with active Arf1 pool in suspended cells, remains largely unchanged (**Fig. 5B**). This suggests that, upon loss of adhesion, the remaining active Arf1 pool in suspended cells may exhibit a higher association with AXL, potentially due to redistribution of Arf1 and/or AXL from the disorganized Golgi into membrane locations where their interaction might be favoured.

Fig 5A and B: GGA3 pull down to detect active Arf1 and AXL in SA vs SUS MDAMB231 cells

Additionally, an increase in AXL phosphorylation at its Y702 residue in suspended cells reflects a change in overall activation status of AXL. This may also influence its ability to show an enhanced association with active Arf1. The contribution of this phosphorylation to AXL-Arf1 interaction is not known. Loss of adhesion dependent changes in AXL activation, together with a reduction in Arf1 activation could affect its localization. This could allow for AXL binding to Arf1 to be enhanced. This could further cause levels of AXL in the GGA3 pulldown of Arf1 in suspended cells to be maintained despite an overall drop in Arf1 activity. This suggests loss of adhesion could alter the spatial localization and activation states of both AXL and Arf1 in a way that preserves their relative association with the active Arf1-bound pool.

13. There is also a drop in Akt activity upon suspension of MB-231 cells (**Fig. 5D**) but this could be due to a loss of more proximal adhesion-dependent signalling (e.g. FAK).

We agree with the reviewer that loss of adhesion likely impacts multiple upstream adhesion-dependent pathways such as FAK, contributing to the observed drop in Akt activation. AXL inhibition by R428 causes only upto a ~50% drop in Akt activation in these studies (**Fig 3C**). This when compared to loss of adhesion mediated drop in Akt activation shows the drop on loss of adhesion to indeed be more prominent (**Fig 5D**). This could reflect a combined effect of AXL dependent regulation of Akt activation and AXL-independent but adhesion-dependent regulation of Akt activation. It is worth noting that adhesion likely also regulates AXL (evidenced by changes in pY702 AXL levels), allowing it to contribute to this regulation of Akt. We have now added this clarification in the revised manuscript. (**Page No. 9-10 and line 28-31**)

14. The authors claim that loss of adhesion does not affect Axl phosphorylation (Y702) in A549 cells (**Fig. 7D**), but the blot shown certainly suggests that it does, and to a similar extent as Axl inhibition with R428.

We acknowledge the reviewer's observation of our western blot images in **Fig 7D** where on loss of adhesion there does seem to be an increase in pAXL levels. However, this change was very variable in these experiments as is reflected in the spread of data for N=5 experiments done. We chose the representative blot poorly and have now replaced the same with a blot that is more reflective of the mean distribution of the data in these cells. The revised data representation has

now been included in the revised manuscript.

Fig 7D REVISED: Representative western blots for Y702 phosphorylated AXL (pAXL) and AXL in cell lysates of stable adherent (SA) vs non-adherent (SUS) A549 cells

15. In their discussion the authors state that Axl inhibition leads to loss of both Axl and Arf1 from the Golgi in both MB-231 and A549 cells, yet Arf1 activation is only affected in MB-231. This doesn't seem possible, as Arf1 activity requires its association with membranes. How do the authors explain this?

AXL inhibitor, R428 treatment leads to loss of both AXL and Arf1 from the Golgi in **adherent** MDAMB231 cells and **only in non-adherent** A549 cells, where loss-of- adhesion (independent of change in AXL activation status) has caused a drop in Arf1 activation. This AXL- mediated disorganisation of the Golgi hence happens (in MDAMB231 cells or A549 cells) when there is a drop in Arf1 activation observed. This we think is dependent on loss of AXL from the Golgi, which could be regulated by its activation (reflected in pY702 AXL levels).

The lack of visible displacement of active Arf1 (detected by ABD-GFP/RFP) from the Golgi upon loss of adhesion in A549 cells could be explained by two possibilities: 1) a portion of the Golgi-localized pool of active Arf1 is indeed retained, even though total active Arf1 levels significantly decrease under loss of adhesion; and/or 2) detection of changes in Golgi-associated active Arf1 using ABD-GFP may be limited by the nature of this overexpression system, such that only substantial changes at the Golgi would be detectable. It is also noteworthy that when active Arf1 (detected by ABD-GFP/RFP) is retained at the Golgi, the Golgi organization remains intact in both MDAMB231 and A549 cells. We have now revised the manuscript carefully to discuss these observations in the discussion section (**Page. 14**)

Reviewer 2:

SUMMARY OF THE ADVANCE MADE IN THIS PAPER AND ITS POTENTIAL SIGNIFICANCE TO THE FIELD

In this manuscript, Prachi Joshi, Arnav Saha, and colleagues set out to characterize proteins that regulate Golgi morphology of cancer cells, in a manner dependent on their adhesion status. After carrying out a literature analysis, they chose AXL, a plasma membrane (PM) receptor tyrosine kinase, for their study. They show that AXL localizes to the Golgi, and claim that AXL interacts at the Golgi with Arf1-GTP (but see point 4 below). The authors show that there is a decrease in phosphorylation of Akt and an increase in phosphorylation of AXL upon treatment of MDAMB231 cancer cells with the AXL inhibitor R428. They show that the Golgi is disorganized upon treatment of these cells with R428, but that MCF7 cells, which do not express AXL, fail to undergo a change in Golgi organization upon R428 treatment. These and other experiments lead the authors to conclude that upon loss of adhesion, AXL is regulating Golgi morphology through interaction with Golgi-localized Arf1-GTP. However, the authors have only shown correlations between perturbations of AXL localization to the Golgi and Golgi morphology changes. The experiments to address mechanisms, by assaying Arf1 activation status upon various conditions, do not

demonstrate that it is the Golgi pool of Arf1 that is involved in the interaction.

It has already been demonstrated that Arf1 is recruited to the PM upon EGFR signalling and interacts with AXL (Haines et al. 2015 *Cancer Biol Ther.* 16(10):1535-1547 Boulay et al. 2008 *J Biol Chem.* 283(52):36425-34). Indeed, previous work has shown that a number of PM signalling pathways affect Golgi structure and function (Chia et al. 2012. *Mol Syst Biol.* 8:629). Here, insufficient data is presented to determine whether the effects seen on the Golgi in this manuscript are indirect effects of previously demonstrated mechanisms at the PM. In addition, there are some critical control experiments that are missing. Because of these limitations, the data presented remain descriptive and do not clearly show a new mechanism by which AXL regulates Golgi structure and function.

Thanks to the reviewer for the insights on our manuscript. We would like to address each of the concerns raised by the reviewer.

We acknowledge that Arf1 can localize to both the plasma membrane (PM) and the Golgi; however, in mammalian cells, the Golgi localized pool of Arf1 is the predominant functional fraction, especially in the context of membrane trafficking and Golgi organization. The PM-associated Arf1 pool is smaller and is primarily implicated in endocytosis and actin cytoskeleton remodelling (Caviston, J. P., et al., *Cytoskeleton*, 2014; Kumari, S., et al., *Nat Cell Biol.*, 2008). While this PM pool could possibly be trafficked to and from the Golgi, the implications such a traffic could have in Golgi organisation remains unclear. The PM-associated Arf1 pool implicated in endocytosis and cytoskeletal remodelling could hence support downstream signalling to the Golgi. The spatial localization of active Arf1 at the Golgi and its regulation by Arf1 GEFs is known to affect Golgi organisation. Our studies identifying the adhesion-dependent regulation of Golgi has shown the spatial localization of ARF GEFs and their differential regulation of Arf1 activation to mediate loss of adhesion mediated Golgi disorganisation and re- adhesion mediated re-organisation (Singh V. et al., *JCS*, 2018; Rajeshwari, B. R., et al., *Bio Open*, 2023)

In this study, we began with assessing total cellular Arf1-GTP levels, knowing that that this could significantly come from and reflect activity at the Golgi. We further show differences (across different conditions) in the localization of active Arf1 in cells using ABD-GFP or RFP which selectively bind to the GTP-bound active Arf1. This has not just allowed us to document the distinct localization of active Arf1 at the Golgi in these cells but also allowed us to quantify its regulation at the Golgi. Together, they strengthen our claims that Golgi-associated Arf1 is involved in AXL-dependent regulation of Golgi organisation and function.

We have also performed quantitative co-localization analysis for endogenous immunostained AXL and active Arf1 (visualized via ABD-GFP) in MDAMB231 cells. This data shows AXL and active Arf1 to predominantly co-localize at a perinuclear region that overlaps with a compact Golgi (confirmed by Golgi marker staining in parallel experiments). Importantly, we observe minimal co-localization of AXL and active Arf1 in other intracellular regions or at the plasma membrane under these conditions (**Fig 4F Revised**). Upon R428 treatment, this distinct perinuclear overlap between AXL and Arf1 is significantly diminished (**Fig 4F Revised**). Both AXL and active Arf1 now display a scattered distribution as smaller punctate structures throughout the cytoplasm, which correlates with the observed Golgi dispersal phenotype (**Fig 4F Revised**).

AXL, as we have shared in this response earlier, on being displaced from the Golgi upon R428 treatment shows a clear colocalization with mCherry-LC3, a fraction that marks membrane bound - autolysosomes. We have not included these findings in our submission as there is indeed more to understanding the autophagy-Golgi connection that we are exploring in an independent project.

Fig 4F REVISITED: Representative merged and zoomed in surface rendered colocalization images for AXL-Active Arf1 (ABD-GFP) overlap between DMSO and R428 treated MDAMB231 cells.

All, these results argue strongly against the notion that the effects on Golgi organization are indirect consequences of AXL-Arf1 interactions at the plasma membrane. Rather, they support a model in which AXL and active Arf1 spatially localize at the Golgi (may possibly also interact) to regulate its structural integrity. The presence of a functional crosstalk between AXL-Arf1 in these studies is supported by GST-GGA3 mediated pulldown of GTP-bound active Arf1 detecting a fraction of AXL being enriched. As discussed earlier this association needs to be better evaluated, but does suggest their ability to work with each other. Additional experiments with GolgicideA (inhibitor of Arf1- GEF - GBF1), as well as rescue of Golgi organization in R428 treated cells by constitutively active Arf1 (Arf1-Q71L), also supports an AXL-Arf1 functional crosstalk. Finally, the companion manuscript to this study (**MS2**), in revision in JCS, interestingly shows matrix stiffness dependent regulation of AXL and Arf1 expression levels. Stable AXL expression in MCF7 cells restores Golgi organization and active Arf1 localization at the Golgi (shown qualitatively), reinforcing our proposed hypothesis of how the AXL-Arf1 crosstalk is regulated to regulate Golgi organization.

Supp Fig (MS2): Representative images and colocalization analysis for Active Arf1 (ABD-GFP) and GM130 overlap between WT-MCF7 and AXL-MCF7 cells.

SUGGESTIONS TO AUTHORS

1. In Figure 1, four different cancer cell lines, transfected with a cis-medial Golgi marker (ManII-GFP) and a trans-Golgi marker (GalTase-RFP), were assayed for Golgi structure. Two of these cell lines showed compact Golgi morphology (called "organized"), and two showed more dispersed Golgi morphology (called "disorganized"). Adherent or non-adherent cells for each cancer cell line were then examined for Golgi organization. For three of the cell lines, both adherent and non-adherent cells had similar levels of Golgi organization. In the fourth cell line, MDAMB231 breast cancer cells, the authors conclude that nonadherent cells had disorganized Golgi and adherent

cells had compact Golgi, stating « In non-adherent MDAMB231 cells the Golgi is dispersed (Fig 1B) as seen in the non-transformed breast epithelial MCF10A cells. ». This result is clear according to the quantifications shown, but the adjacent images show the opposite result (Fig. 1B). Whether this discrepancy is due to mislabelling of "SA" and "SUS" images or not must be addressed. In Fig. 1C as well, SA and SUS labels are reversed compared to the adjacent quantifications.

We thank the reviewer for pointing out this discrepancy in the Golgi morphology panels. Upon review, we agree that the representative images shown in Fig 1B for MDAMB231 cells were inadvertently mislabelled, and the "SA" and "SUS" panels were interchanged during figure assembly. This mislabelling also carried over to the corresponding labels in Fig 1C. We have corrected the image labels in the revised version of the figures and will update this in the manuscript. This now accurately reflects the quantification and the corresponding imaging results, showing that non-adherent MDAMB231 cells have disorganized Golgi, while adherent MDAMB231 cells exhibit compact Golgi morphology. We sincerely apologize for this oversight.

Fig 1B and 1C Revised: Representative MIP and SR images for predominant Golgi phenotype of adherent (SA) and non-adherent (SUS) (B) MDAMB231 and (C) MCF7 cells expressing ManII-GFP (Cis/medial) and GalTase-RFP (Trans) Golgi markers. Graph shows the percentage distribution profile for cells showing organized (white) and disorganized (gray) Golgi in SA vs SUS cells, respectively

2. In the "in silico" analysis shown in Fig. 2, were any proteins whose major function is at the Golgi identified in the pipeline shown in Fig. 2A?

Yes, since our primary listing was derived by identifying genes associated with Golgi structure and function, several genes that have major functions or subcellular localization at the Golgi were indeed part of our *in-silico* analysis. However, due to relatively smaller fold changes in expression between the two cancer cell lines evaluated or lower combined scores in our analysis pipeline, they did not make the cut as candidates for this study.

Some examples of such genes that localize to the Golgi and are directly known to regulate Golgi structure and function are given below. Since in both breast and lung cancer models, none of them had a fold change difference in gene expression of ≥ 5 , they were not considered further. The revised manuscript now mentions about some of these known Golgi associated genes (Page No. 6).

GENE NAME (NCBI)	Breast cancer (Fold change)	Lung cancer (Fold change)
GOLGA2 (GM130)	2.33113166 (MCF7>MDA)	1.636297376 (CaLu1>A549)
ARF1	1.011320755 (MDA>MCF7)	1.45797227 (CaLu1>A549)
AURKA	1.151721207 (MDA>MCF7)	1.15234375 (CaLu1>A549)
GORASP1 (GRASP65)	1.052786499 (MCF7>MDA)	2.037190083 (CaLu1>A549)
GORASP2 (GRASP55)	1.17170586 (MCF7>MDA)	1.207396664 (CaLu1>A549)

Table - List of known Golgi-associated genes picked up in the in-silico analysis

3. It is necessary for the authors to carry out a rescue experiment in MCF7 cells in order to determine whether the phenotypes reported are due to lack of expression of AXL or some other difference between these cells and those expressing AXL, such as MDAMB231 cells.

We also recognize the importance of this point. In a parallel study presented in an accompanying manuscript (MS2), we investigated the regulation of Golgi organization by AXL-Arf1 crosstalk under matrix stiffness conditions in MDAMB231 versus MCF7 cells. In this work, stable reconstitution of AXL in MCF7 cells partially restored Golgi organization, reflected by a significant reduction in Golgi object numbers and a profile more closely resembling that of MDAMB231 cells.

Importantly, AXL expression in MCF7 cells (AXL-MCF7) also increased Arf1 activity compared with parental MCF7 cells, further supporting our conclusion that AXL expression/activation regulates Arf1 to drive Golgi organization.

Fig (MS2): AXL expression in MCF7 cells restores compact/organized Golgi morphology in MCF7 cells. Quantification to show Golgi object count comparisons and Percentage distribution profile of cells with organized and disorganized Golgi in AXL-MCF7 vs. WT-MCF7 cells

4. In Fig. 4C, the effect of R428 on the level of Arf1-GTP in MDAMB231 cells, as monitored by a GGA3 pull-down assay, is very small. The variability in the control levels of Arf1-GTP is masked by normalizing the levels in the R428-treated samples to the control condition. Why are Arf1-GTP levels normalized in this way when such normalization is not carried out in other cases, such as in Fig. 4D? What concentration of R428 was used in this experiment?

We agree with the reviewer that normalization can sometimes mask biological variation. However, in the case of Arf1-GTP pulldown assays, normalization is generally necessary due to

the inherently small fraction of endogenous Arf1 that is active in cells and the variability associated with it. We also acknowledge that the accompanying AXL pull-down data should have been presented normalized to the respective controls. We have now made this change consistently across the manuscript, as shown in the revised figures.

Fig 4D Revised: GGA3 pull down to compare AXL levels in DMSO (control) vs R428 (1 μ M) treated MDAMB231 cells. The graphs for AXL (in GGA3-PD)/Total AXL levels are shown as non-normalized (Old) and normalized relative to DMSO (Revised).

Regarding the use of R428 in these experiments, adherent MDAMB231 cells were treated with **1 μ M R428** or DMSO (control) for 12 hours prior to lysis and the GGA3 pulldown assay. We recognize that this methodological detail was missing in the original figure legends and will ensure that it is now clearly mentioned in the revised version (**Page No. 27 Fig. 4D legends**). We thank the reviewer for bringing this to our attention.

5. Page 16, lines 29-31. The conclusion that "GGA3-GST pulldown of active Arf1 also brings down AXL with it; this association unaffected by R428 treatment (Fig 4D)" requires further controls, such as experiments to determine whether AXL binds to beads alone or to GGA3 without Arf1.

We appreciate this concern and have performed appropriate bead-only control experiments to address it. We carried out pulldowns using GST-only beads, and GST- GGA3 beads with whole-cell lysates from MDAMB231 cells. Representative immunoblots from these controls show that Arf1-GTP is pulled down in GST-GGA3 bound beads, but not with GST-only beads. AXL showed enhanced detection in GST- GGA3 bound bead pulldowns over GST-only beads.

Fig S4B NEW: Representative immunoblots for Arf1 and AXL detection in GST only beads and GST-GGA3 pull down and whole cell lysate fractions

We also performed siRNA-mediated Arf1 knockdown followed by GST-GGA3 pulldown. siRNA treatment caused a robust total Arf1 depletion without affecting total AXL levels (**Fig. S4C**). This further reflects in a distinct drop in active Arf1 pulled down by GST-GGA3 beads from knockdown lysates. This also affects AXL levels being brought down in the GST-GGA3 pulldown fraction (**Fig.**

S4C) suggesting the presence of Arf1 to be vital for this association.

Fig S4C NEW: Representative immunoblots for Arf1 and AXL detection in GST-GGA3 pull down and whole cell lysate fractions of Control and Arf1 knockdown MDAMB231 cells.

Some residual AXL binding to GST-GGA3 beads incubated with Arf1 siRNA knockdown lysates, suggests some of this binding could be nonspecific. Together they suggest AXL binding in these pulldowns is Arf1-dependent. Considering the binding of GST-GGA3 is specific for active Arf1, it does suggest this association could be dependent on Arf1 activation.

6. In the Abstract, Page 2 lines 16-19, the sentence "AXL prominently localized at the Golgi, undergoes displacement from the Golgi when inhibited by R428 and knocked down using siRNA, causing the Golgi to disorganize" is hard to understand. The authors are stating that AXL undergoes displacement from the Golgi when knocked down by AXL siRNA treatment, which is of course the case because AXL levels are drastically reduced. What exactly is "causing the Golgi to disorganize" in the statement above is not clear.

We thank the reviewer for pointing out the ambiguity in our original abstract. We agree that the previous sentence does not clearly distinguish the effects from pharmacological inhibition (with R428) and genetic knockdown of AXL. We have now revised the sentence to convey our findings clearly:

"AXL, was seen to prominently localize at the Golgi, and is displaced from the Golgi upon inhibitor treatment (R428), causing Golgi to disorganize. Similarly, siRNA-mediated knockdown of AXL also led to Golgi disorganization, supporting the Golgi-localized AXL to play a vital role in maintaining Golgi organization"

We believe this revised statement in the abstract (**Page No. 6**), now clarifies our finding that Golgi disorganization is observed when AXL is displaced from the Golgi, either by treatment with R428 or by genetic knockdown of AXL.

References

1. Chen, F., Song, Q., & Yu, Q. (2018). Axl inhibitor R428 induces apoptosis of cancer cells by blocking lysosomal acidification and recycling independent of Axl inhibition. *American Journal of Cancer Research*, 8(8), 1466-1482.
2. Rajeshwari, B. R., Shah, N., Joshi, P., Madhusudan, M. S., & Balasubramanian, N. (2023). Kinetics of Arf1 inactivation regulates Golgi organisation and function in non-adherent fibroblasts. *Biology Open*, 12(4), bio059669.
3. Caviston, J. P., Cohen, L. A., & Donaldson, J. G. (2014). Arf1 and Arf6 promote ventral actin structures formed by acute activation of protein kinase C and Src. *Cytoskeleton*, 71(6), 380-394.
4. Kumari, S., & Mayor, S. (2008). ARF1 is directly involved in dynamin-independent endocytosis. *Nature Cell Biology*, 10(1), 30-41.
5. Holland, S. J., Pan, A., Franci, C., Hu, Y., Chang, B., et al. (2010). R428, a selective small molecule inhibitor of Axl kinase, blocks tumor spread and prolongs survival in models of metastatic breast cancer. *Cancer Research*, 70(4), 1544-1554.
6. Haines, E., Schlienger, S., & Claing, A. (2015). The small GTPase ADP-ribosylation factor 1 mediates the sensitivity of triple-negative breast cancer cells to EGFR tyrosine kinase

- inhibitors. *Cancer Biology & Therapy*, 16(10), 1535-1547.
7. Singh, V., Erady, C., & Balasubramanian, N. (2018). Cell-matrix adhesion controls Golgi organization and function through Arf1 activation in anchorage-dependent cells. *Journal of Cell Science*, 131(16), jcs215855.
 8. Woo, S. M., Min, K., Seo, S. U., Kim, S., Kubatka, P., Park, J.-W., & Kwon, T. K. (2019). Axl inhibitor R428 enhances TRAIL-mediated apoptosis through downregulation of c-FLIP and survivin expression in renal carcinoma. *International Journal of Molecular Sciences*, 20(13), 3253.
-

Second decision letter

MS Title: AXL receptor tyrosine kinase regulates Golgi organization and function via an adhesion-Arf1 signalling axis in breast and lung cancer cell lines

Authors: Nagaraj Balasubramanian, Prachi Joshi, Arnav Saha, Radhika Malaviya, Debiprasad Panda, Manojee Pattanayak, Grishma Mehta and Vibha Singh

Thank you for submitting your revised article to us (and apologies for the delay in getting back to you with a decision - I needed time to assess the revised article alongside the companion article that you also submitted). I have now had the chance to read through both articles in detail. In order for us to reconsider the above paper for publication, you will need to update your paper with the additional data and changes you indicated in your rebuttal letter/revised article. Once you send the revised paper back to us, we will then send it back to the reviewers for re-review. As you have requested that this paper be submitted back to back with another paper, please make sure these two studies are edited to be in line with each other and to ensure that there is no overlap in figures, etc. I look forward to seeing the revised papers.

Please upload both a 'clean' version of your Word file, along with a highlighted version clearly showing where you have made changes made in the revised manuscript. Please avoid using 'Tracked changes' in Word files as these are lost in PDF conversion.

I should be grateful if you would also provide a point-by-point response detailing how you have dealt with the points raised by the reviewers in the 'Response to Reviewers' box. Please attend to all of the reviewers' comments. If you do not agree with any of their criticisms or suggestions please explain clearly why this is so.

Second revision

Author response to reviewers' comments

Reviewer 1: SUMMARY OF THE ADVANCE MADE IN THIS PAPER AND ITS POTENTIAL SIGNIFICANCE TO THE FIELD

This revised manuscript describes a potential connection between the transmembrane tyrosine kinase Axl and the small GTPase Arf1 in which the authors speculate that Axl somehow regulates Arf1 activity and consequently Golgi organization and function. Interestingly Axl does appear to localize at least partially to the Golgi across several cancer cell lines (although it is also clearly elsewhere), and co-precipitates with active Arf1, suggesting that the two proteins may interact (although this is not shown directly). Pharmacological inhibition or knockdown of Axl does reduce Arf1 activity in breast cancer MB- MDA-231 cells, but not in lung cancer A549 cells, suggesting that the relationship between the two proteins is not generalizable. The authors propose that Axl activity is at least partially controlled by adhesion and that this by extension controls Arf1 activity and therefore Golgi organization. Their data suggest that this is not a general phenomenon, but occurs in a subset of cancers through mechanisms that remain unidentified.

We do agree with the reviewer that the AXL-Arf1 relationship is not universally conserved across cancer types. Important aspects such as - Golgi localization of AXL, active Arf1- mediated rescue of Golgi phenotype on AXL targeting, and AXL targeting mediated changes in Golgi-associated function (effect on tubulin acetylation) - are conserved in breast and lung cancer cells. Our goal in this study was hence to define the similarities and differences in how adhesion-AXL-Arf1-Golgi axis operates in MDAMB231 breast cancer cells, and A549 lung cancer cells.

Similarities in AXL-Arf1 regulation/crosstalk in breast cancer (MDAMB231) and Lung cancer (A549) cells

- AXL localises to the Golgi in both cell types (Fig 2E & F).
- AXL knockdown disrupts the Golgi organization in both cell types (Fig 3G, 6H).
- R428 treatment causes Golgi disorganisation accompanied by a drop in Arf1 activation (MDAMB231) or loss of its Golgi localization (MDAMB231 and A549) (Fig 4C, SF4 & 7G, H).
- R428 treatment mediated Golgi disorganisation is reversed by Q71L -Arf1 expression in SA MDAMB231 cells (Fig 4G) and non-adherent A549 cells (Fig 7H).

Differences in AXL-Arf1 crosstalk in breast cancer (MDAMB231) and Lung cancer (A549) cells

- In stable adherent MDAMB231 and A549 cells R428 treatment causes a loss of AXL localization at the Golgi, an increase in p702 AXL levels. This is accompanied by a drop in Arf1 activation in MDAMB231 cells, that does not happen in A549 cells. This we think causes the Golgi to disorganise in SA MDAMB231 but not SA A549 cells on R428 treatment.
- Loss of adhesion mediated regulation of AXL (loss of its localization at the Golgi and regulation of pY702 AXL levels) are seen in MDAMB231 cells, but not in A549 cells. This could be the reason the Golgi stays organised in A549 cells on loss of adhesion (unlike MDAMB231 cells).

It is likely that the AXL-Golgi pathway is regulated differently across cancer cell types, as suggested by our observations, and we aimed to capture this possibility in the manuscript. We would be happy to revise the Discussion and the schematic model to present this aspect more clearly.

Previous reviews indicated that the inclusion of multiple cancer cell lines (4 in total) was confusing, largely because the data were contradictory, and that even if these contradictions could be attributed to distinct modes of regulation, no mechanistic explanation for how Axl might control Arf1 function and/or Golgi organization was provided. Unfortunately, these issues remain in the revised manuscript. Despite 23 pages of explanation in their rebuttal, it is still not clear how (or if) Axl interacts with Arf1 or how Axl might control Arf1 activity. Despite the authors' claims that Axl selectively binds Arf1-GTP, its association with Arf1 in activity-related pulldowns does not appear to change, even in the presence of GTPγS, and in A549 cells remains the same even as Arf1 activity decreases by 50% (Fig. 7B, C). Thus, it is difficult to assess whether and how Axl might regulate Arf1 activity.

We acknowledge that a comprehensive mechanism explaining how AXL regulates Arf1 function and/or Golgi organization is not fully addressed in the current manuscript. This has been an important question for us as well. The data presented in the manuscript, when considered together with additional studies we have recently carried out, begin to provide a clearer framework for what may underlie this mechanism.

As reported in the earlier version of the manuscript, a direct association between AXL and Arf1 could contribute to this regulatory mechanism. Our initial hypothesis that these proteins might interact arose from experiments in which their association was assessed in the GST-GGA3 pulldown assay, leading us to speculate that the interaction and their spatial localization at the Golgi might be crucial for the regulatory crosstalk. However, our experiments using GTPγS suggest that Arf1 activation status may not be essential for this association. We have since removed this data from the MS. The presence of a direct interaction between AXL and Arf1 could still be important, as such an association may allow these two proteins to regulate each other's activation and localization at the Golgi and thereby contribute to AXL-mediated regulation of Golgi organization.

The possibility that an AXL-dependent signalling pathway regulating Arf1, and thereby Golgi organization, is something we had been actively exploring. Previous studies from our group (Rajeshwari et al., 2023; Singh et al., 2018) as well as others (Altan-Bonnet et al., 2003) have established that Arf1 activation and localization is critical for maintaining Golgi organization. In this context, our new data now suggest the existence of a potential AXL-AMPK-GBF1-Arf1 signalling axis in MDA-MB-231 cells. Several components of this pathway and supporting observations were already presented in the earlier version of the manuscript.

In the current version of the manuscript, in MDA-MB-231 cells, we show that R428 treatment leads to an increase in AXL pY702 phosphorylation (Fig. 3D), a response that has also been reported in another study (Chen et al., 2018) and is something we are independently investigating further. While the levels of the Arf GEF GBF1 remain unchanged upon R428 treatment (Fig. S4E), independent targeting of Arf1 activation through pharmacological inhibition of its GBF1 using Golgicide A resulted in Golgi disorganization in MDA-MB-231 cells, hinting at a possible role for GBF1 in the AXL-mediated regulation of Golgi organization. Importantly, similar signalling changes are also observed upon loss of adhesion in MDA-MB-231 cells (Fig. 5A, C, D, S5D), suggesting that this regulatory axis may operate under physiologically relevant conditions.

Figure: Representative blots and quantification showing AXL phosphorylation (pY702), Akt activation (pS473), AMPK activation (pThr172) (New data), total GBF1 levels, and Arf1 activation in MDA-MB-231 cells under DMSO vs R428 treatment and under stable adherent (SA) vs suspended (SUS) conditions.

In addition, our new data shows that AMPK activation (pThr172) increases in a concentration-dependent manner following R428 treatment as well as upon loss of adhesion in MDA-MB-231 cells (*unpublished data shown above*). Notably, AMPK has been reported to phosphorylate GBF1 (at Thr1337), leading to its displacement from the Golgi (Freemantle et al., 2024; Mao et al., 2013), which could in turn affect Arf1 activation at the Golgi. Together, these observations support the possibility of an **AXL-AMPK-GBF1-Arf1 signalling axis** that contributes to the AXL mediated regulation of Golgi organization.

To further examine this potential regulatory pathway, we carried out the following experiments:

(1) Since AMPK activation increased upon R428 treatment, we tested whether inhibition of AMPK could reverse the R428-mediated disruption of Golgi organization. Using the AMPK inhibitor Compound C, we found that inhibition of AMPK partially restored Golgi organization in MDAMB231 cells treated with R428.

(2) As detecting GBF1 phosphorylation was limited by the availability of a reliable antibody, we instead examined GBF1 localization at the Golgi, which is known to be regulated by its phosphorylation status (Freemantle et al., 2024; Mao et al., 2013). Using GM130 as a Golgi marker, we observed that R428 treatment reduced the overlap between GBF1 and GM130, consistent with Golgi disorganization. Notably, this effect was reversed upon Compound C treatment, concomitant with Golgi re-organization. These changes were quantitatively reflected by corresponding differences in the Pearson's coefficient of colocalization between GBF1 and GM130.

Figure (New): R428-induced Golgi disorganization and reduced GBF1-GM130 colocalization in MDA-MB- 231 cells are partially reversed by AMPK inhibition (Compound C).

Together, these observations support the presence of a potential **AXL-AMPK-GBF1-Arf1 signalling pathway** through which AXL may regulate Golgi organization and function.

An additional layer of complexity involves Axl-related changes in tubulin acetylation, which might be expected to affect Golgi organization independent of Arf1, but which is never explained or integrated into any proposed mechanism. What is known about the relationship between Axl and post-translational modifications of tubulin? Is tubulin acetylation upstream or downstream of Golgi organization? Because Golgi clustering near the MTOC requires microtubule motors, one might expect that its organization would be disrupted if motor activity related to tubulin acetylation is perturbed.

We appreciate the reviewer's concern regarding the mechanistic integration of AXL-dependent changes in tubulin acetylation and Golgi organization. We agree that our initial framing did not sufficiently clarify whether tubulin acetylation is causal to Golgi organization or a downstream consequence. At present, there is no direct evidence in the literature demonstrating that AXL regulates tubulin post-translational modifications such as α -tubulin acetylation.

Prior literature supports a bi-directional relationship between Golgi integrity and microtubule (MT) acetylation rather than a strictly linear pathway. Notably, Deakin et al. demonstrated that Golgi organization itself can regulate MT acetylation in MDA-MB-231 cells through modulation of HDAC6 activity, a major α -tubulin deacetylase (Deakin & Turner, 2014). Conversely, studies in other cell types have shown that perturbations in MT acetylation can also influence Golgi organization, highlighting a reciprocal regulatory relationship between these two processes (Wu et al., 2016; Zhang et al., 2019). Consistent with this framework, an independent study from our laboratory (Chakraborty et al., 2025); accepted for publication in *Traffic*) shows that in WT-MEFs, microtubule acetylation decreases upon loss of adhesion, a condition in which the Golgi becomes disorganized, and is restored upon re-adhesion, when the Golgi reorganizes. These findings further suggest that an organized Golgi can support and maintain acetylated microtubule populations in its vicinity.

In agreement with these observations, in MDA-MB-231 cells, loss of adhesion-mediated Golgi disorganization is also accompanied by a marked reduction in tubulin acetylation (Fig. 5H), like what is observed during stiffness-dependent Golgi reorganization (Saha et al., 2026).

Figure: Tubulin-acetylation levels in MDA-MB-231 cells on loss of adhesion mediated disorganisation of the Golgi (Fig. 5H in MS) and matrix-stiffness dependent re-organisation of the Golgi (Refer Fig. 1E- Saha et al., 2026)

Further, we find that disrupting Golgi organization using R428, Golgicide A, siRNA-mediated knockdown of AXL or Arf1, or combined targeting of these pathways is consistently accompanied by a concomitant reduction in tubulin acetylation (Saha et al., 2026; Fig. 7B & D)

Figure: Acetylated tubulin and total tubulin levels in MDA-MB-231 cells treated with DMSO, R428, Golgicide A (GCA), or the combination of R428+GCA, as well as in cells subjected to siRNA-mediated knockdown of AXL, Arf1, or combined AXL+Arf1 knockdown, compared with control cells. (Refer Fig. 7B & D; Saha et al., 2026)

Additional evidence from our independent study (Saha et al., 2026), further supports this relationship. In AXL-expressing MCF7 cells, where Golgi organization is restored, GCA-mediated inhibition of Arf1 results in Golgi disorganization accompanied by a significant reduction in tubulin acetylation (Refer Fig. 6D; Saha et al., 2026). Together with the observations described above, this finding further strengthens the association between Golgi organization and tubulin acetylation and suggests that this regulation occurs in the presence of AXL along an Arf1-dependent pathway.

Figure: GCA-mediated disruption of Golgi organization in AXL-expressing MCF7 cells is reflected in their Golgi distribution profile and is accompanied by a reduction in tubulin acetylation in these cells. (Refer Fig. 6C & D; Saha et al., 2026)

Figure provided in confidence to reviewers has been removed.

We therefore interpret the AXL-linked changes in tubulin acetylation as an outcome of its regulation of Arf1 activation and Golgi organization. The persistence of this correlation in MDA-MB-231 cells upon loss of adhesion and under conditions of altered matrix stiffness further suggests that tubulin acetylation serves as a sensitive and relevant marker of Golgi organization in these

cells.

A potentially related issue is that the pharmacological Axl inhibitor R428 inhibits Axl activity within 30 minutes of addition, yet the authors do all of their assays after 12 h in the continuous presence of the drug. Thus, it is very likely that at least some of the effects of Axl inhibition (or knockdown) are very indirect.

We thank the reviewer for raising this important point. The AXL inhibition experiments performed using R428 (1 μ M) for 12 hours are consistent with treatment conditions employed in several previous studies investigating AXL signalling and its downstream cellular outcomes, and this precedent informed our choice of experimental window. Representative studies using R428 treatment durations of 12 hours or longer to examine AXL-dependent signalling and cellular responses are listed below.

1. Chen F, Song Q, Yu Q. Axl inhibitor R428 induces apoptosis of cancer cells by blocking lysosomal acidification and recycling independent of Axl inhibition. *Am J Cancer Res.* 2018 Aug 1;8(8):1466-1482. PMID: 30210917; PMCID: PMC6129480. (12-36 hrs of treatment at concentrations-2.5 μ M)
2. Qi, L.-F.-R., Liu, S., Liu, Y.-C., Li, P., & Xu, X. (2021). Ganoderic Acid A Promotes Amyloid- B Clearance (In Vitro) and Ameliorates Cognitive Deficiency in Alzheimer's Disease (Mouse Model) through Autophagy Induced by Activating Axl. *International Journal of Molecular Sciences*, 22(11), 5559. <https://doi.org/10.3390/ijms22115559> - (Cells were pre- treated with R428 (5 μ M) before stimulation., Signalling was evaluated after 6-12 h treatments.)
3. Pinato, D.J., Brown, M.W., Trousil, S. *et al.* Integrated analysis of multiple receptor tyrosine kinases identifies Axl as a therapeutic target and mediator of resistance to sorafenib in hepatocellular carcinoma. *Br J Cancer* 120, 512-521 (2019). <https://doi.org/10.1038/s41416-018-0373-6> (Cells treated with R428- 1-3 μ M for 6 h and 24 h)
4. Woo, S. M., Min, K.-j., Seo, S. U., Kim, S., Kubatka, P., Park, J.-W., & Kwon, T. K. (2019). Axl Inhibitor R428 Enhances TRAIL-Mediated Apoptosis Through Downregulation of c- FLIP and Survivin Expression in Renal Carcinoma. *International Journal of Molecular Sciences*, 20(13), 3253. <https://doi.org/10.3390/ijms20133253> (Cancer cells treated with R428 (3-5 μ M) for ~24 h)
5. Han S, Wang Y, Ge C, Gao M, Wang X, Wang F, Sun L, Li S, Dong T, Dang Z, Cui W, Zhang G, Liu N. Pharmaceutical inhibition of AXL suppresses tumor growth and invasion of esophageal squamous cell carcinoma. *Exp Ther Med.* 2020 Nov;20(5):41. doi: 10.3892/etm.2020.9169. Epub 2020 Sep 2. PMID: 32952632; PMCID: PMC7480165. (Cells treated with R428 (\approx 1 μ M) for 24 hrs or more).

If the Golgi disorganization observed upon R428 treatment were primarily due to non-specific effects of the inhibitor, similar outcomes would not necessarily be expected upon siRNA-mediated knockdown of AXL. However, many of the phenotypes observed following R428 treatment are also recapitulated upon AXL knockdown using siRNA, further suggesting that the effects of R428 in our experiments are unlikely to arise from non-specific effects.

- **Golgi organization:** (Fig. 3G, 6H) - MDAMB231 and A549
- **Tubulin acetylation:** (Fig. 3I) - MDAMB231
- **Arf1 activation:** (Fig. 4E) - MDAMB231
- **Akt activation:** (Fig. S3E) - MDAMB231.

Overall, this is a potentially interesting story that lacks mechanistic insight. The authors attempt to explain away conflicting data with hypotheses that either don't make sense or are not supported by any evidence.

We appreciate the reviewer's perspective. Throughout the revised manuscript, we have been careful to avoid unsupported or overly speculative claims, and we explicitly acknowledge instances where the underlying mechanism remains unclear. Rather than over-interpreting conflicting data, we have presented these findings transparently and framed them as open questions that require further investigation. We agree that aspects of this regulatory pathway are not entirely conserved between cancer cell types and we have been careful to call this out and

present the variation with the intent that captures the AXL-Golgi regulatory pathway more accurately.

SUGGESTIONS TO AUTHORS

Major comments [Please request additional experiments only if they are essential for supporting the conclusions; authors should be encouraged to highlight any claims that are preliminary or speculative, or to discuss any pitfalls or alternative interpretations in a 'Limitations' section]

We appreciate this guidance and will avoid adding non-essential experiments, instead clearly distinguishing preliminary or speculative interpretations, and revising the Discussion to better highlight limitations, alternative explanations, and unresolved mechanistic questions.

Reviewer 2:

SUMMARY OF THE ADVANCE MADE IN THIS PAPER AND ITS POTENTIAL SIGNIFICANCE TO THE FIELD

In this manuscript, the authors set out to characterise proteins that regulate Golgi morphology in a manner dependent on the adhesion status of cancer cells. They choose AXL, a plasma membrane (PM) receptor tyrosine kinase for their study. They show that AXL localizes to the Golgi, where it interacts with Arf1-GTP. The authors show that there is a decrease in phosphorylation of Akt and an increase in phosphorylation of AXL upon treatment of MDAMB231 cancer cells with the AXL inhibitor R428. They show that the Golgi is disorganized upon treatment of these cells with R428, but that MCF7 cells that do not express AXL do not undergo a change in Golgi organization upon R428 treatment. These and other experiments lead the authors to conclude that upon loss of adhesion, AXL is regulating Golgi morphology through effects on Golgi-localized Arf1-GTP.

SUGGESTIONS TO AUTHORS

The authors have addressed my concerns in this revised manuscript, adding a number of additional experiments. I have no further comments.

We thank the reviewer for their positive assessment and for acknowledging that the additional experiments have addressed their concerns. We are grateful for their constructive feedback, which has helped strengthen both the mechanistic clarity and the overall impact of the manuscript.

References

- Altan-Bonnet, N., Phair, R. D., Polishchuk, R. S., Weigert, R., & Lippincott-Schwartz, J. (2003). A role for Arf1 in mitotic Golgi disassembly, chromosome segregation, and cytokinesis. *Proceedings of the National Academy of Sciences of the United States of America*, 100(23), 13314-13319. <https://doi.org/10.1073/pnas.2234055100>
- Chakraborty, A., Pitke, S. M., Rajeshwari, B., Dasgupta, A., Buwa, N., Behera, R., Jayakrishnan, M., & Balasubramanian, N. (2025). *KIF5B and Dynein regulate adhesion-dependent Golgi organization and microtubule acetylation*. <https://doi.org/10.1101/2025.05.11.653231>
- Freemantle, J. B., Towler, M. C., Hudson, E. R., Macartney, T., Zwirek, M., Liu, D. J. K., Pan, D. A., Ponnambalam, S., & Hardie, D. G. (2024). AMPK associates with and causes fragmentation of the Golgi by phosphorylating the guanine nucleotide exchange factor GBF1. *Journal of Cell Science*, 137(24). <https://doi.org/10.1242/jcs.262182>
- Mao, L., Li, N., Guo, Y., Xu, X., Gao, L., Xu, Y., Zhou, L., & Liu, W. (2013). AMPK phosphorylates GBF1 for mitotic Golgi disassembly. *Journal of Cell Science*. <https://doi.org/10.1242/jcs.121954>
- Rajeshwari, B. R., Shah, N., Joshi, P., Madhusudan, M. S., & Balasubramanian, N. (2023). Kinetics of Arf1 inactivation regulates Golgi organisation and function in non-adherent fibroblasts. *Biology Open*, 12(4). <https://doi.org/10.1242/bio.059669>
- Saha, A., Sherkhane, T., & Balasubramanian, N. (2026). Differential AXL expression and Arf1 regulation control stiffness-dependent Golgi organization in breast cancer cells. *Journal of Cell Science*, 139(2). <https://doi.org/10.1242/jcs.263956>

- Singh, V., Erady, C., & Balasubramanian, N. (2018). Cell-matrix adhesion controls Golgi organization and function through Arf1 activation in anchorage-dependent cells. *Journal of Cell Science*, 131(16). <https://doi.org/10.1242/jcs.215855>
- Wu, J., de Heus, C., Liu, Q., Bouchet, B. P., Noordstra, I., Jiang, K., Hua, S., Martin, M., Yang, C., Grigoriev, I., Katrukha, E. A., Altelaar, A. F. M., Hoogenraad, C. C., Qi, R. Z., Klumperman, J., & Akhmanova, A. (2016). Molecular Pathway of Microtubule Organization at the Golgi Apparatus. *Developmental Cell*, 39(1), 44-60. <https://doi.org/10.1016/j.devcel.2016.08.009>
- Zhang, J., Tan, J., Hu, Z., Chen, C., & Zeng, L. (2019). HDAC6 Inhibition Protects against OGDR-Induced Golgi Fragmentation and Apoptosis. *Oxidative Medicine and Cellular Longevity*, 2019, 1-12. <https://doi.org/10.1155/2019/6507537>

Third decision letter

MS Title: AXL receptor tyrosine kinase regulates Golgi organization and function via an adhesion-Arf1 signalling axis in breast and lung cancer cell lines

Authors: Nagaraj Balasubramanian, Prachi Joshi, Arnav Saha, Radhika Malaviya, Debiprasad Panda, Manojjeet Pattanayak, Grishma Mehta and Vibha Singh

We have now reached a decision on the above manuscript.

As you will see from their reports, the reviewers raise a number of substantial criticisms that prevent me from accepting your paper for publication.

I am very sorry to give you such disappointing news, but it takes a very enthusiastic recommendation by the referees for a manuscript to be accepted.

I do hope you find the comments of the reviewers helpful in allowing you to revise the manuscript for submission elsewhere, and many thanks for sending your work to us.

Comments from the Reviewers:

Reviewer 1: SUMMARY OF THE ADVANCE MADE IN THIS PAPER AND ITS POTENTIAL SIGNIFICANCE TO THE FIELD

This revised manuscript describes a potential connection between the transmembrane tyrosine kinase Axl and the small GTPase Arf1 in which the authors speculate that Axl somehow regulates Arf1 activity and consequently Golgi organization and function. Interestingly Axl does appear to localize at least partially to the Golgi across several cancer cell lines (although it is also clearly elsewhere), and co-precipitates with active Arf1, suggesting that the two proteins may interact (although this is not shown directly). Pharmacological inhibition or knockdown of Axl does reduce Arf1 activity in breast cancer MB-MDA-231 cells, but not in lung cancer A549 cells, suggesting that the relationship between the two proteins is not generalizable. The authors propose that Axl activity is at least partially controlled by adhesion and that this by extension controls Arf1 activity and therefore Golgi organization. Their data suggest that this is not a general phenomenon, but occurs in a subset of cancers through mechanisms that remain unidentified.

Previous reviews indicated that the inclusion of multiple cancer cell lines (4 in total) was confusing, largely because the data were contradictory, and that even if these contradictions could be attributed to distinct modes of regulation, no mechanistic explanation for how Axl might control Arf1 function and/or Golgi organization was provided. Unfortunately, these issues remain in the revised manuscript. Despite 23 pages of explanation in their rebuttal, it is still not clear how (or if) Axl interacts with Arf1 or how Axl might control Arf1 activity. Despite the authors' claims that Axl selectively binds Arf1-GTP, its association with Arf1 in activity-related pulldowns does not appear to change, even in the presence of GTPγS, and in A549 cells remains the same even as Arf1 activity

decreases by 50% (Fig. 7B, C). Thus it is difficult to assess whether and how Axl might regulate Arf1 activity.

An additional layer of complexity involves Axl-related changes in tubulin acetylation, which might be expected to affect Golgi organization independent of Arf1, but which is never explained or integrated into any proposed mechanism. What is known about the relationship between Axl and post-translational modifications of tubulin? Is tubulin acetylation upstream or downstream of Golgi organization? Because Golgi clustering near the MTOC requires microtubule motors, one might expect that its organization would be disrupted if motor activity related to tubulin acetylation is perturbed. A potentially related issue is that the pharmacological Axl inhibitor R428 inhibits Axl activity within 30 minutes of addition, yet the authors do all of their assays after 12 h in the continuous presence of the drug. Thus it is very likely that at least some of the effects of Axl inhibition (or knockdown) are very indirect.

Overall, this is a potentially interesting story that lacks mechanistic insight. The authors attempt to explain away conflicting data with hypotheses that either don't make sense or are not supported by any evidence.

SUGGESTIONS TO AUTHORS

Major comments [Please request additional experiments only if they are essential for supporting the conclusions; authors should be encouraged to highlight any claims that are preliminary or speculative, or to discuss any pitfalls or alternative interpretations in a 'Limitations' section]

Minor comments

Reviewer 2: SUMMARY OF THE ADVANCE MADE IN THIS PAPER AND ITS POTENTIAL SIGNIFICANCE TO THE FIELD

In this manuscript, the authors set out to characterise proteins that regulate Golgi morphology in a manner dependent on the adhesion status of cancer cells. They choose AXL, a plasma membrane (PM) receptor tyrosine kinase for their study. They show that AXL localizes to the Golgi, where it interacts with Arf1-GTP. The authors show that there is a decrease in phosphorylation of Akt and an increase in phosphorylation of AXL upon treatment of MDAMB231 cancer cells with the AXL inhibitor R428. They show that the Golgi is disorganized upon treatment of these cells with R428, but that MCF7 cells that do not express AXL do not undergo a change in Golgi organization upon R428 treatment. These and other experiments lead the authors to conclude that upon loss of adhesion, AXL is regulating Golgi morphology through effects on Golgi-localized Arf1-GTP.

SUGGESTIONS TO AUTHORS

The authors have addressed my concerns in this revised manuscript, adding a number of additional experiments. I have no further comments.

Transfer to Biology Open

First decision letter

MS ID#: bio.062581

MS Title: AXL receptor tyrosine kinase regulates Golgi organization and function via an adhesion-Arf1 signalling axis in breast and lung cancer cell lines

Authors: Nagaraj Balasubramanian, Prachi Joshi, Arnav Saha, Radhika Malaviya, Debiprasad Panda, Manojjeet Pattanayak, Grishma Mehta and Vibha Singh

Thank you for putting forth the time and effort to address concerns raised in the second round of review. I am happy to tell you that your manuscript has been accepted for publication in Biology Open on the condition that the title be modified and pending our standard publication integrity checks. The revised title should read "AXL receptor tyrosine kinase regulates Golgi organization and function via an adhesion-Arf1 signalling axis in breast and lung cancer cell lines". It was accepted on 31st March 2026.